# Quantifying Treatment Effects: Estimating Risk Ratios via Observational Studies

**Ahmed Boughdiri** [1]   **Julie Josse** [1]   **Erwan Scornet** [2]

## Abstract

The Risk Difference (RD), an absolute measure of effect, is widely used and well-studied in both randomized controlled trials (RCTs) and observational studies. Complementary to the RD, the Risk Ratio (RR), as a relative measure, is critical for a comprehensive understanding of intervention effects: RD can downplay small absolute changes, while RR can highlight them. Despite its significance, the theoretical study of RR has received less attention, particularly in observational settings. This paper addresses this gap by tackling the estimation of RR in observational data. We propose several RR estimators and establish their theoretical properties, including asymptotic normality and confidence intervals. Through analyses on simulated and real-world datasets, we evaluate the performance of these estimators in terms of bias, efficiency, and robustness to generative data models. We also examine the coverage and length of the associated confidence intervals. Due to the non-linear nature of RR, influence function theory yields two distinct efficient estimators with different convergence assumptions. Based on theoretical and empirical insights, we recommend, among all estimators, one of the two doubly-robust estimators, which, intriguingly, challenges conventional expectations.

## 1. Introduction

**Treatment effect estimation in trials.**   Modern evidence-based medicine prioritizes Randomized Controlled Trials (RCTs) as the cornerstone of clinical evidence. Randomization in RCTs allows for the quantification of the average treatment effect (ATE) by removing confounding influences from extraneous or undesirable factors. The medical guideline CONSORT (Moher et al., 2010) recommends reporting the treatment effect with relative measures like the Risk Ratio (RR) along with absolute measures like the Risk Difference (RD) to provide a more comprehensive understanding of the effect and its implications, as neither measure alone offers a complete picture. Indeed, selecting one measure over another carries several implication. For instance, with 3% baseline mortality reduced to 1% by treatment, RD shows a 2% drop, while RR shows controls have three times the risk: RD suggests a small effect; RR highlights a larger one. In addition, (Naylor et al., 1992) and (Forrow et al., 1992) demonstrated that physicians' inclination to treat patients, based on their perception of therapeutic impact, is influenced by the scale utilized to present clinical effects. Finally, the treatment effect may be heterogeneous in one scale, i.e. the treatment effect varies according to patient characteristics, but homogeneous in another scale (Rothman, 2011), which significantly disrupts interpretation. (Colnet et al., 2024) discusses causal measure properties with a focus on generalization of the treatment effect from a trial to a target population.

Consequently, both RD and RR measures are widely used in the analysis of clinical trial data as explained in Malenka et al. (1993), Sinclair & Bracken (1994) and Nakayama et al. (1998). The Risk Ratio is particularly relevant in scenarios where outcomes are always either positive or negative (Malenka et al., 1993) and in cases where both expected potential outcomes being compared are small, as it is more stable and interpretable than the RD. Furthermore, in cases of rare treatment occurrences, the Risk Ratio closely approximates the Odds Ratio (OR), further enhancing its relevance and utility in clinical analyses (Schechtman, 2002). Barratt et al. (2004) recommend using the RR from clinical trials with an estimation of the individual patient baseline to provide the right treatment.

**Treatment effect estimation in observational data.**   Despite being the gold standard to assess treatment effects, RCTs may face limitations due to stringent eligibility criteria, unrealistic real-world compliance, short study durations, and limited sample sizes. Medical journals such as JAMA (Bibbins-Domingo, 2024) and others (Hernan & Robins,

---
[*]Equal contribution  [1]INRIA Sophia-Antipolis  [2]Sorbonne Université and Université Paris Cité. Correspondence to: Ahmed Boughdiri <ahmed.boughdiri@inria.fr>.

*Proceedings of the 42nd International Conference on Machine Learning*, Vancouver, Canada. PMLR 267, 2025. Copyright 2025 by the author(s).

2016) have advocated the use of real-world data, often referred to as observational data, to provide additional sources of evidence. These data sets are typically less expensive to collect, more representative of the target population, and usually encompass large sample sizes.

In the context of observational studies, various estimators exist to measure the treatment effect, mainly on an absolute scale. Different methods such as re-weighting using Inverse Probability Weighting (IPW) (Hirano et al., 2003), outcome modeling with the G-formula, and doubly-robust approaches like Augmented Inverse Probability Weighting (AIPW) (Robins, 1986) aim to estimate the RD while handling confounding effects.

However, to the best of our knowledge, there exist no work proposing or using estimators of the ATE measured with the Risk Ratio in observational studies based on (non-)parametric estimation (G-formula or AIPW), nor derivations of their theoretical properties. A clear gap exists, needing robust estimators and analyses to improve treatment effect assessments and support medical recommendations in observational studies. Noteworthily, compared to the Risk Difference, studying the Risk Ratio induces additional technical difficulties, due to its non-linear nature.

**Contributions.** In this paper, we propose and analyze different estimators of the Risk Ratio in observational studies. Considering first the well studied RCT setting in Section 2, we analyze the first RR estimator introduced by Cornfield (1956), establishing a Central Limit Theorem and asymptotic confidence intervals. We prove that in RCT adjusting for covariates or estimating the probability of being treated reduces the estimator variance. As the probability of being treated varies across individuals in observational studies, the above estimator is no longer valid. In Section 3, we detail different estimators that can be used for estimating the RR in observational studies: Inverse Propensity Weighting (RR-IPW), G-formula (RR-G) and two doubly-robust estimators. For the first two, we prove their asymptotic normality when the nuisance functions (surface responses and propensity score) are known or estimated via parametric models. Our analyses show that estimating nuisance function increases the variance of RR-G and decreases that of RR-IPW. Besides, using influence function theory (see, e.g., Kennedy, 2022), we derive two doubly-robust estimators, RR-OS, for one-step correction and RR-AIPW based on estimating equations. Contrary to the RD, due to the non-linearity of the RR, both estimators differ. We prove that they are both asymptotically unbiased and have the minimal variance among all asymptotically unbiased estimators. Surprisingly, the RR-AIPW estimator turns out to be a plug-in version of AIPW estimators for both the numerators and denominators, which requires weaker assumptions than RR-OS to be asymptotically normal. For all estimators, our asymptotic

analyses allow us to build asymptotic confidence intervals. In Section 4, we evaluate all estimators on observational data, and study the empirical properties of confidence intervals in terms of coverage and lengths. In Section 5, we extend our analysis to a semi-synthetic and a real-world dataset. We recommend the RR-AIPW due to $(i)$ less stringent assumptions needed to ensure asymptotic normality, $(ii)$ empirical performances (consistent and with a small asymptotical variance), $(iii)$ the ease of implementation, as a ratio of two simple AIPW estimators.

**Related work - Estimation of RR.** To the best of our knowledge, Cornfield (1956) was the first to propose an estimate of the RR, together with exact and asymptotic confidence intervals, for binary responses, in a RCT scenario, followed by Kupper et al. (1975); Katz et al. (1978); Bailey (1987); Morris & Gardner (1988); Sato (1992). Considering a logistic model, Schouten et al. (1993) propose a RR estimator. Later on, exact confidence intervals were derived by Wang & Shan (2015). Recently, Inverse Propensity Weighting schemes have been used in different study designs to estimate the Odds Ratios (Staus et al., 2022) or directly the RR in some simple settings (Hernán & Robins, 2006). Besides, pseudo-Poisson and pseudo normal distribution have been proposed with IPW strategies to estimate RD and RR in clinical trials (Noma et al., 2023). Other methods include G-formula approaches (Dukes & Vansteelandt, 2018) and semiparametric density estimation (Kennedy et al., 2023).

In observational studies, one can mention the work of Richardson et al. (2017),Yadlowsky et al. (2021) and Shirvaikar & Holmes (2023) who focus on estimators of the conditional average treatment effect (CATE) for the RR. Curth et al. (2020) introduces an "IF-learning" approach with pseudo outcome regression and derive the influence function for the CATE of the RR. However, since the expectation of a ratio is not the ratio of the expectations, CATE estimations do not directly yield ATE for the RR. This departs from the RD, for which the ATE is simply the expectation of the CATE.

## 2. A Well-Known Risk Ratio Estimator in RCT

**Problem setting** Following the potential outcome framework (see Rubin, 1974; Splawa-Neyman et al., 1990), we consider the random variables $(X, T, Y^{(0)}, Y^{(1)})$, where $X \in \mathbb{R}^p$ denotes covariates describing a patient, $T$ is the treatment assignment ($T = 1$ when the treatment is given to an individual, $T = 0$ otherwise) and $Y^{(0)}$ (resp. $Y^{(1)}$) is the outcome of interest, describing the status of a patient without treatment (and with treatment respectively). In practice, we do not have access simultaneously to $Y^{(1)}$ and $Y^{(0)}$, and

we only observe

$$Y = TY^{(1)} + (1-T)Y^{(0)}.$$

Causal effect measures are functions of the joint distribution of potential outcomes (see Pearl, 2009). In particular, the Risk Difference (RD) and the Risk Ratio (RR) contrast the two states as followed

$$\tau_{\text{RD}} = \mathbb{E}[Y^{(1)}] - \mathbb{E}[Y^{(0)}] \quad \text{and} \quad \tau_{\text{RR}} = \frac{\mathbb{E}[Y^{(1)}]}{\mathbb{E}[Y^{(0)}]}. \quad (1)$$

The aim of this paper is to propose and study estimators of $\tau_{\text{RR}}$ for binary and continuous outcomes. Indeed, all our theoretical results (except Proposition 3.9) are valid for binary and continuous outcomes. To estimate $\tau_{\text{RR}}$, we assume to be given an i.i.d dataset $(X_1, T_1, Y_1), \ldots, (X_n, T_n, Y_n)$.

The most simple estimator of the risk ratio consists in replacing expectations by empirical means in $\tau_{\text{RR}}$ (Equation 1). Such an estimator has already been proposed outside the causal inference framework, with confidence intervals for the binary case (where $Y \in \{0,1\}$, see Katz et al., 1978; Bailey, 1987). In the potential outcome framework, inspired by the Neyman estimator of the Risk Difference (Splawa-Neyman et al., 1990), we call this estimator the Risk Ratio Neyman estimator.

**Definition 2.1** (**Risk Ratio Neyman estimator**). Let $N_1 = \sum_{i=1}^n T_i$ and $N_0 = n - n_1$. The Risk Ratio Neyman estimator, denoted $\hat{\tau}_{\text{RR,N}}$, is defined as

$$\hat{\tau}_{\text{RR,N}} = \frac{\frac{1}{N_1} \sum_{i=1}^n T_i Y_i}{\frac{1}{N_0} \sum_{i=1}^n (1-T_i) Y_i}, \quad (2)$$

if the denominator is nonzero and 0 otherwise.

With our notation, the 95% confidence interval for $\tau_{\text{RR}}$ for binary outcomes (Katz et al., 1978) takes the form

$$\left[ \hat{\tau}_{\text{RR,N}} \exp(-z_{1-\alpha/2}\hat{\sigma}_n), \hat{\tau}_{\text{RR,N}} \exp(z_{1-\alpha/2}\hat{\sigma}_n) \right], \quad (3)$$

where $z_{1-\alpha/2}$ is the $1-\alpha/2$ quantile of a standard Gaussian $\mathcal{N}(0,1)$ and

$$\hat{\sigma}_n = \sqrt{\frac{1}{\sum_{i=1}^n T_i Y_i} - \frac{1}{N_1} + \frac{1}{\sum_{i=1}^n (1-T_i)Y_i} - \frac{1}{N_0}}. \quad (4)$$

In the sequel, we establish under which theoretical assumptions the RR-N is an accurate estimator of the Risk Ratio in Randomized Clinical Trials.

RCT randomly assign treatment to patients in order to evaluate treatment effects. We focus on the Bernoulli design, one of the most widely used RCT designs (Rubin, 1974; Imbens & Rubin, 2015), where each participant has the same probability $e \in (0,1)$ of being treated, independently of the treatments of others.

**Assumption 2.2** (Bernoulli Trial). We assume:

(i) **Ignorability/Exchangeability:** $T \perp\!\!\!\perp (Y^{(0)}, Y^{(1)})$.

(ii) **SUTVA:** $Y = TY^{(1)} + (1-T)Y^{(0)}$.

(iii) **i.i.d.:** $(X_i, T_i, Y_i^{(0)}, Y_i^{(1)})_{i \in [n]} \overset{\text{i.i.d.}}{\sim} \mathcal{P}$.

(iv) **Trial Positivity:** Each participant has a fixed probability $e \in (0,1)$ of assignment: $\mathbb{P}[T_i = 1] = e$.

To ensure our estimates are valid, we need to guarantee the existence of the ratio we aim to estimate.

**Assumption 2.3** (**Outcome positivity**). We suppose that both $Y^{(0)}$ and $Y^{(1)}$ are squared integrable and that $\mathbb{E}[Y^{(0)}] > 0$.

**Proposition 2.4** (**Asymptotic normality of $\hat{\tau}_{\text{RR,N}}$**). *Grant Assumption 2.2 and Assumption 2.3, the Risk Ratio Neyman estimator is asymptotically unbiased and satisfies*

$$\sqrt{n}\left(\hat{\tau}_{RR,N} - \tau_{RR}\right) \overset{d}{\to} \mathcal{N}\left(0, V_{RR,N}\right) \quad (5)$$

*where*

$$V_{RR,N} = \tau_{RR}^2 \left( \frac{\text{Var}\left(Y^{(1)}\right)}{e\mathbb{E}[Y^{(1)}]^2} + \frac{\text{Var}\left(Y^{(0)}\right)}{(1-e)\mathbb{E}[Y^{(0)}]^2} \right).$$

Proposition 2.4 establishes the asymptotic normality of the RR-N estimator, a simple ratio of mean estimates, which leads to asymptotic confidence intervals (CI). Indeed, according to Proposition 2.4, for all $\alpha \in (0,1)$, a $(1-\alpha)$ asymptotic confidence interval for $\tau_{\text{RR}}$ is given by

$$\left[ \hat{\tau}_{\text{RR,N}} \pm z_{1-\alpha/2} \sqrt{\widehat{V_{\text{RR,N}}}/n} \right], \quad (6)$$

where $\widehat{V_{\text{RR,N}}}$ is any estimation of $V_{\text{RR,N}}$. Throughout this paper, based on the Central Limit Theorems we establish, we will consider such CI. The properties of the different CI are studied in Section 4.2.

Contrary to Katz et al. (1978), Proposition 2.4 is valid for both continuous and binary outcomes. However, considering binary outcomes in Proposition 2.4 leads to an asymptotic confidence interval equivalent to that presented in Katz et al. (1978) (see Appendix 7.2.3). Deriving a Central Limit Theorem for $\log(\hat{\tau}_{\text{RR,N}})$ instead of $\hat{\tau}_{\text{RR,N}}$ would lead to the exact same CI (see Appendix 7.2.4).

**Probability of receiving treatment** As the probability of treatment $e$ is known in an RCT, one could be tempted to consider what we call the Risk Ratio Horvitz-Thomson estimator (in reference of the Risk Difference Horvitz-Thomson estimator, see Horvitz & Thompson, 1952) defined as

$$\hat{\tau}_{\text{RR,HT}} = \frac{\frac{1}{n} \sum_{i=1}^n \frac{T_i Y_i}{e}}{\frac{1}{n} \sum_{i=1}^n \frac{(1-T_i)Y_i}{1-e}} \quad (7)$$

if $\sum_{i=1}^{n} T_i < n$ and 0 otherwise. Indeed, the frequency of treatments assignments in the sample may be different from the actual probability of receiving treatment $e$. Similarly to what Hirano et al. (2003); Hahn (1998); Robins et al. (1992) noticed for the RD, we prove in Appendix 7.2.2 that opting for $\hat{e}$ over $e$ in the Risk Ratio estimator (thereby employing the RR-N instead of the RR-HT) results in a reduced asymptotic variance, with a larger reduction when $e$ is close to zero or one. More precisely, letting $V_{RR,HT}$ the asymptotic variance of $\hat{\tau}_{RR,HT}$, we have

$$V_{RR,N} = V_{RR,HT} - \tau_{RR}^2/e(1-e). \tag{8}$$

# 3. Risk Ratio Estimators in Observational Studies

A key difference between RCTs and observational studies is the handling of confounding variables. If not properly addressed, these can distort the true causal association between exposure and outcome due to their correlation with both. Therefore, estimating the Risk Ratio in observational studies is more complex than in RCTs, as randomization assumptions do not apply (i.e. the propensity score now depends on the covariates $X$).

**Assumption 3.1** (**Observational Study Identifiability Assumptions**). We assume:

(i) **Unconfoundedness/Conditional Exchangeability:** $(Y(0), Y(1)) \perp\!\!\!\perp T \mid X$.

(ii) **Overlap/Positivity:** $\exists \eta \in (0, 1/2]$ such that $\eta \leq \mathbb{P}[T = 1|X] \leq 1 - \eta$ almost surely.

(iii) **SUTVA:** $Y = TY^{(1)} + (1 - T)Y^{(0)}$.

(iv) **i.i.d.:** The data is $(X_i, T_i, Y_i^{(0)}, Y_i^{(1)})_{i \in [n]} \overset{i.i.d.}{\sim} \mathcal{P}$.

Unconfoundedness means that after accounting for known confounding variables, no hidden factors affect both treatment assignment and outcomes. It is a relaxed form of exchangeability.

RR-N and RR-HT estimators cannot be used in the context of observational studies, since they are built on the assumption of a constant propensity score. But the RR-N estimator can be extended to observational studies as follows.

## 3.1. Risk Ratio Inverse Propensity Weighting (RR-IPW)

Treatment effect in observational studies can be estimated via reweighting individuals by the inverse of their propensity score, thus giving more weights to people who are very likely/unlikely to be treated. Such a method, called Inverse Propensity Weighting (IPW, see Hirano et al., 2003) for estimating the Risk Difference, can be straightforwardly extended to build Risk Ratio estimators.

**Definition 3.2** (**RR-IPW**). Grant Assumption 2.3 and Assumption 3.1. Given an estimator $0 < \hat{e}(\cdot) < 1$ of the propensity score $e(x) = \mathbb{P}[T = 1|X = x]$, the Risk Ratio IPW, denoted by $\hat{\tau}_{RR,IPW,n}$, is defined as

$$\hat{\tau}_{RR,IPW} = \frac{\frac{1}{n}\sum_{i=1}^{n}\frac{T_i Y_i}{\hat{e}(X_i)}}{\frac{1}{n}\sum_{i=1}^{n}\frac{(1-T_i)Y_i}{1-\hat{e}(X_i)}}.$$

Proposition 3.3 demonstrates the asymptotic normality of the Oracle Ratio IPW estimator, defined as the RR-IPW but where $\hat{e}(\cdot)$ is replaced by the oracle propensity score $e(\cdot)$.

**Proposition 3.3** (**RR-IPW asymptotic normality**). *Grant Assumptions 2.3 and 3.1. Then the Oracle Risk Ratio IPW is asymptotically unbiased and satisfies*

$$\sqrt{n}\left(\tau_{RR,IPW}^{\star} - \tau_{RR}\right) \overset{d}{\to} \mathcal{N}\left(0, V_{RR,IPW}\right) \tag{9}$$

*where* $\frac{V_{RR,IPW}}{\tau_{RR}^2} = \mathbb{E}\left[\frac{(Y^{(1)})^2}{e(X)\mathbb{E}[Y^{(1)}]^2}\right] + \mathbb{E}\left[\frac{(Y^{(0)})^2}{(1-e(X))\mathbb{E}[Y^{(0)}]^2}\right].$

Note that when the propensity score is constant, one can retrieve the variance of the RR-HT as expected. Note also that the asymptotic variance may be large, due to strata on which the propensity score is close to zero or one. In other words, a correct estimation is difficult when some subpopulations are unlikely to be treated (or untreated).

If we assume a logistic model for the true propensity score and estimate it using maximum likelihood estimation (MLE), the variance of the RR-IPW can be derived.

**Assumption 3.4.** We assume $\mathbb{E}[XX^\top]$ is positive definite, $X$ is Sub-Gaussian, and for all $X \in \mathbb{R}^p$,

$$\mathbb{P}(T = 1|X) = \{1 + \exp(-X^\top \beta_\infty^1 - \beta_\infty^0)\}^{-1},$$

where $\boldsymbol{\beta}_\infty := (\beta_\infty^0, \beta_\infty^1) \in \mathbb{R}^{p+1}$.

For any positive semi-definite matrix $A$ and any vector $X$, let $\|X\|_A = \sqrt{X^\top A X}$.

**Proposition 3.5** (Asymptotics of $\hat{\tau}_{RR,IPW}$ under a logistic model). *Under Assumptions 3.1 and 3.4, the RR-IPW estimator, with the propensity score estimated via MLE, satisfies*

$$\sqrt{n}(\hat{\tau}_{RR,IPW} - \tau_{RR}) \overset{d}{\to} \mathcal{N}(0, V_{RR-MLE}),$$

*with* $V_{RR-MLE} = V_{RR-IPW} - \tau_{RR}^2 \left\| \frac{c_{10}}{\mathbb{E}[Y(0)]} + \frac{c_{01}}{\mathbb{E}[Y(1)]} \right\|_{Q^{-1}}^2,$

*where* $c_{10} = \mathbb{E}[\widetilde{X}\, e(X)Y^{(0)}]$, $c_{01} = \mathbb{E}[\widetilde{X}(1-e(X))Y^{(1)}]$ *and* $Q = \mathbb{E}[e(X)(1-e(X))\widetilde{X}\widetilde{X}^\top]$ *with* $\widetilde{X} := (1, X)$.

The variance of $\hat{\tau}_{RR,IPW}$ is notably smaller than that of the oracle estimator $\tau_{RR,IPW}^{\star}$. While this might initially seem counterintuitive, similar observations have been made in RCT, as highlighted in studies by Hirano et al. (2003); Hahn

(1998); Robins et al. (1992) and in observational studies by Lunceford & Davidian (2004b). Choosing $\hat{e}$ over $e$ in the Risk Ratio estimator (thus using $\hat{\tau}_{\text{RR,IPW}}$ instead of $\tau^\star_{\text{RR,IPW}}$) leads to a reduction in asymptotic variance.

### 3.2. Risk Ratio G-formula estimator (RR-G)

For all $(x, t) \in \mathbb{R}^p \times \{0, 1\}$, let $\mu_{(t)}(x) = \mathbb{E}\left[Y^{(t)}|X = x\right]$ be the surface response of the potential outcome. Assume that we have at our disposal two estimators $\hat{\mu}_{(0)}(\cdot)$ and $\hat{\mu}_{(1)}(\cdot)$ which respectively estimate $\mu_{(0)}(\cdot)$ and $\mu_{(1)}(\cdot)$. We then employ the ratio of these two potential outcome estimations to compute the Risk Ratio. This method, termed the plug-in G-formula or outcome-based modeling, was first introduced by Robins (1986) for the Risk Difference.

**Definition 3.6** (**RR G-formula**). Given two estimators $\hat{\mu}_{(0)}(\cdot)$ and $\hat{\mu}_{(1)}(\cdot)$, the Risk Ratio G-formula estimator, denoted $\hat{\tau}_{\text{RR,G}}$, is defined as

$$\hat{\tau}_{\text{RR,G}} = \frac{\frac{1}{n}\sum_{i=1}^n \hat{\mu}_{(1)}(X_i)}{\frac{1}{n}\sum_{i=1}^n \hat{\mu}_{(0)}(X_i)}, \tag{10}$$

if $\frac{1}{n}\sum_{i=1}^n \hat{\mu}_{(0)}(X_i) \neq 0$ and zero otherwise.

The properties of RR-G depend on the estimators $\hat{\mu}_{(0)}$ and $\hat{\mu}_{(1)}$. We analyze in the following the behavior of Oracle Risk Ratio G-formula estimator defined as $\tau^\star_{\text{RR,G}} = \left(\frac{1}{n}\sum_{i=1}^n \mu_{(1)}(X_i)\right)/\left(\frac{1}{n}\sum_{i=1}^n \mu_{(0)}(X_i)\right)$.

**Proposition 3.7** (**Asymptotic Normality of $\tau^\star_{\text{RR,G}}$**). *Grant Assumptions 2.2 and 2.3. Then, the Oracle Risk Ratio G-formula estimator, $\tau^\star_{RR,G}$, is asymptotically unbiased and satisfies*

$$\sqrt{n}\left(\tau^\star_{RR,G} - \tau_{RR}\right) \xrightarrow{d} \mathcal{N}\left(0, V_{RR,G}\right), \tag{11}$$

*where* $V_{RR,G} = \tau^2_{RR} \text{Var}\left(\frac{\mu_{(1)}(X)}{\mathbb{E}[Y^{(1)}]} - \frac{\mu_{(0)}(X)}{\mathbb{E}[Y^{(0)}]}\right)$.

Proposition 3.7 establishes that the Oracle Risk Ratio G-formula estimator is asymptotically normal. Surprisingly, in the case where there is no effect (i.e. $\tau_{\text{RR}} = 1$), the asymptotic variance is driven by the variance of the Risk Difference on each strata determined by $X$, namely $\text{Var}(\mu_{(1)}(X) - \mu_{(0)}(X))$. By considering the Oracle RR-G instead of RR-G, we remove the additional randomness related to the estimation of the surface responses. It is thus likely that the true variance of RR-G is larger than that of Oracle RR-G, as shown below.

Assuming a linear model for $Y^{(t)}$ and estimating both response surfaces $\hat{\mu}_{(0)}$ and $\hat{\mu}_{(1)}$ using ordinary least squares, the variance of the RR-G can be derived.

**Assumption 3.8** (Linear model). For all $t \in \{0, 1\}$,

$$Y^{(t)} = c_{(t)} + X^\top \beta_{(t)} + \varepsilon_{(t)} \quad \mathbb{E}[X] = \mu$$
$$\mathbb{E}[\varepsilon_{(t)}|X] = 0 \quad \text{Var}[\varepsilon_{(t)}|X] = \sigma^2,$$

where we assume that $Y^{(t)} \geq c > 0$ for some $c$.

**Proposition 3.9** (Asymptotic normality of $\hat{\tau}_{\text{RR,OLS}}$). *Grant Assumptions 3.1 and 3.8. Then, the RR G-formula estimator $\hat{\tau}_{RR,OLS}$ that uses linear regression to estimate $\mu_{(t)}$ satisfies*

$$\sqrt{n}(\hat{\tau}_{RR,OLS} - \tau_{RR}) \xrightarrow{d} \mathcal{N}\left(0, V_{RR\text{-}OLS}\right),$$

*where, letting $\nu_t = \mathbb{E}[X|T = t]$ and $\Sigma_t = \text{Var}(X|T = t)$,*

$$\frac{V_{RR\text{-}OLS}}{\tau^2_{RR}} = \left\|\frac{\beta_{(1)}}{\mathbb{E}\left[Y^{(1)}\right]} - \frac{\beta_{(0)}}{\mathbb{E}\left[Y^{(0)}\right]}\right\|^2_\Sigma + \sigma^2$$

$$\times \left(\frac{1 + (1-e)^2\|\nu_1 - \nu_0\|^2_{\Sigma_1^{-1}}}{e\mathbb{E}\left[Y^{(1)}\right]^2} + \frac{1 + e^2\|\nu_1 - \nu_0\|^2_{\Sigma_0^{-1}}}{(1-e)\mathbb{E}\left[Y^{(0)}\right]^2}\right).$$

The variance of RR-OLS can be decomposed in two terms: the first term corresponds to the oracle variance of RR-G, that is $V_{\text{RR,G}}/\tau^2_{\text{RR}}$; the second term appears due to the estimation of response surfaces via OLS. Contrary to RR-IPW, the variance of the oracle estimator for the G-formula is smaller than that of the OLS estimator. Note that if we use RR-OLS in a Bernoulli Trial, then one can show that even in an RCT setting, adjusting for covariates is beneficial as the variance of the RR-OLS is smaller than the variance of RR-N. These results are provided in Appendix 7.3.4.

### 3.3. Risk Ratio One-step estimator (RR-OS)

A popular estimator for the RD is the augmented inverse probability weighted estimator (AIPW, see Robins et al., 1992). AIPW combines the properties of G-formula and IPW estimators and is *doubly-robust* in the sense that it is consistent as soon as either the propensity or outcome models are correctly specified. By calculating the influence function of the statistical estimand $\psi_{\text{RD}} = \mathbb{E}\left[\mathbb{E}\left[Y|T = 1, X\right] - \mathbb{E}\left[Y|T = 0, X\right]\right]$ we obtain an efficient estimator: it has no asymptotic bias and the minimal asymptotic variance (Kennedy, 2022). Therefore, to estimate the Risk Ratio (RR), a natural approach is to derive an efficient estimator using semi-parametric theory (Tsiatis, 2006), as presented below.

**Definition 3.10** (**Crossfitted RR-OS**). For all $t \in \{0, 1\}$ and all $x$, let $\mu_{(t)}(x) = \mathbb{E}\left[Y^{(0)}|X = x\right]$ and $e_{(t)}(x) = \mathbb{P}\left[T = t|X = x\right]$. We denote $\mathcal{I} = \{1, \ldots, n\}$, let $\mathcal{I}_1, \mathcal{I}_2, \ldots, \mathcal{I}_K$ be a partition of $\mathcal{I}$. Let $\hat{\mu}_{(t)}^{\mathcal{I}_{-k}}(X)$ and $\hat{e}_{(t)}^{\mathcal{I}_{-k}}(X)$ be estimators of $\mu_{(t)}$ and $e_{(t)}$ built on the sample $\mathcal{I}_{-k} = \mathcal{I}\backslash\mathcal{I}_k$. For all $t \in \{0, 1\}$, let

$$\hat{\tau}_{\text{AIPW,t}} = \frac{1}{n}\sum_{k=1}^K \sum_{i \in \mathcal{I}_k}\left(\hat{\mu}_{(t)}^{\mathcal{I}_{-k}}(X_i) + \frac{Y_i - \hat{\mu}_{(t)}^{\mathcal{I}_{-k}}(X_i)}{\hat{e}_{(t)}^{\mathcal{I}_{-k}}(X_i)}\mathbb{1}_{T_i=t}\right) \tag{12}$$

and $\quad \hat{\tau}_{\text{G,t}} = \frac{1}{n}\sum_{k=1}^K \sum_{i \in \mathcal{I}_k}\hat{\mu}_{(t)}^{\mathcal{I}_{-k}}(X_i), \tag{13}$

The crossfitted Risk Ratio One-Step (RR-OS) estimator $\hat{\tau}_{\text{RR-OS}}$ is defined as

$$\hat{\tau}_{\text{RR-OS}} = \frac{\hat{\tau}_{\text{G},1}}{\hat{\tau}_{\text{G},0}} \left( 1 - \frac{\hat{\tau}_{\text{AIPW},0}}{\hat{\tau}_{\text{G},0}} \right) + \frac{\hat{\tau}_{\text{AIPW},1}}{\hat{\tau}_{\text{G},0}}.$$

Considering the statistical estimand $\psi_{\text{RR}} = \frac{\mathbb{E}[\mathbb{E}[Y|T=1,X]]}{\mathbb{E}[\mathbb{E}[Y|T=0,X]]}$, we obtained the estimator RR-OS, which is efficient when nuisance components are estimated via cross-fitting (Chernozhukov et al., 2017) and with non-parametric methods.

**Proposition 3.11** (**Asymptotic normality of $\hat{\tau}_{\text{RR-OS}}$**). *Grant Assumption 2.3 and Assumption 3.1. Assume that for all $1 \leq k \leq K$, and for all $t \in \{0,1\}$,*

$$\mathbb{E}\left[\left(\hat{\mu}_{(t)}^{\mathcal{I}-k}(X) - \mu_{(t)}(X)\right)^2\right]\mathbb{E}\left[\left(\hat{e}^{\mathcal{I}-k}(X) - e(X)\right)^2\right] = o\left(\tfrac{1}{n}\right) \quad (14)$$

$$\mathbb{E}\left[\hat{\mu}_{(0)}^{\mathcal{I}-k}(X)\right] - \mathbb{E}\left[\mu_{(0)}(X)\right] = o\left(n^{-1/4}\right) \quad (15)$$

$$\mathbb{E}\left[(\hat{\mu}_{(0)}^{\mathcal{I}-k}(X) - \mu_{(0)}(X))^2\right]\mathbb{E}\left[(\hat{\mu}_{(1)}^{\mathcal{I}-k}(X) - \mu_{(1)}(X))^2\right] = o\left(\tfrac{1}{n}\right), \quad (16)$$

*with $\eta \leq \hat{e}^{\mathcal{I}-k}(\cdot) \leq 1 - \eta$ (see Assumption 3.1). Then the One-Step estimator is asymptotically unbiased and satisfies*

$$\sqrt{n}\left(\hat{\tau}_{\text{RR-OS}} - \tau_{\text{RR}}\right) \xrightarrow{d} \mathcal{N}\left(0, V_{RR,OS}\right),$$

*where*

$$\frac{V_{RR,OS}}{\tau_{RR}^2} = \text{Var}\left(\frac{\mu_{(1)}(X)}{\mathbb{E}\left[Y^{(1)}\right]} - \frac{\mu_{(0)}(X)}{\mathbb{E}\left[Y^{(0)}\right]}\right)$$
$$+ \mathbb{E}\left[\frac{\text{Var}\left(Y^{(1)}|X\right)}{e(X)\mathbb{E}\left[Y^{(1)}\right]^2}\right] + \mathbb{E}\left[\frac{\text{Var}\left(Y^{(0)}|X\right)}{(1-e(X))\mathbb{E}\left[Y^{(0)}\right]^2}\right].$$

This estimator is efficient: its asymptotic variance is minimal. The semi parametric theory develops efficient estimators by compensating for the first-order bias (Kennedy, 2022), this can be achieved either by estimating and subtracting the first-order bias, leading to the RR-OS estimator or by finding values for the target parameter and nuisance parameters that solve the estimating equation (see A.Schuler, 2024, for the RD case), resulting in RR-AIPW presented below (calculations are detailed in Appendix 7.3.6).

### 3.4. Risk Ratio Augmented Inverse Propensity Weighting (RR-AIPW)

**Definition 3.12** (**Crossfitted RR-AIPW**). The Risk Ratio AIPW crossfitted is defined as

$$\hat{\tau}_{\text{RR,AIPW}} := \frac{\hat{\tau}_{\text{AIPW},1}}{\hat{\tau}_{\text{AIPW},0}},$$

where $\hat{\tau}_{\text{AIPW},0}$, and $\hat{\tau}_{\text{AIPW},1}$ are defined in 3.10.

The RR-AIPW is simply the ratio of two one-step estimators, one for $\mathbb{E}\left[Y^{(1)}\right]$ and one for $\mathbb{E}\left[Y^{(0)}\right]$. This method may seem simplistic at first glance, since approximating both the numerator and denominator usually results in a non-zero asymptotic bias. However, RR-AIPW is derived via the estimating equation method using influence function theory, which results in an efficient (asymptotically unbiased) estimator. Note that in the case of the Risk Difference (RD), both approaches (One-step bias correction and estimating equation) yield the same AIPW estimator. However, because our statistical estimand for the Risk Ratio is nonlinear, the resulting estimators differ. It remains that they are both efficient, as shown below.

**Proposition 3.13** (**RR AIPW asymptotic normality**). *Grant Assumptions 2.3 and 3.1. Assume that (14) holds and that, for all $1 \leq k \leq K$ and all $t \in \{0,1\}$,*

$$\mathbb{E}\left[\left(\hat{\mu}_{(t)}^{\mathcal{I}-k}(X) - \mu_{(t)}(X)\right)^2\right] = o(1), \quad \mathbb{E}\left[\left(\hat{e}^{\mathcal{I}-k}(X) - e(X)\right)^2\right] = o(1), \quad (17)$$

*with $\eta \leq \hat{e}^{\mathcal{I}-k}(\cdot) \leq 1 - \eta$. Then, the crossfitted Risk Ratio AIPW estimator is asymptotically unbiased and satisfies*

$$\sqrt{n}\left(\hat{\tau}_{RR,AIPW} - \tau_{RR}\right) \xrightarrow{d} \mathcal{N}\left(0, V_{RR,OS}\right),$$

*where $V_{RR,OS}$ is defined in Proposition 3.11.*

Assumptions in Proposition 3.13 are the same as those used in the Risk Difference AIPW estimator (Wager, 2020) to achieve double robustness. Specifically, RR-AIPW benefits from *weak* double robustness, meaning consistency is maintained as long as either the outcome model or the propensity score model is estimated consistently. This contrasts with RR-OS, which lacks this flexibility: it requires consistency of both outcome models ($\mu_0, \mu_1$) or the joint consistency of ($e, \mu_0$) to achieve reliable inference. Additionally, condition 14 often referred to as a risk decay condition ensures *strong* double robustness for RR-AIPW. This property ensures asymptotic normality when both the propensity score and outcome regression converge at sufficiently fast rates. We recommend RR-AIPW over RR-OS, not only because both share identical efficiency in their asymptotic distributions, but also because RR-AIPW operates under weaker assumptions. RR-AIPW's strong and weak double robustness properties provide greater resilience to model misspecification compared to RR-OS.

## 4. Simulation

Simulations for RCT are provided in Appendix 8. For observational studies, we generate datasets $(X, T, Y^{(0)}, Y^{(1)})$ according to the general model

$$
\begin{array}{ll}
Y^{(1)} = m(X) + b(X) + \varepsilon_{(1)} & \mathbb{P}\left[T=1|X\right] = e(X), \\
Y^{(0)} = b(X) + \varepsilon_{(0)} & \text{with } \varepsilon_{(t)} \sim \mathcal{N}\left(0, \sigma^2\right).
\end{array}
$$

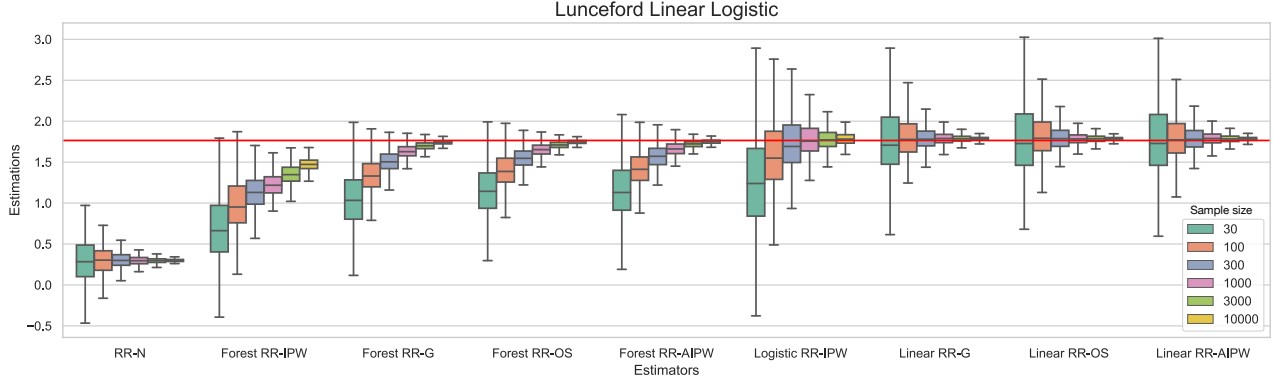

*Figure 1.* Risk Ratio estimators computed for a Linear/Logistic DGP, with 3000 repetitions.

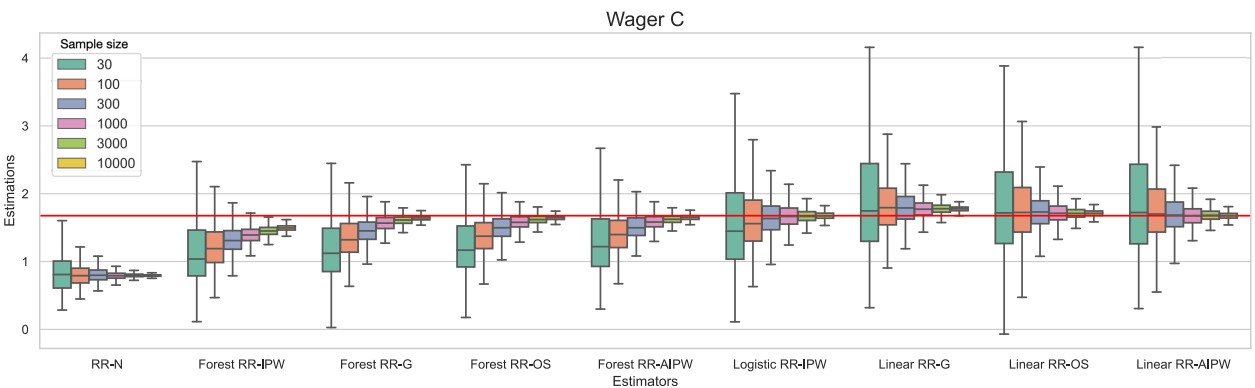

*Figure 2.* Risk Ratio estimators computed for a non-Linear-Logistic DGP, with 3000 repetitions.

where $m(.)$, $b(.)$, and $e(.)$ respectively correspond to the treatment effect, the baseline and the propensity score. The true Risk Ratio can be expressed as $\tau_{\text{RR}} = \mathbb{E}\left[Y^{(1)}\right] / \mathbb{E}\left[Y^{(0)}\right] = \mathbb{E}\left[m(X)\right] / \mathbb{E}\left[b(X)\right] + 1$. We compare the performances of all estimators defined in Section 3 where nuisance components (regression surfaces and propensity score) are estimated via parametric (linear/logistic regression) or non-parametric methods (random forests). More details are provided in Appendix 8.

### 4.1. Linear and Logistic DGP

The first observational data-generating process (DGP), introduced in Lunceford & Davidian (2004a), uses linear outcome models (treatment effect and baseline) and a logistic propensity score, defined as $m(X, V) = 2$ with:

$$b(X, V) = \beta_0^\top [X, V], \quad e(X) = (1 + \exp(-\beta_e^\top X))^{-1}$$
$$\beta_0 = (-1, 1, -1, -1, 1, 1) \quad \beta_e = (-0.6, 0.6, -0.6).$$

Covariates $X = (X_1, X_2, X_3)^\top$ influence treatment and response, while $V = (V_1, V_2, V_3)^\top$ only affect the response. $[X, V]$ are jointly distributed, with $X_3 \sim$ Bernoulli(0.2) and $V_3 \sim$ Bernoulli($P(V_3 = 1 \mid X_3) = 0.75X_3 +$

$0.25(1 - X_3)$). Conditional on $X_3$, $(X_1, V_1, X_2, V_2)^\top \sim \mathcal{N}(\lambda_{X_3}, \Sigma)$, where:

$$\lambda_1 = -\lambda_0 = \begin{bmatrix} 1 \\ 1 \\ -1 \\ -1 \end{bmatrix}, \Sigma = \begin{bmatrix} 1 & 0.5 & -0.5 & -0.5 \\ 0.5 & 1 & -0.5 & -0.5 \\ -0.5 & -0.5 & 1 & 0.5 \\ -0.5 & -0.5 & 0.5 & 1 \end{bmatrix}$$

Results are depicted in Figure 1. Only confounding variables are used as inputs in the different estimators. As expected, since the generative process is linear, methods that use parametric estimators (logistic/linear regression) outperform those using non-parametric approaches (random forests) in finite-sample settings. While all methods (except RR-N) converge to the correct RR, methods based on parametric estimators exhibit a faster rate of convergence and are unbiased (except for Logistic RR-IPW) even for small sample sizes. Indeed, random forests are not suited for linear generative process and require here more than 10000 samples to estimate correctly the RR. All in all, when the outcome modelling and the propensity scores are linear, the two doubly-robust estimators (RR-AIPW and RR-OS) and the RR-G, all based on linear estimators, achieve the

best performances: they are unbiased, even for small sample sizes, and converge quickly to the true RR.

Furthermore, both Linear RR-OS and RR-AIPW estimators give very similar results. More simulations on a Non-Linear and (Non-)Logistic DGPs are available in 8.2. As anticipated, estimators with non-parametric nuisance functions outperform their parametric counterparts when the DGP deviates from linear or logistic structures.

### 4.1.1. NON-LINEAR AND LOGISTIC DGP

We use a semi-parametric setup (see Nie & Wager, 2020) with non-linear baseline models, a constant treatment effect and a logistic propensity score:

$$b(X) = 2 \log \left(1 + e^{X_1 + X_2 + X_3}\right),$$
$$e(X) = 1 / \left(1 + e^{X_2 + X_3}\right) \quad \text{and} \quad m(X) = 1,$$

where $X \sim \mathcal{N}(0, I_{d \times d})$. Results are presented in Figure 2. The Forest IPW and Linear G-formula estimators yield poor RR estimates for the largest sample size. The forest IPW uses random forests to estimate a logistic model, which may still converge, but at a slower rate than other methods. The Linear G-formula employs linear regressions to estimate the response surfaces, potentially leading to an irreducible asymptotic bias. The Forest RR-G, Forest RR-AIPW, and Forest RR-OS estimators converge slowly to the true RR. This simulation highlights the doubly robust properties of the Linear RR-AIPW and Linear RR-OS estimators: they target the true RR even at small sample sizes, as they have at least one well-specified model.

### 4.2. Confidence intervals (CI)

In both the Linear/Logistic and Non-Linear/Logistic data-generating processes (DGPs), we build asymptotic 95% CI for the RR-AIPW, RR-G and RR-N estimators based on their asymptotic normality. Variances were estimated via equation 136, 33, 47, 52 and 30. In Figure 3, we present the distribution of the length and coverage (probability that the CI contains the risk ratio) for each estimator (300 repetitions). IPW was excluded for its poor performance (excessively large CIs), caused by propensity scores close to 0 or 1 (see Figure 8). As expected, RR-N CI has nearly zero coverage due to non-RCT setting. The Forest RR-G and RR-AIPW confidence intervals also exhibit poor coverage, which is in agreement with the Linear/Logistic DGP. In contrast, Linear RR-G and RR-AIPW demonstrate good coverage. Note that the CI for OLS RR-G, built on Proposition 3.9, has a better coverage than the Linear RR-G method, as it includes the uncertainty due to linear estimations. Although only the RR AIPW has coverage above 95%, the OLS RR-G has a shorter average predicted length compared to the Linear RR AIPW.

Turning to the Non-Linear/Logistic DGP, Figure 4 shows the CI length and coverage for the same estimators. In this setting, only the Linear RR-AIPW estimator maintains acceptable coverage—likely because propensity scores are well-estimated, whereas modeling complex, non-linear response surfaces is difficult with either linear models or forest-based methods when $n = 1000$. As before, Forest RR-G and Forest RR-AIPW struggle with coverage, and the RR-N estimator again shows almost no coverage, reflecting its limitations in observational settings. More simulations on a Non-Linear/Non-Logistic DGP are available in Appendix 8.2.

## 5. Real-World Experiment

To better illustrate the practical application and behavior of our estimators, we include a real-world study from Mayer et al. (2020) involving 8,270 patients with traumatic brain injury (TBI), using data extracted from the Traumabase. The Traumabase is a continuously updated database that collects comprehensive clinical data from the scene of the accident through to hospital discharge. The causal effect of tranexamic acid (TXA) on 28-day mortality is estimated by adjusting for 17 confounding variables. These variables include key metrics for severe trauma cases, such as systolic and diastolic blood pressure, heart rate, oxygen saturation, and details of interventions. We subsample the real data to obtain different sample sizes, with results averaged over 3,000 simulations for each sample size. Since this is a real dataset, the true value of $\tau_{RR}$ is unknown. Semi-synthetic simulations, in which the treatment and potential outcomes are generated (allowing us to know the true RR value), are described in detail in 8.3. Results are displayed in Figure 5.

As observed, most estimators yield values larger than one, suggesting a potential deleterious effect of TXA (increased mortality). This trend aligns with findings from previous studies (e.g., (Mayer et al., 2020)). Notably, Forest RR-IPW is the only estimator indicating a beneficial effect of the treatment. RR-N, which is not suited for use with observational data, produces a higher value compared to other estimators. Interestingly, estimators based on linear models (Linear RR-AIPW, Linear RR-G, Linear RR-OS) exhibit the largest variances, particularly for small sample sizes.

## 6. Conclusion

Quantifying treatment effects presents challenges, since different measures may lead to different understanding of the same phenomenon. In our study, we focus on one of these measures, the Risk Ratio and introduced several estimators, valid in RCT or observational studies. Using dedicated mathematical tools (influence function theory, M-estimation), we establish their asymptotic normality, limiting variance and

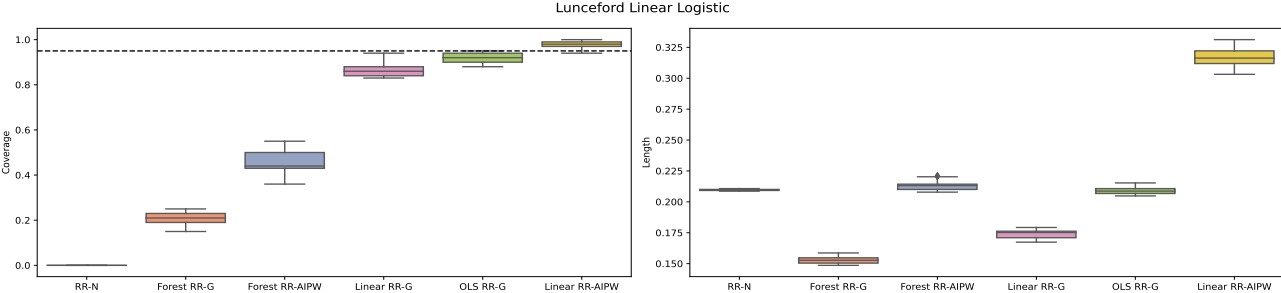

*Figure 3.* Coverage (left) and Length (right) of asymptotic CI derived from Section 2 and Section 3 with $n = 1000$ and 300 repetitions.

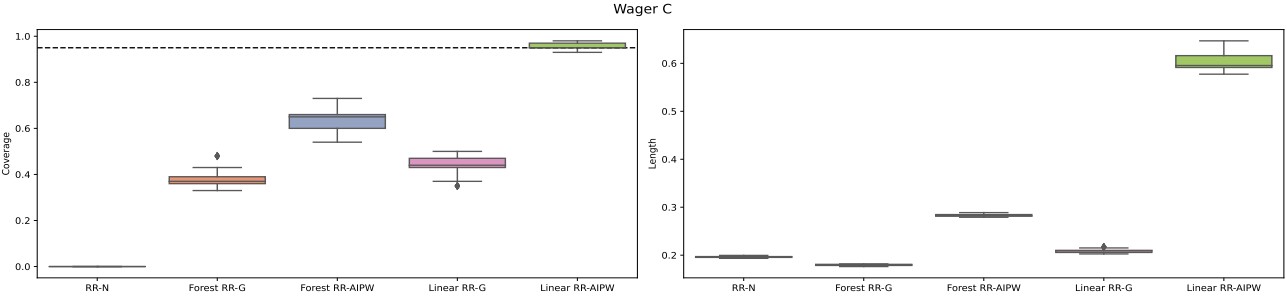

*Figure 4.* Average coverage (left) and average length (right) of asymptotic confidence interval derived from Section 2 and Section 3 for different estimators with $n = 1000$ and 300 repetitions for a Non-Linear and Logistic DGP.

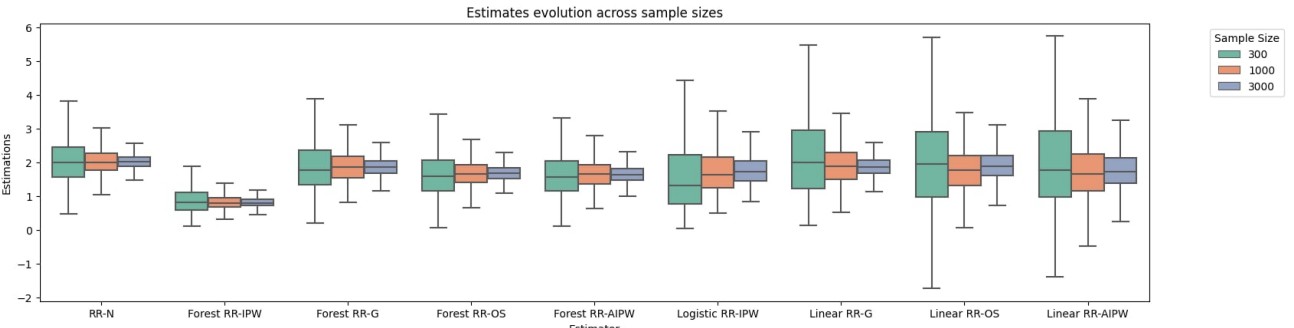

*Figure 5.* RR estimations with weighting, outcome based and augmented estimators as a function of the sample size for the Traumabase. Parametric (Linear) and non parametric (Forest) estimations of nuisance are displayed.

derive asymptotic confidence intervals. Empirical evaluations show that RR-N and RR-IPW have poor performances. Either Linear or Forest RR-AIPW (or RR-OS) show similar (good) behaviors to estimate the Risk Ratio, with the best theoretical guarantees among all studied estimators. Since RR-AIPW requires fewer assumption and is simpler to compute, we would recommend to use RR-AIPW. As for the doubly-robust approaches, G-formula is competitive, with performances that depend on the setting and the estimation method used for the nuisance components. Identifying guidelines establishing when linear nuisance components should be used instead of non-parametric ones still remains an open problem. In practice, observational studies may be used to generalize the treatment effect from a RCT population to the general population of interest. Our work is a first step toward proposing procedures to generalize the Risk Ratio to general populations.

## Acknowledgments

This work has been done in the frame of the PEPR SN SMATCH project and has benefited from a governmental grant managed by the Agence Nationale de la Recherche under the France 2030 programme, reference ANR-22-PESN-0003.

## Impact Statement

This paper proposes and studies estimators for the Risk Ratio in observational studies, addressing a methodological gap in causal inference. The Risk Ratio is widely used in clinical and epidemiological research to assess relative treatment effects. However, it is important to consider the Risk Ratio alongside other effect measures, as each provides a different perspective on the data and may have different implications for practice. As these methods rely on standard causal assumptions such as ignorability—which can be difficult to satisfy in practice—caution is needed when interpreting the results. Violations of these assumptions may lead to biased estimates and incorrect conclusions about treatment effects.

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

# 7. Appendix

### 7.1. Preliminary results

Since we are studying the asymptotic properties of the risk ratio, we cannot directly apply a central limit theorem as in (Wager, 2020). We will therefore rely on Theorem 7.1 to prove most of our asymptotic results.

**Theorem 7.1** (**Asymptotic normality of the ratio of two estimators**). *Let $(Z_1, \ldots, Z_n)$ be $n$ i.i.d. random variables, $g_0$ and $g_1$ two functions square integrable such that $\mathbb{E}\left[g_0(Z_i)\right] = \tau_0$ and $\mathbb{E}\left[g_1(Z_i)\right] = \tau_1$, where $\tau_0 \neq 0$. Then, we have that*

$$\sqrt{n}\left(\frac{\sum_{i=1}^n g_1(Z_i)}{\sum_{i=1}^n g_0(Z_i)} - \frac{\tau_1}{\tau_0}\right) \xrightarrow{d} \mathcal{N}\left(0, V_{RR}^\star\right),$$

*where*

$$V_{RR}^\star = \left(\frac{\tau_1}{\tau_0}\right)^2 \mathrm{Var}\left(\frac{g_1(Z)}{\tau_1} - \frac{g_0(Z)}{\tau_0}\right).$$

*Proof.* We rely on M-estimation theory to prove Theorem 7.1. Let

$$\hat{\boldsymbol{\theta}}_n = \begin{pmatrix} \frac{1}{n}\sum_{i=1}^n g_0(Z_i) \\ \frac{1}{n}\sum_{i=1}^n g_1(Z_i) \\ \frac{\sum_{i=1}^n g_1(Z_i)}{\sum_{i=1}^n g_0(Z_i)} \end{pmatrix} \quad \text{and} \quad \psi(Z, \boldsymbol{\theta}) = \begin{pmatrix} g_0(Z) - \theta_0 \\ g_1(Z) - \theta_1 \\ \theta_1 - \theta_2\theta_0 \end{pmatrix}, \tag{18}$$

where $\boldsymbol{\theta} = (\theta_0, \theta_1, \theta_2)$. We have that

$$\sum_{i=1}^n \left(g_0(Z_i) - \frac{1}{n}\sum_{j=1}^n g_0(Z_j)\right) = \sum_{i=1}^n g_0(Z_i) - \sum_{j=1}^n g_0(Z_j) = 0,$$

and similarly

$$\sum_{i=1}^n \left(g_1(Z_i) - \frac{1}{n}\sum_{j=1}^n g_1(Z_j)\right) = \sum_{i=1}^n g_1(Z_i) - \sum_{j=1}^n g_1(Z_j) = 0.$$

Besides,

$$\sum_{i=1}^n \left(\frac{1}{n}\sum_{j=1}^n g_1(Z_j) - \frac{\sum_{j=1}^n g_1(Z_j)}{\sum_{j=1}^n g_0(Z_j)}\frac{1}{n}\sum_{j=1}^n g_0(Z_j)\right) = 0.$$

Gathering the three previous equalities, we obtain

$$\sum_{i=1}^n \psi(Z_i, \boldsymbol{\theta}_n) = 0, \tag{19}$$

which proves that $\hat{\boldsymbol{\theta}}_n$ is an M-estimator of type $\psi$ (see Stefanski & Boos, 2002). Furthermore, letting $\boldsymbol{\theta}_\infty = (\tau_0, \tau_1, \tau_1/\tau_0)$, simple calculations show that

$$\mathbb{E}\left[\psi(Z, \boldsymbol{\theta}_\infty)\right] = 0. \tag{20}$$

Since the first two components of $\psi$ are linear with respect to $\theta_0$ and $\theta_1$ and since the third component is linear with respect to $\theta_2$, $\boldsymbol{\theta}_\infty$ defined above is the only value satisfying (20). Define

$$A\left(\theta_\infty\right) = \mathbb{E}\left[\frac{\partial \psi}{\partial \theta}\Big|_{\theta=\theta_\infty}\right] \quad \text{and} \quad B(\theta_\infty) = \mathbb{E}\left[\psi(Z, \theta_\infty)\psi(Z, \theta_\infty)^T\right]. \tag{21}$$

We now check the conditions of Theorem 7.2 in Stefanski & Boos (2002). First, let us compute $A\left(\theta_\infty\right)$ and $B\left(\theta_\infty\right)$. Since

$$\frac{\partial \psi}{\partial \theta}(Z, \theta) = \begin{pmatrix} -1 & 0 & 0 \\ 0 & -1 & 0 \\ -\theta_2 & 1 & -\theta_0 \end{pmatrix}, \tag{22}$$

we obtain

$$A\left(\boldsymbol{\theta_\infty}\right) = \begin{pmatrix} -1 & 0 & 0 \\ 0 & -1 & 0 \\ -\frac{\tau_1}{\tau_0} & 1 & -\tau_0 \end{pmatrix}, \tag{23}$$

which leads to

$$A^{-1}\left(\boldsymbol{\theta_\infty}\right) = \begin{pmatrix} -1 & 0 & 0 \\ 0 & -1 & 0 \\ \frac{\tau_1}{\tau_0^2} & -\frac{1}{\tau_0} & -\frac{1}{\tau_0} \end{pmatrix}. \tag{24}$$

Regarding $B\left(\theta_\infty\right)$, elementary calculations show that

$$\psi(Z, \theta_\infty)\psi(Z, \theta_\infty)^T = \begin{pmatrix} (g_0(Z) - \tau_0)^2 & (g_0(Z) - \tau_0)\left(g_1(Z) - \tau_1\right) & 0 \\ (g_0(Z) - \tau_0)\left(g_1(Z) - \tau_1\right) & \left(g_1(Z) - \tau_1\right)^2 & 0 \\ 0 & 0 & 0 \end{pmatrix},$$

which leads to

$$B(\theta_\infty) = \begin{pmatrix} \mathrm{Var}\left[g_0(Z)\right] & \mathrm{Cov}\left(g_0(Z), g_1(Z)\right) & 0 \\ \mathrm{Cov}\left(g_0(Z), g_1(Z)\right) & \mathrm{Var}\left[g_1(Z)\right] & 0 \\ 0 & 0 & 0 \end{pmatrix}.$$

Based on the previous calculations, we have

- $\psi(z, \boldsymbol{\theta})$ and its first two partial derivatives with respect to $\boldsymbol{\theta}$ exist for all $z$ and for all $\boldsymbol{\theta}$ in the neighborhood of $\boldsymbol{\theta_\infty}$.

- For each $\boldsymbol{\theta}$ in the neighborhood of $\boldsymbol{\theta_\infty}$, we have for all $i, j, k \in \{0, 2\}$:

$$\left|\frac{\partial^2}{\partial\theta_i\partial\theta_j}\psi_k(z, \boldsymbol{\theta})\right| \leq 1$$

  and 1 is integrable.

- $A(\theta_\infty)$ exists and is nonsingular.

- $B(\theta_\infty)$ exists and is finite.

Since we have:

$$\sum_{i=1}^n \psi(T_i, Y_i, \hat{\boldsymbol{\theta}}_n) = 0 \quad \text{and} \quad \hat{\boldsymbol{\theta}}_n \xrightarrow{p} \theta_\infty.$$

Then the conditions of Theorem 7.2 in Stefanski & Boos (2002) are satisfied, we have

$$\sqrt{n}\left(\hat{\boldsymbol{\theta}}_n - \theta_\infty\right) \xrightarrow{d} \mathcal{N}\left(0, A(\theta_\infty)^{-1}B(\theta_\infty)(A(\theta_\infty)^{-1})^\top\right),$$

where

$$A(\theta_\infty)^{-1}B(\theta_\infty)(A(\theta_\infty)^{-1})^\top$$

$$= \begin{bmatrix} \mathrm{Var}\left[g_0(Z)\right] & \mathrm{Cov}\left(g_0(Z), g_1(Z)\right) & \frac{\mathrm{Cov}(g_0(Z),g_1(Z))}{\tau_0} - \frac{\tau_1\,\mathrm{Var}[g_0(Z)]}{\tau_0^2} \\ \mathrm{Cov}\left(g_0(Z), g_1(Z)\right) & \mathrm{Var}\left[g_1(Z)\right] & -\frac{\mathrm{Cov}(g_0(Z),g_1(Z))\tau_1}{\tau_0^2} + \frac{\mathrm{Var}[g_1(Z)]}{\tau_0} \\ \frac{\mathrm{Cov}(g_0(Z),g_1(Z))}{\tau_0} - \frac{\tau_1\,\mathrm{Var}[g_0(Z)]}{\tau_0^2} & -\frac{\mathrm{Cov}(g_0(Z),g_1(Z))\tau_1}{\tau_0^2} + \frac{\mathrm{Var}[g_1(Z)]}{\tau_0} & V_{\mathrm{RR}}^\star \end{bmatrix}, \tag{25}$$

with

$$V_{\text{RR}}^{\star} = \left(\frac{\tau_1}{\tau_0}\right)^2 \text{Var}\left(\frac{g_1(Z)}{\tau_1} - \frac{g_0(Z)}{\tau_0}\right). \tag{26}$$

In particular,

$$\sqrt{n}\left(\frac{\sum_{i=1}^n g_1(Z_i)}{\sum_{i=1}^n g_0(Z_i)} - \frac{\tau_1}{\tau_0}\right) \xrightarrow{d} \mathcal{N}\left(0, V_{\text{RR}}^{\star}\right). \tag{27}$$

$\square$

**Assumption 7.2.** We suppose that both $Y^{(0)}$ and $Y^{(1)}$ are squared integrable and that $\mathbb{E}\left[Y^{(0)}|X\right], \mathbb{E}\left[Y^{(1)}|X\right] > 0$.

**Theorem 7.3** (**Finite sample bias and variance of the ratio of two estimators**). *Let $T_1(\boldsymbol{Z})$ and $T_0(\boldsymbol{Z})$ be two unbiased estimators of $\tau_1$ and $\tau_0 > 0$ where $\boldsymbol{Z} = (Z_1, \ldots, Z_n)$ be $n$ i.i.d. random variables. We assume that $M_0 \geq T_0(\boldsymbol{Z}) \geq m_0 > 0$, $|T_1(\boldsymbol{Z})| \leq M_1$. We also assume that $\text{Var}(T_1(\boldsymbol{Z})) = O_p\left(\frac{1}{n}\right)$ and $\text{Var}(T_0(\boldsymbol{Z})) = O_p\left(\frac{1}{n}\right)$ Then, we have that*

$$\text{Bias}\left(\frac{T_1(\boldsymbol{Z})}{T_0(\boldsymbol{Z})}, \frac{\tau_1}{\tau_0}\right) = \left|\mathbb{E}\left[\frac{T_1(\boldsymbol{Z})}{T_0(\boldsymbol{Z})}\right] - \frac{\tau_1}{\tau_0}\right| \leq \frac{M_1 M_0}{n m_0^2}\left(\frac{M_0}{m_0} + 1\right),$$

*and*

$$\left|\text{Var}\left(\frac{T_1(\boldsymbol{Z})}{T_0(\boldsymbol{Z})}\right) - \left(\frac{\tau_1}{\tau_0}\right)^2 \text{Var}\left(\frac{T_1(\boldsymbol{Z})}{\tau_1} - \frac{T_0(\boldsymbol{Z})}{\tau_0}\right)\right| \leq \frac{2M_0 M_1}{n m_0^4}\left(\frac{M_0 M_1}{m_0^2} + 1\right)$$

.

*Proof.* We rely on the multivariate version of Taylor's theorem to prove Theorem 7.3. We first introduce the multi-index notation:

$$|\alpha| = \alpha_1 + \cdots + \alpha_n, \quad \alpha! = \alpha_1! \cdots \alpha_n!, \quad \boldsymbol{x}^\alpha = x_1^{\alpha_1} \cdots x_n^{\alpha_n}$$

and

$$D^\alpha f = \frac{\partial^{|\alpha|} f}{\partial x_1^{\alpha_1} \cdots \partial x_n^{\alpha_n}}, \qquad |\alpha| \leq k$$

Let $f$ be the ratio function

$$f: \mathbb{R}_+^* \times \mathbb{R}_+^* \longrightarrow \mathbb{R}$$
$$(x_1, x_2) \longmapsto x_1/x_2.$$

Since $f$ is two times continuously differentiable then one can derive an exact formula for the remainder in terms of 2nd order partial derivatives of $f$. Namely, if we define $\boldsymbol{x} = (x_1, x_2)$ and for $\boldsymbol{a} \in \mathbb{R}_+^* \times \mathbb{R}_+^*$

$$f(\boldsymbol{x}) = \sum_{|\alpha| \leq 1} \frac{D^\alpha f(\boldsymbol{a})}{\alpha!}(\boldsymbol{x} - \boldsymbol{a})^\alpha + R_{k+1}(\boldsymbol{x}), \tag{28}$$

with

$$R_{k+1}(\boldsymbol{x}) = \sum_{|\beta|=k+1} (\boldsymbol{x} - \boldsymbol{a})^\beta \frac{|\beta|}{\beta!} \int_0^1 (1-t)^{|\beta|-1} D^\beta f(\boldsymbol{a} + t(\boldsymbol{x} - \boldsymbol{a})) \, dt.$$

**Bias:**

Computing 28 for the ratio function with $\boldsymbol{x} = (T_1(\boldsymbol{Z}), T_0(\boldsymbol{Z}))$, $\boldsymbol{a} = (\tau_1, \tau_0)$ and taking the expectation gives us:

$$
\mathbb{E}\left[f(T_1(\boldsymbol{Z}), T_0(\boldsymbol{Z}))\right]
$$

$$
= \mathbb{E}\left[f(\tau_1, \tau_0) + \frac{\partial f(\tau_1, \tau_0)}{\partial T_1(\boldsymbol{Z})}(T_1(\boldsymbol{Z}) - \tau_1) + \frac{\partial f(\tau_1, \tau_0)}{\partial T_0(\boldsymbol{Z})}(T_0(\boldsymbol{Z}) - \tau_0) + R_2(T_1(\boldsymbol{Z}), T_0(\boldsymbol{Z}))\right]
$$

$$
= \mathbb{E}\left[f(\tau_1, \tau_0)\right] + \frac{\partial f(\tau_1, \tau_0)}{\partial T_1(\boldsymbol{Z})}\mathbb{E}\left[(T_1(\boldsymbol{Z}) - \tau_1)\right]
$$

$$
+ \frac{\partial f(\tau_1, \tau_0)}{\partial T_0(\boldsymbol{Z})}\mathbb{E}\left[(T_0(\boldsymbol{Z}) - \tau_0)\right] + \mathbb{E}\left[R_2(T_1(\boldsymbol{Z}), T_0(\boldsymbol{Z}))\right]
$$

$$
= \frac{\tau_1}{\tau_0} + \mathbb{E}\left[R_2(T_1(\boldsymbol{Z}), T_0(\boldsymbol{Z}))\right]
$$

In order to produce Theorem 7.3, we just need to show that $\mathbb{E}\left[R_2(T_1(\boldsymbol{Z}), T_0(\boldsymbol{Z}))\right] = O_p\left(\frac{1}{n}\right)$. To do so, we first compute $R_2(T_1(\boldsymbol{Z}), T_0(\boldsymbol{Z}))$

$$
R_2(T_1(\boldsymbol{Z}), T_0(\boldsymbol{Z})) = \underbrace{2(T_0(\boldsymbol{Z}) - \tau_0)^2 \int_0^1 \frac{(1-t)(\tau_1 + t(T_1(\boldsymbol{Z}) - \tau_1))}{(\tau_0 + t(T_0(\boldsymbol{Z}) - \tau_0))^3}\, dt}_{R_2^1(T_1(\boldsymbol{Z}), T_0(\boldsymbol{Z}))}
$$

$$
- \underbrace{2(T_0(\boldsymbol{Z}) - \tau_0)(T_1(\boldsymbol{Z}) - \tau_1) \int_0^1 \frac{1-t}{(\tau_0 + t(T_0(\boldsymbol{Z}) - \tau_0))^2}\, dt}_{R_2^2(T_1(\boldsymbol{Z}), T_0(\boldsymbol{Z}))}
$$

Since we assume that $T_0(\boldsymbol{Z}) \geq m_0 > 0$ and that $|T_1(\boldsymbol{Z})| \leq M_1$ we have:

$$
\left|R_2^1(T_1(\boldsymbol{Z}), T_0(\boldsymbol{Z}))\right| = \left|2(T_0(\boldsymbol{Z}) - \tau_0)^2 \int_0^1 \frac{(1-t)(\tau_1 + t(T_1(\boldsymbol{Z}) - \tau_1))}{(\tau_0 + t(T_0(\boldsymbol{Z}) - \tau_0))^3}\, dt\right|
$$

$$
\leq 2(T_0(\boldsymbol{Z}) - \tau_0)^2 \int_0^1 \left|\frac{(1-t)\max(\tau_1, M_1)}{\min(m_0, \tau_0)^3}\right| dt
$$

$$
= (T_0(\boldsymbol{Z}) - \tau_0)^2 \underbrace{\frac{M_1}{m_0^3}}_{C_1}
$$

Similarly, we have:

$$
\left|R_2^2(T_1(\boldsymbol{Z}), T_0(\boldsymbol{Z}))\right| = \left|2(T_0(\boldsymbol{Z}) - \tau_0)(T_1(\boldsymbol{Z}) - \tau_1) \int_0^1 \frac{1-t}{(\tau_0 + t(T_0(\boldsymbol{Z}) - \tau_0))^2}\, dt\right|
$$

$$
= 2\left|(T_0(\boldsymbol{Z}) - \tau_0)(T_1(\boldsymbol{Z}) - \tau_1)\right| \left|\int_0^1 \frac{1-t}{(\tau_0 + t(T_0(\boldsymbol{Z}) - \tau_0))^2}\, dt\right|
$$

$$
\leq \left|(T_0(\boldsymbol{Z}) - \tau_0)(T_1(\boldsymbol{Z}) - \tau_1)\right| \underbrace{\frac{1}{m_0^2}}_{C_2}
$$

Finally we get that:

$$
\left|\mathbb{E}\left[R_2(T_1(\boldsymbol{Z}), T_0(\boldsymbol{Z}))\right]\right| \leq \mathbb{E}\left[\left|R_2^1(T_1(\boldsymbol{Z}), T_0(\boldsymbol{Z}))\right| + \left|R_2^2(T_1(\boldsymbol{Z}), T_0(\boldsymbol{Z}))\right|\right]
$$

$$
\leq C_1 \operatorname{Var}(T_0(\boldsymbol{Z})) + C_2 \mathbb{E}\left[\left|(T_0(\boldsymbol{Z}) - \tau_0)(T_1(\boldsymbol{Z}) - \tau_1)\right|\right]
$$

$$
\leq C_1 \operatorname{Var}(T_0(\boldsymbol{Z})) + C_2 \sqrt{\operatorname{Var}(T_0(\boldsymbol{Z})) \operatorname{Var}(T_1(\boldsymbol{Z}))}
$$

$$
\leq C_1 M_0^2 + C_2 M_0 M_1
$$

Since we have that $\mathrm{Var}(T_1(\boldsymbol{Z})) = O_p\left(\frac{1}{n}\right)$ and $\mathrm{Var}(T_0(\boldsymbol{Z})) = O_p\left(\frac{1}{n}\right)$, we can conclude by:

$$\mathrm{Bias}\left(\frac{T_1(\boldsymbol{Z})}{T_0(\boldsymbol{Z})}, \frac{\tau_1}{\tau_0}\right) = \left|\mathbb{E}\left[\frac{T_1(\boldsymbol{Z})}{T_0(\boldsymbol{Z})}\right] - \frac{\tau_1}{\tau_0}\right| \lesssim \frac{M_1 M_0}{n m_0^2}\left(\frac{M_0}{m_0} + 1\right)$$

**Variance:**

Let us begin by expanding the variance of the function $f$:

$$\mathrm{Var}\, f(T_1(\boldsymbol{Z}), T_0(\boldsymbol{Z})) = \mathbb{E}\left[(f(T_1(\boldsymbol{Z}), T_0(\boldsymbol{Z})) - \mathbb{E}\left[f(T_1(\boldsymbol{Z}), T_0(\boldsymbol{Z}))\right])^2\right]$$

Next, apply Taylor's expansion around the means $\tau_1$ and $\tau_0$:

$$= \mathbb{E}\left[(f(\tau_1, \tau_0) + \frac{\partial f(\tau_1, \tau_0)}{\partial T_1(\boldsymbol{Z})}(T_1(\boldsymbol{Z}) - \tau_1) + \frac{\partial f(\tau_1, \tau_0)}{\partial T_0(\boldsymbol{Z})}(T_0(\boldsymbol{Z}) - \tau_0)\right.$$
$$\left. + R_2(T_1(\boldsymbol{Z}), T_0(\boldsymbol{Z})) - \mathbb{E}\left[f(T_1(\boldsymbol{Z}), T_0(\boldsymbol{Z}))\right])^2\right]$$

Simplify by focusing on the first-order derivatives and residual terms:

$$= \mathbb{E}\left[(\frac{\partial f(\tau_1, \tau_0)}{\partial T_1(\boldsymbol{Z})}(T_1(\boldsymbol{Z}) - \tau_1) + \frac{\partial f(\tau_1, \tau_0)}{\partial T_0(\boldsymbol{Z})}(T_0(\boldsymbol{Z}) - \tau_0)\right.$$
$$\left. + R_2(T_1(\boldsymbol{Z}), T_0(\boldsymbol{Z})) - \mathbb{E}\left[R_2(T_1(\boldsymbol{Z}), T_0(\boldsymbol{Z}))\right])^2\right]$$

Decompose the variance into linear, cross-term, and residual contributions:

$$= \mathbb{E}\left[(\frac{\partial f(\tau_1, \tau_0)}{\partial T_1(\boldsymbol{Z})}(T_1(\boldsymbol{Z}) - \tau_1) + \frac{\partial f(\tau_1, \tau_0)}{\partial T_0(\boldsymbol{Z})}(T_0(\boldsymbol{Z}) - \tau_0))^2\right]$$
$$+ 2\mathbb{E}\left[\frac{\partial f(\tau_1, \tau_0)}{\partial T_1(\boldsymbol{Z})}(T_1(\boldsymbol{Z}) - \tau_1)\right.$$
$$\left. + \frac{\partial f(\tau_1, \tau_0)}{\partial T_0(\boldsymbol{Z})}(T_0(\boldsymbol{Z}) - \tau_0)\right]\mathbb{E}\left[R_2(T_1(\boldsymbol{Z}), T_0(\boldsymbol{Z})) - \mathbb{E}\left[R_2(T_1(\boldsymbol{Z}), T_0(\boldsymbol{Z}))\right]\right]$$
$$+ \mathrm{Var}(R_2(T_1(\boldsymbol{Z}), T_0(\boldsymbol{Z})))$$

Finally, re-express the result using a simplified ratio of variances:

$$= \left(\frac{\tau_1}{\tau_0}\right)^2 \mathrm{Var}\left(\frac{T_1(\boldsymbol{Z})}{\tau_1} - \frac{T_0(\boldsymbol{Z})}{\tau_0}\right) + \mathrm{Var}(R_2(T_1(\boldsymbol{Z}), T_0(\boldsymbol{Z})))$$

We now focus on $\mathrm{Var}(R_2(T_1(\boldsymbol{Z}), T_0(\boldsymbol{Z})))$:

$$\mathrm{Var}(R_2(T_1(\boldsymbol{Z}), T_0(\boldsymbol{Z}))) \leq \mathbb{E}\left[(R_2(T_1(\boldsymbol{Z}), T_0(\boldsymbol{Z})))^2\right]$$
$$\leq 2\mathbb{E}\left[|R_2^1(T_1(\boldsymbol{Z}), T_0(\boldsymbol{Z}))|^2\right] + 2\mathbb{E}\left[|R_2^2(T_1(\boldsymbol{Z}), T_0(\boldsymbol{Z}))|^2\right]$$

We first focus on the first term:

$$\mathbb{E}\left[|R_2^1(T_1(\boldsymbol{Z}), T_0(\boldsymbol{Z}))|^2\right] \leq \mathbb{E}\left[\left(C_{02}(T_0(\boldsymbol{Z}) - \tau_0)^2\right)^2\right]$$
$$\leq C_1^2 \mathbb{E}\left[(T_0(\boldsymbol{Z}) - \tau_0)^4\right]$$
$$\leq C_1^2 M_0^2 \mathbb{E}\left[(T_0(\boldsymbol{Z}) - \tau_0)^2\right] \quad T_0(\boldsymbol{Z}) \leq M_0$$
$$\leq C_1^2 M_0^2 \mathrm{Var}(T_0(\boldsymbol{Z}))$$

For the second term we have:

$$\mathbb{E}\left[|R_2^2(T_1(\boldsymbol{Z}), T_0(\boldsymbol{Z}))|^2\right] = C_2^2 \mathbb{E}\left[(T_0(\boldsymbol{Z}) - \tau_0)^2(T_1(\boldsymbol{Z}) - \tau_1)^2\right]$$
$$\leq C_2^2 \sqrt{\mathbb{E}\left[(T_0(\boldsymbol{Z}) - \tau_0)^4\right]\mathbb{E}\left[(T_1(\boldsymbol{Z}) - \tau_1)^4\right]}$$
$$T_1(\boldsymbol{Z}), T_0(\boldsymbol{Z})\ \text{bounded} \quad \leq C_2^2 M_0 M_1 \sqrt{\mathbb{E}\left[(T_0(\boldsymbol{Z}) - \tau_0)^2\right]\mathbb{E}\left[(T_1(\boldsymbol{Z}) - \tau_1)^2\right]}$$
$$\leq C_2^2 M_0 M_1 \sqrt{\mathrm{Var}(T_0(\boldsymbol{Z}))\mathrm{Var}(T_1(\boldsymbol{Z}))}$$

Hence we get that:

$$\text{Var}(R_2(T_1(\boldsymbol{Z}), T_0(\boldsymbol{Z}))) \lesssim \frac{2M_0 M_1}{nm_0^4}\left(\frac{M_0 M_1}{m_0^2} + 1\right)$$

$\square$

## 7.2. Proofs of Section 2

### 7.2.1. RISK RATIO NEYMAN ESTIMATOR

*Proof of Proposition 2.4.*
**Asymptotic Bias and Variance:** we proceed with M-estimations to prove asymptotic bias and variance of the Ratio Neyman estimator, we first define the following:

$$\hat{\boldsymbol{\theta}}_n = \begin{pmatrix} \frac{1}{n_0}\sum_{T_i=0} Y_i \\ \frac{1}{n_1}\sum_{T_i=1} Y_i \\ \hat{\tau}_{R-N,n} \end{pmatrix} \quad \text{and} \quad \psi(T, Y, \boldsymbol{\theta}) = \begin{pmatrix} \psi_0(\boldsymbol{\theta}) \\ \psi_1(\boldsymbol{\theta}) \\ \psi_2(\boldsymbol{\theta}) \end{pmatrix} =: \begin{pmatrix} (1-T)(Y-\theta_0) \\ T(Y-\theta_1) \\ \theta_1 - \theta_2\theta_0 \end{pmatrix}, \tag{29}$$

where $\boldsymbol{\theta} = (\theta_0, \theta_1, \theta_2)$.

Next, we verify that for $\hat{\boldsymbol{\theta}}_n = (\frac{1}{n_0}\sum_{T_i=0} Y_i, \frac{1}{n_1}\sum_{T_i=1} Y_i, \hat{\tau}_{R-N,n})$, we have:

$$\sum_{i=1}^{n} \psi(T_i, Y_i, \hat{\boldsymbol{\theta}}_n) = 0.$$

We begin by demonstrating this for $\psi_1$:

$$\begin{aligned}
\sum_{i=1}^{n} \psi_1(T_i, Y_i, \hat{\boldsymbol{\theta}}_n) &= \sum_{i=1}^{n} T_i\left(Y_i - \frac{1}{n_1}\sum_{T_j=1} Y_j\right) \\
&= \sum_{i=1}^{n} T_i\left(Y_i - \frac{1}{n_1}\sum_{j=1}^{n} T_j Y_j\right) \\
&= \sum_{i=1}^{n} T_i Y_i - \frac{1}{n_1}\sum_{i=1}^{n} T_i \sum_{j=1}^{n} T_j Y_j \\
&= \sum_{i=1}^{n} T_i Y_i - \sum_{j=1}^{n} T_j Y_j \\
&= 0.
\end{aligned}$$

Similarly, we can show:

$$\sum_{i=1}^{n} \psi_0(T_i, Y_i, \hat{\boldsymbol{\theta}}_n) = 0.$$

Moreover, by construction:

$$\sum_{i=1}^{n} \psi_2(T_i, Y_i, \hat{\boldsymbol{\theta}}_n) = 0.$$

Thus, we have established that $\hat{\boldsymbol{\theta}}_n$ is an M-estimator of type $\psi$ (see Stefanski & Boos, 2002). Given that we are in a Bernoulli

Trial, we now demonstrate that $\mathbb{E}\left[\psi(T, Y, \theta_\infty)\right] = 0$ where $\theta_\infty = (\mathbb{E}[Y^{(0)}], \mathbb{E}[Y^{(1)}], \tau_{RR})$. Therefore, we have:

$$
\begin{aligned}
\mathbb{E}\left[\psi_1(\theta_\infty)\right] &= \mathbb{E}\left[T\left(Y - \mathbb{E}[Y^{(1)}]\right)\right] \\
&= \mathbb{E}\left[T\left(Y^{(1)} - \mathbb{E}[Y^{(1)}]\right)\right] && \text{(by SUTVA)} \\
&= \mathbb{E}\left[T\right]\mathbb{E}\left[Y^{(1)} - \mathbb{E}[Y^{(1)}]\right] && \text{(by ignorability)} \\
&= 0.
\end{aligned}
$$

Similarly, we can show:

$$
\mathbb{E}\left[\psi_0(\theta_\infty)\right] = 0.
$$

Furthermore, we have:

$$
\mathbb{E}\left[\psi_2(\theta_\infty)\right] = \mathbb{E}[Y^{(1)}] - \tau_{RR}\mathbb{E}[Y^{(0)}] = 0.
$$

At this point, we note that $\theta_\infty$ is the only value of $\boldsymbol{\theta}$ such that $\mathbb{E}\left[\psi(T, Y, \boldsymbol{\theta})\right] = 0$. We proceed by defining:

$$
A\left(\theta_\infty\right) = \mathbb{E}\left[\frac{\partial\psi}{\partial\theta}\Big|_{\theta=\theta_\infty}\right] \quad \text{and} \quad B(\theta_\infty) = \mathbb{E}\left[\psi(Z, \theta_\infty)\psi(Z, \theta_\infty)^T\right].
$$

Next, we check the conditions of Theorem 7.2 in Stefanski & Boos (2002). First, we compute $A\left(\theta_\infty\right)$ and $B\left(\theta_\infty\right)$. Since:

$$
\frac{\partial\psi}{\partial\theta}(Z, \theta) = \begin{pmatrix} -(1-T) & 0 & 0 \\ 0 & -T & 0 \\ -\theta_2 & 1 & -\theta_0 \end{pmatrix},
$$

we obtain:

$$
A\left(\boldsymbol{\theta_\infty}\right) = \begin{pmatrix} -(1-e) & 0 & 0 \\ 0 & -e & 0 \\ -\tau_{RR} & 1 & -\mathbb{E}[Y^{(0)}] \end{pmatrix},
$$

which leads to:

$$
A^{-1}\left(\boldsymbol{\theta_\infty}\right) = \begin{pmatrix} \frac{1}{e-1} & 0 & 0 \\ 0 & -\frac{1}{e} & 0 \\ \tau_{RR}\frac{1}{\mathbb{E}[Y^{(0)}](1-e)} & -\frac{1}{e\mathbb{E}[Y^{(0)}]} & -\frac{1}{\mathbb{E}[Y^{(0)}]} \end{pmatrix}.
$$

Regarding $B\left(\theta_\infty\right)$, elementary calculations show that:

$$
\begin{aligned}
&\psi(Z, \theta_\infty)\psi(Z, \theta_\infty)^T \\
&= \begin{pmatrix} \left((1-T)(Y - \mathbb{E}[Y^{(0)}])\right)^2 & (1-T)(Y - \mathbb{E}[Y^{(0)}])T(Y - \mathbb{E}[Y^{(1)}]) & 0 \\ (1-T)(Y - \mathbb{E}[Y^{(0)}])T(Y - \mathbb{E}[Y^{(1)}]) & \left(T(Y - \mathbb{E}[Y^{(1)}])\right)^2 & 0 \\ 0 & 0 & 0 \end{pmatrix},
\end{aligned}
$$

which leads to:

$$
B(\theta_\infty) = \begin{pmatrix} (1-e)\operatorname{Var}\left[Y^{(0)}\right] & 0 & 0 \\ 0 & e\operatorname{Var}\left[Y^{(1)}\right] & 0 \\ 0 & 0 & 0 \end{pmatrix}.
$$

Based on the previous calculations, we have:

- $\psi(z, \boldsymbol{\theta})$ and its first two partial derivatives with respect to $\boldsymbol{\theta}$ exist for all $z$ and for all $\boldsymbol{\theta}$ in the neighborhood of $\boldsymbol{\theta_\infty}$.

- For each $\boldsymbol{\theta}$ in the neighborhood of $\boldsymbol{\theta}_\infty$, we have for all $i, j, k \in \{0, 2\}$:

$$\left| \frac{\partial^2}{\partial \theta_i \partial \theta_j} \psi_k(z, \boldsymbol{\theta}) \right| \leq 1$$

and 1 is integrable.

- $A(\theta_\infty)$ exists and is nonsingular.

- $B(\theta_\infty)$ exists and is finite.

Since we have:

$$\sum_{i=1}^n \psi(T_i, Y_i, \hat{\boldsymbol{\theta}}_n) = 0 \quad \text{and} \quad \hat{\boldsymbol{\theta}}_n \xrightarrow{p} \theta_\infty.$$

Then, the conditions of Theorem 7.2 in Stefanski & Boos (2002) are satisfied, we have:

$$\sqrt{n} \left( \hat{\boldsymbol{\theta}}_n - \theta_\infty \right) \xrightarrow{d} \mathcal{N} \left( 0, A(\theta_\infty)^{-1} B(\theta_\infty) (A(\theta_\infty)^{-1})^\top \right),$$

where:

$$A(\theta_\infty)^{-1} B(\theta_\infty) (A(\theta_\infty)^{-1})^\top = \begin{bmatrix} \frac{\mathrm{Var}\left[Y^{(0)}\right]}{(1-e)} & 0 & -\frac{\tau \, \mathrm{Var}\left[Y^{(0)}\right]}{\tau_0(1-e)} \\ 0 & \frac{\mathrm{Var}\left[Y^{(1)}\right]}{e} & \frac{\mathrm{Var}\left[Y^{(1)}\right]}{e\tau_0} \\ -\frac{\tau \, \mathrm{Var}\left[Y^{(0)}\right]}{\tau_0(1-e)} & \frac{\mathrm{Var}\left[Y^{(1)}\right]}{e\tau_0} & V_{R-N} \end{bmatrix},$$

with:

$$V_{R-N} = \tau_{RR}^2 \left( \frac{\mathrm{Var}\left(Y^{(1)}\right)}{e\mathbb{E}[Y^{(1)}]^2} + \frac{\mathrm{Var}\left(Y^{(0)}\right)}{(1-e)\mathbb{E}[Y^{(0)}]^2} \right).$$

In particular, we obtain:

$$\sqrt{n} \left( \hat{\tau}_{\mathrm{RR,N,n}} - \tau_{\mathrm{RR}} \right) \xrightarrow{d} \mathcal{N} \left( 0, V_{\mathrm{RR,N}} \right).$$

Finally, note that:

$$\begin{aligned} V_{R-N} &= \tau_{RR}^2 \left( \frac{\mathrm{Var}\left(Y^{(1)}\right)}{e\mathbb{E}[Y^{(1)}]^2} + \frac{\mathrm{Var}\left(Y^{(0)}\right)}{(1-e)\mathbb{E}[Y^{(0)}]^2} \right) \\ &= \tau_{RR}^2 \left( \frac{\mathbb{E}[(Y^{(1)})^2] - \mathbb{E}[Y^{(1)}]^2}{e\mathbb{E}[Y^{(1)}]^2} + \frac{\mathbb{E}[(Y^{(0)})^2] - \mathbb{E}[Y^{(0)}]^2}{(1-e)\mathbb{E}[Y^{(0)}]^2} \right) \\ &= V_{R-HT} - \frac{\tau_{RR}^2}{e(1-e)}. \end{aligned}$$

As a consequence an estimator $\hat{V}_{R-N}$ can be derived :

$$\hat{V}_{R-N} = \hat{\tau}_{\mathrm{RR,N,n}}^2 \left( \frac{\frac{1}{n} \sum_{T_i=1} \left( Y_i - \frac{1}{n} \sum_{T_i=1} Y_i \right)^2}{\hat{e} \left( \frac{1}{n} \sum_{T_i=1} Y_i \right)^2} + \frac{\frac{1}{n} \sum_{T_i=0} \left( Y_i - \frac{1}{n} \sum_{T_i=0} Y_i \right)^2}{(1-\hat{e}) \left( \frac{1}{n} \sum_{T_i=0} Y_i \right)^2} \right) \tag{30}$$

**Optimal choice of $e$:** the optimal value of $e_{opt}$ is the one that minimizes the variance of the Ratio Neyman estimator. Therefore, we need to solve:

$$\inf_{e \in (0,1)} \tau_{\mathrm{RR}}^2 \left( \frac{\mathrm{Var}\left(Y^{(1)}\right)}{e\mathbb{E}\left[Y^{(1)}\right]^2} + \frac{\mathrm{Var}\left(Y^{(0)}\right)}{(1-e)\mathbb{E}\left[Y^{(0)}\right]^2} \right)$$

Noting that the variance we want to minimize is convex in $e$, we can derive the variance and set it to $0$ to find $e_{opt}$. We have:

$$\frac{C_1}{e_{opt}^2} = \frac{C_0}{(1 - e_{opt})^2}$$

where $C_1 = \frac{\text{Var}(Y^{(1)})}{\mathbb{E}[Y^{(1)}]^2}$ and $C_0 = \frac{\text{Var}(Y^{(0)})}{\mathbb{E}[Y^{(0)}]^2}$.

- If $\frac{\text{Var}(Y^{(1)})}{\mathbb{E}[Y^{(1)}]^2} = \frac{\text{Var}(Y^{(0)})}{\mathbb{E}[Y^{(0)}]^2}$:

$$e_{opt} = 0.5$$

- otherwise:

$$e_{opt} = \frac{C_1 - \sqrt{C_1 C_0}}{C_1 - C_0} \in (0, 1)$$

$\square$

### 7.2.2. RISK RATIO HORVITZ-THOMSON ESTIMATOR

**Definition 7.4** (**Risk Ratio Horvitz-Thomson estimator**). Grant Assumption 2.2 and Assumption 2.3. The Risk Ratio Horvitz-Thomson estimator denoted $\hat{\tau}_{\text{RR,HT,n}}$ is defined as,

$$\hat{\tau}_{\text{RR,HT,n}} = \frac{\sum_{i=1}^n \frac{T_i Y_i}{e}}{\sum_{i=1}^n \frac{(1-T_i)Y_i}{1-e}} \tag{31}$$

if $\sum_{i=1}^n T_i < n$ and $0$ otherwise.

Within the context of a Bernoulli trial, Proposition 7.5 proves that the Risk Ratio Horvitz-Thompson estimator is asymptotically unbiased and normally distributed.

**Proposition 7.5** (**Asymptotic normality of $\hat{\tau}_{\text{RR,HT,n}}$**). *Under Assumption 2.2 and Assumption 2.3, the Risk Ratio Horvitz-Thompson estimator is asymptotically unbiased and satisfies*

$$\sqrt{n}\left(\hat{\tau}_{\text{RR,HT,n}} - \tau_{RR}\right) \xrightarrow{d} \mathcal{N}\left(0, V_{RR,HT}\right) \tag{32}$$

*where* $V_{RR,HT} = \tau_{RR}^2 \left( \frac{\mathbb{E}[(Y^{(1)})^2]}{e\mathbb{E}[Y^{(1)}]^2} + \frac{\mathbb{E}[(Y^{(0)})^2]}{(1-e)\mathbb{E}[Y^{(0)}]^2} \right)$.

*If we assume that for all $i$, $M \geq Y_i \geq m > 0$ and $0 < \sum_{i=1}^n T_i < n$, we also have:*

$$|Bias(\hat{\tau}_{RR,HT,n})| \leq \frac{2M^3(1-e)^3}{nm^3e^3}$$

$$|\text{Var}(\hat{\tau}_{RR,HT,n})| \leq \frac{4M^4(1-e)^6}{nm^6e^4}$$

*Proof of Proposition 7.5.*

**Asymptotic Bias and Variance.** Let $Z_i := (T_i, Y_i)$ and define $g_0(Z_i) = \frac{(1-T_i)Y_i}{1-e}$ and $g_1(Z_i) = \frac{T_i Y_i}{e}$. First, we evaluate the expectation of $g_1(Z_i)$:

$$\mathbb{E}\left[g_1(Z_i)\right] = \mathbb{E}\left[\frac{T_i Y_i}{e}\right] \qquad \text{(by i.i.d)}$$

$$= \mathbb{E}\left[\frac{T_i Y_i^{(1)}}{e}\right] \qquad \text{(by SUTVA)}$$

$$= \mathbb{E}\left[\frac{T_i}{e}\right] \mathbb{E}\left[Y_i^{(1)}\right] \qquad \text{(by ignorability)}$$

$$= \mathbb{E}\left[Y_i^{(1)}\right] \qquad \text{(by Trial positivity)}$$

Similarly, we can find the expectation of $g_0(Z_i)$:

$$\mathbb{E}\left[g_0(Z_i)\right] = \mathbb{E}\left[Y^{(0)}\right] > 0.$$

Thus, according to Theorem 7.1, we have $\sqrt{n}\left(\hat{\tau}_{\text{RR-HT, n}} - \tau_{\text{RR}}\right) \xrightarrow{d} \mathcal{N}\left(0, V_{\text{RR-HT}}\right)$, with

$$V_{\text{RR-HT}} = \left(\frac{\tau_1}{\tau_0}\right)^2 \text{Var}\left(\frac{g_1(Z)}{\tau_1} - \frac{g_0(Z)}{\tau_0}\right)$$

$$= \tau_{\text{RR}}^2 \, \text{Var}\left(\frac{TY}{e\mathbb{E}\left[Y^{(1)}\right]} - \frac{(1-T)Y}{(1-e)\mathbb{E}\left[Y^{(0)}\right]}\right).$$

Next, we evaluate the variance terms separately:

$$\text{Var}\left(\frac{TY}{e\mathbb{E}\left[Y^{(1)}\right]}\right) = \frac{1}{\mathbb{E}\left[Y^{(1)}\right]^2 e^2} \text{Var}\left(TY\right)$$

$$= \frac{1}{\mathbb{E}\left[Y^{(1)}\right]^2 e^2}\left(\mathbb{E}\left[(TY)^2\right] - \mathbb{E}\left[TY\right]^2\right)$$

$$= \frac{1}{\mathbb{E}\left[Y^{(1)}\right]^2 e^2}\left(\mathbb{E}\left[T\left(Y\right)^2\right] - \mathbb{E}\left[TY\right]^2\right) \qquad (T \text{ is binary})$$

$$= \frac{1}{\mathbb{E}\left[Y^{(1)}\right]^2 e^2}\left(\mathbb{E}\left[T\left(Y^{(1)}\right)^2\right] - \mathbb{E}\left[TY^{(1)}\right]^2\right) \qquad (\text{by SUTVA})$$

$$= \frac{1}{\mathbb{E}\left[Y^{(1)}\right]^2 e^2}\left(e\mathbb{E}\left[\left(Y^{(1)}\right)^2\right] - e^2\mathbb{E}\left[Y^{(1)}\right]^2\right) \qquad (\text{by ignorability})$$

$$= \frac{\mathbb{E}\left[\left(Y^{(1)}\right)^2\right]}{e\mathbb{E}\left[Y^{(1)}\right]^2} - 1.$$

Similarly, we find the variance of the second term:

$$\text{Var}\left(\frac{(1-T)Y}{(1-e)\mathbb{E}\left[Y^{(0)}\right]}\right) = \frac{\mathbb{E}\left[\left(Y^{(0)}\right)^2\right]}{(1-e)\mathbb{E}\left[Y^{(0)}\right]^2} - 1.$$

Finally, we compute the covariance between the two terms:

$$\text{Cov}\left(\frac{TY}{e\mathbb{E}\left[Y^{(1)}\right]}, \frac{(1-T)Y}{(1-e)\mathbb{E}\left[Y^{(0)}\right]}\right) = \frac{\text{Cov}(TY, (1-T)Y)}{e\mathbb{E}\left[Y^{(1)}\right](1-e)\mathbb{E}\left[Y^{(0)}\right]}$$

$$= \frac{\left(\mathbb{E}\left[T(1-T)Y^2\right] - \mathbb{E}\left[TY\right]\mathbb{E}\left[(1-T)Y\right]\right)}{e\mathbb{E}\left[Y^{(1)}\right](1-e)\mathbb{E}\left[Y^{(0)}\right]}$$

$$= \frac{-\mathbb{E}\left[TY\right]\mathbb{E}\left[(1-T)Y\right]}{e\mathbb{E}\left[Y^{(1)}\right](1-e)\mathbb{E}\left[Y^{(0)}\right]}$$

$$= -1.$$

Using Bienayme's identity, we finally obtain:

$$V_{\text{RR-HT}} = \tau_{\text{RR}}^2 \left( \frac{\mathbb{E}\left[\left(Y^{(1)}\right)^2\right]}{e\mathbb{E}\left[Y^{(1)}\right]^2} + \frac{\mathbb{E}\left[\left(Y^{(0)}\right)^2\right]}{(1-e)\mathbb{E}\left[Y^{(0)}\right]^2} \right).$$

As a consequence an estimator $\hat{V}_{RR-HT}$ can be derived:

$$\hat{V}_{RR-HT} = \hat{\tau}_{\text{RR,HT,n}}^2 \left( \frac{\frac{1}{n}\sum_{T_i=1} Y_i^2}{\hat{e}\left(\frac{1}{n}\sum_{T_i=1} Y_i\right)^2} + \frac{\frac{1}{n}\sum_{T_i=0} Y_i^2}{(1-\hat{e})\left(\frac{1}{n}\sum_{T_i=0} Y_i\right)^2} \right) \tag{33}$$

**Finite sample Bias and Variance.** Let $T_1(\boldsymbol{Z}) = \frac{1}{n}\sum_{i=1}^n \frac{T_i Y_i}{e}$ and $T_0(\boldsymbol{Z}) = \frac{1}{n}\sum_{i=1}^n \frac{(1-T_i)Y_i}{1-e}$ where $\boldsymbol{Z} = (Z_1, \ldots, Z_n)$. First, consider the variance of $T_1(\boldsymbol{Z})$:

$$
\begin{aligned}
\text{Var}(T_1(\boldsymbol{Z})) &= \frac{1}{ne^2} \text{Var}\left(T_i Y_i\right) && \text{(by i.i.d)} \\
&= \frac{1}{ne^2}\left( \mathbb{E}\left[(T_i Y_i)^2\right] - \mathbb{E}\left[T_i Y_i\right]^2 \right) \\
&= \frac{\mathbb{E}\left[\left(Y^{(1)}\right)^2\right] - e\mathbb{E}\left[Y^{(1)}\right]^2}{ne}.
\end{aligned}
$$

Thus $\text{Var}(T_1(\boldsymbol{Z})) = O_p(1/n)$ and similarly $\text{Var}(T_0(\boldsymbol{Z})) = O_p(1/n)$. Next, we show that $T_0(\boldsymbol{Z})$ is bounded:

$$
\begin{aligned}
T_0(\boldsymbol{Z}) &= \frac{1}{n}\sum_{i=1}^n \frac{(1-T_i)Y_i}{1-e} \\
&= \frac{1}{n(1-e)}\sum_{i=1}^n (1-T_i)Y_i \\
&\geq \frac{m}{(1-e)}\sum_{i=1}^n (1-T_i) && \text{(since } Y_i \geq m > 0\text{)} \\
&\geq \frac{m}{(1-e)} && \left(\text{as } \sum_{i=1}^n T_i < n\right).
\end{aligned}
$$

Similarly, we also have the upper bound

$$
\begin{aligned}
T_0(\boldsymbol{Z}) &= \frac{1}{n}\sum_{i=1}^n \frac{(1-T_i)Y_i}{1-e} \\
&= \frac{1}{ne}\sum_{i=1}^n (1-T_i)Y_i \\
&\leq \frac{1}{ne}\sum_{i=1}^n Y_i && \text{(since } T \text{ is binary)} \\
&\leq \frac{M}{e} && \text{(since } Y_i \leq M\text{)}.
\end{aligned}
$$

Similarly, we have $T_1(\boldsymbol{Z}) \leq \frac{M}{e}$. Therefore, we have shown that $T_1(\boldsymbol{Z})$ and $T_0(\boldsymbol{Z})$ are unbiased estimators of $\mathbb{E}\left[Y^{(1)}\right]$ and $\mathbb{E}\left[Y^{(0)}\right] > 0$, respectively. We also established that $M/e \geq T_0(\boldsymbol{Z}) \geq m/(1-e) > 0$ and $|T_1(\boldsymbol{Z})| \leq M/e$. Furthermore,

we pointed out that $\text{Var}(T_1(\boldsymbol{Z})) = O_p\left(\frac{1}{n}\right)$ and $\text{Var}(T_0(\boldsymbol{Z})) = O_p\left(\frac{1}{n}\right)$. Applying Theorem 7.3, we obtain:

$$\left|\mathbb{E}\left[\hat{\tau}_{\text{RR, HT, n}}\right] - \tau_{\text{RR}}\right| \leq \frac{M^2(1-e)^2}{ne^2m^2}\left(\frac{M(1-e)}{me}+1\right) \leq \frac{2M^3(1-e)^3}{nm^3e^3},$$

and

$$\left|\text{Var}(\hat{\tau}_{\text{RR, HT, n}}) - V_{\text{RR, HT}}\right| \leq \frac{2M^2(1-e)^4}{nm^4e^2}\left(\frac{M^2(1-e)^2}{m^2e^2}+1\right) \leq \frac{4M^4(1-e)^6}{nm^6e^4}.$$

**Optimal choice of** $e$ The optimal value of $e_{opt}$ is the one that minimizes the variance of the Ratio Horvitz-Thomson estimator. Therefore, we need to solve:

$$\inf_{e \in (0,1)} \tau_{\text{RR}}^2 \left(\frac{\mathbb{E}\left[\left(Y^{(1)}\right)^2\right]}{e\mathbb{E}\left[Y^{(1)}\right]^2} + \frac{\mathbb{E}\left[\left(Y^{(0)}\right)^2\right]}{(1-e)\mathbb{E}\left[Y^{(0)}\right]^2}\right)$$

Noting that the variance we want to minimize is convex in $e$, we can derive the variance and set it to 0 to find $e_{opt}$. We have:

$$\frac{C_1}{e_{opt}^2} = \frac{C_0}{(1-e_{opt})^2}$$

where $C_1 = \frac{\mathbb{E}\left[\left(Y^{(1)}\right)^2\right]}{\mathbb{E}\left[Y^{(1)}\right]^2}$ and $C_0 = \frac{\mathbb{E}\left[\left(Y^{(0)}\right)^2\right]}{\mathbb{E}\left[Y^{(0)}\right]^2}$.

- If $\frac{\text{Var}\left(Y^{(1)}\right)}{\mathbb{E}[Y^{(1)}]^2} = \frac{\text{Var}\left(Y^{(0)}\right)}{\mathbb{E}[Y^{(0)}]^2}$:

$$e_{opt} = 0.5$$

- otherwise:

$$e_{opt} = \frac{\mathbb{E}\left[\left(Y^{(1)}\right)^2\right]\mathbb{E}\left[Y^{(0)}\right]^2 - \sqrt{\mathbb{E}\left[\left(Y^{(1)}\right)^2\right]\mathbb{E}\left[\left(Y^{(0)}\right)^2\right]}\mathbb{E}\left[Y^{(1)}\right]\mathbb{E}\left[Y^{(0)}\right]}{\mathbb{E}\left[\left(Y^{(1)}\right)^2\right]\mathbb{E}\left[Y^{(0)}\right]^2 - \mathbb{E}\left[\left(Y^{(0)}\right)^2\right]\mathbb{E}\left[Y^{(1)}\right]^2} \in (0,1)$$

$\square$

### 7.2.3. Link with existing asymptotic confidence intervals

According to Proposition 2.4, a $(1-\alpha)$ asymptotic confidence interval for $\tau_{\text{RR}}$ is given by

$$\left[\hat{\tau}_{\text{RR,N,n}} \pm \frac{\sqrt{\widehat{V_{\text{RR,N}}}}z_{1-\alpha/2}}{n}\right] \tag{34}$$

with $\widehat{V_{\text{RR,N}}}$ an estimator of

$$V_{\text{RR,N}} = \tau_{\text{RR}}^2 \left(\frac{\text{Var}\left(Y^{(1)}\right)}{e\mathbb{E}[Y^{(1)}]^2} + \frac{\text{Var}\left(Y^{(0)}\right)}{(1-e)\mathbb{E}[Y^{(0)}]^2}\right).$$

Now, assume that $Y^{(0)}, Y^{(1)} \in \{0, 1\}$ with associated probabilities $\mathbb{P}[Y^{(0)} = 1] = p_0$ and $\mathbb{P}[Y^{(1)} = 1] = p_1$. In this setting, the variance $V_{\mathrm{RR,N}}$ takes the form

$$
\begin{aligned}
\frac{V_{\mathrm{RR,N}}}{n} &= \frac{\tau_{\mathrm{RR}}^2}{n} \left( \frac{\mathrm{Var}\left(Y^{(1)}\right)}{e\mathbb{E}[Y^{(1)}]^2} + \frac{\mathrm{Var}\left(Y^{(0)}\right)}{(1-e)\mathbb{E}[Y^{(0)}]^2} \right) \\
&= \tau_{\mathrm{RR}}^2 \left( \frac{p_1(1-p_1)}{N_1 p_1^2} + \frac{p_0(1-p_0)}{N_0 p_0^2} \right) \\
&= \tau_{\mathrm{RR}}^2 \left( \frac{1-p_1}{N_1 p_1} + \frac{1-p_0}{N_0 p_0} \right) \\
&= \tau_{\mathrm{RR}}^2 \left( \frac{1}{N_1 p_1} - \frac{1}{N_1} + \frac{1}{N_0 p_0} - \frac{1}{N_0} \right) \\
&= \tau_{\mathrm{RR}}^2 \left( \frac{1}{N_1 p_1} - \frac{1}{N_1} + \frac{1}{N_0 p_0} - \frac{1}{N_0} \right).
\end{aligned}
$$

An estimation of such a quantity can be constructed by replacing $p_1$ (resp. $p_0$) by $(1/N_1) \sum_{i=1}^n T_i Y_i$ (resp. $(1/N_0) \sum_{i=1}^n (1 - T_i) Y_i$), which leads to

$$
\frac{\widehat{V_{\mathrm{RR,N}}}}{n} = \hat{\tau}_{\mathrm{RR}}^2 \left( \frac{1}{\sum_{i=1}^n T_i Y_i} - \frac{1}{N_1} + \frac{1}{\sum_{i=1}^n (1 - T_i) Y_i} - \frac{1}{N_0} \right). \tag{35}
$$

Thus, a $(1 - \alpha)$ asymptotic confidence interval for $\tau_{\mathrm{RR}}$ is given by

$$
\left[ \hat{\tau}_{\mathrm{RR,N,n}} \pm z_{1-\alpha/2} \hat{\tau}_{\mathrm{RR,N,n}} \sqrt{\left( \frac{1}{\sum_{i=1}^n T_i Y_i} - \frac{1}{N_1} + \frac{1}{\sum_{i=1}^n (1 - T_i) Y_i} - \frac{1}{N_0} \right)} \right] \tag{36}
$$

$$
= \left[ \hat{\tau}_{\mathrm{RR,N,n}} \left( 1 \pm z_{1-\alpha/2} \sqrt{\left( \frac{1}{\sum_{i=1}^n T_i Y_i} - \frac{1}{N_1} + \frac{1}{\sum_{i=1}^n (1 - T_i) Y_i} - \frac{1}{N_0} \right)} \right) \right]. \tag{37}
$$

Finally, since $e^x$ is equivalent to $1 + x$ near $x = 0$, the above interval is equivalent to that given by (3), which concludes the proof.

### 7.2.4. DELTA METHOD WITH log FUNCTION

According to Proposition 2.4, we know that

$$
\sqrt{n} \left( \hat{\tau}_{\mathrm{RR,N,n}} - \tau_{\mathrm{RR}} \right) \xrightarrow{d} \mathcal{N} \left( 0, V_{\mathrm{RR,N}} \right), \tag{38}
$$

where

$$
V_{\mathrm{RR,N}} = \tau_{\mathrm{RR}}^2 \left( \frac{\mathrm{Var}\left(Y^{(1)}\right)}{e\mathbb{E}[Y^{(1)}]^2} + \frac{\mathrm{Var}\left(Y^{(0)}\right)}{(1-e)\mathbb{E}[Y^{(0)}]^2} \right).
$$

Using the Delta method, with the function $\theta \mapsto \log(\theta)$, we obtain

$$
\sqrt{n} \left( \log(\hat{\tau}_{\mathrm{RR,N,n}}) - \log(\tau_{\mathrm{RR}}) \right) \xrightarrow{d} \mathcal{N} \left( 0, (1/\tau_{\mathrm{RR}})^2 V_{\mathrm{RR,N}} \right). \tag{39}
$$

Thus, a $(1 - \alpha)$ asymptotic confidence interval for $\log(\tau_{\mathrm{RR}})$ is given by

$$
\left[ \log(\hat{\tau}_{\mathrm{RR,N,n}}) \pm z_{1-\alpha/2} \sqrt{\frac{V_{\mathrm{RR,N}}}{n \tau_{\mathrm{RR}}^2}} \right]. \tag{40}
$$

Letting $V_{\log \mathrm{RR,N}} = V_{\mathrm{RR,N}} / \tau_{\mathrm{RR}}^2$, a $(1 - \alpha)$ asymptotic confidence interval for $\tau_{\mathrm{RR}}$ is

$$
\left[ \hat{\tau}_{\mathrm{RR,N,n}} \exp \left( \pm z_{1-\alpha/2} \sqrt{\frac{V_{\log \mathrm{RR,N}}}{n}} \right) \right]. \tag{41}
$$

Now, note that, if $Y^{(0)}, Y^{(1)} \in \{0, 1\}$ with $\mathbb{P}[Y^{(t)} = 1] = p_t$, we have

$$V_{\text{log RR,N}} = \frac{\text{Var}\left(Y^{(1)}\right)}{e\mathbb{E}[Y^{(1)}]^2} + \frac{\text{Var}\left(Y^{(0)}\right)}{(1 - e)\mathbb{E}[Y^{(0)}]^2} \tag{42}$$

$$= \frac{p_1(1 - p_1)}{ep_1^2} + \frac{p_0(1 - p_0)}{(1 - e)p_0^2} \tag{43}$$

$$= \frac{1}{ep_1} - \frac{1}{e} + \frac{1}{ep_0} - \frac{1}{1 - e}. \tag{44}$$

Hence,

$$\frac{V_{\text{log RR,N}}}{n} = \frac{1}{enp_1} - \frac{1}{en} + \frac{1}{enp_0} - \frac{1}{n(1 - e)}, \tag{45}$$

which can be estimated replacing $ne$ (resp. $n(1 - e)$) by $N_1 = \sum_{i=1}^{n} T_i$ (resp. $N_0 = n - N_1$ and $enp_1$ (resp. $enp_0$) by $\sum_{i=1}^{n} Y_i T_i$ (resp. $\sum_{i=1}^{n} Y_i(1 - T_i)$). Replacing $V_{\text{log RR,N}}/n$ by such an estimate in the asymptotic confidence interval (41) leads to the well-known formula presented in Equation (3).

### 7.3. Proofs of Section 3

#### 7.3.1. RISK RATIO INVERSE PROPENSITY WEIGHTING

*Proof of Proposition 3.3.*
**Asymptotic bias and variance of the oracle Risk Ratio IPW estimator** Recall that the oracle Risk Ratio IPW is defined as

$$\tau_{\text{RR,IPW}}^{\star} = \left(\sum_{i=1}^{n} \frac{T_i Y_i}{e(X_i)}\right) \Big/ \left(\sum_{i=1}^{n} \frac{(1 - T_i)Y_i}{1 - e(X_i)}\right),$$

where the propensity score $e$ is assumed to be known. Let us define $g_1(Z) = TY/e(X)$ and $g_0(Z) = (1 - T)Y/(1 - e(X))$ with $Z = (X, T, Y)$. Since

$$\frac{m}{1 - \eta} \leq g_1(Z) \leq \frac{M}{\eta} \quad \text{and} \quad g_0(Z) \leq \frac{M}{\eta},$$

the function $g_0$ and $g_1$ are bounded from above and below and thus square integrable. Besides, $\mathbb{E}[g_0(Z_i)] = \mathbb{E}\left[Y^{(0)}\right]$ and $\mathbb{E}[g_1(Z_i)] = \mathbb{E}\left[Y^{(1)}\right]$. We can therefore apply Theorem 7.3 and conclude that

$$\sqrt{n}(\tau_{\text{RR,IPW}}^{\star} - \tau_{\text{RR}}) \to \mathcal{N}(0, V_{\text{RR,IPW}}),$$

where

$$V_{\text{RR,IPW}} = \tau_{\text{RR}}^2 \, \text{Var}\left(\frac{\frac{T_i Y_i}{e(X_i)}}{\mathbb{E}\left[Y^{(1)}\right]} - \frac{\frac{(1 - T_i)Y_i}{1 - e(X_i)}}{\mathbb{E}\left[Y^{(1)}\right]}\right). \tag{46}$$

Moreover,

$$\text{Var}\left(\frac{TY}{e(X)}\right) = \mathbb{E}\left[\left(\frac{TY}{e(X)}\right)^2\right] - \mathbb{E}\left[\frac{TY}{e(X)}\right]^2$$

$$= \mathbb{E}\left[\frac{TY^2}{e(X)^2}\right] - \mathbb{E}\left[Y^{(1)}\right]^2$$

$$= \mathbb{E}\left[\frac{1}{e(X)^2}\mathbb{E}\left[T(Y^{(1)})^2|X\right]\right] - \mathbb{E}\left[Y^{(1)}\right]^2$$

$$= \mathbb{E}\left[\frac{1}{e(X)}\mathbb{E}\left[(Y^{(1)})^2|X\right]\right] - \mathbb{E}\left[Y^{(1)}\right]^2$$

$$= \mathbb{E}\left[\frac{1}{e(X)}\mathbb{E}\left[(Y^{(1)})^2|X\right]\right] - \mathbb{E}\left[Y^{(1)}\right]^2$$

$$= \mathbb{E}\left[\frac{(Y^{(1)})^2}{e(X)}\right] - \mathbb{E}\left[Y^{(1)}\right]^2.$$

Similarly

$$\text{Var}\left(\frac{(1-T)Y}{1-e(X)}\right) = \mathbb{E}\left[\frac{(Y^{(0)})^2}{1-e(X)}\right] - \mathbb{E}\left[Y^{(0)}\right]^2.$$

Additionally, the covariance satisfies

$$
\begin{aligned}
\text{Cov}\left(\frac{TY}{e(X)}, \frac{(1-T)Y}{1-e(X)}\right) &= \mathbb{E}\left[\left(\frac{TY}{e(X)} - \mathbb{E}\left[Y^{(1)}\right]\right)\left(\frac{(1-T)Y}{1-e(X)} - \mathbb{E}\left[Y^{(0)}\right]\right)\right] \\
&= \mathbb{E}\left[\frac{TY}{e(X)}\frac{(1-T)Y}{1-e(X)}\right] - \mathbb{E}\left[Y^{(1)}\right]\mathbb{E}\left[\frac{(1-T)Y}{1-e(X)}\right] \\
&\quad - \mathbb{E}\left[Y^{(0)}\right]\mathbb{E}\left[\frac{TY}{e(X)}\right] + \mathbb{E}\left[Y^{(1)}\right]\mathbb{E}\left[Y^{(0)}\right] \\
&= -\mathbb{E}\left[Y^{(1)}\right]\mathbb{E}\left[Y^{(0)}\right].
\end{aligned}
$$

Therefore, we get that

$$V_{\text{RR,IPW}} = \tau_{\text{RR}}^2 \left(\frac{\mathbb{E}\left[\frac{(Y^{(1)})^2}{e(X)}\right]}{\mathbb{E}\left[Y^{(1)}\right]^2} + \frac{\mathbb{E}\left[\frac{(Y^{(0)})^2}{1-e(X)}\right]}{\mathbb{E}\left[Y^{(0)}\right]^2}\right).$$

As a consequence an estimator $\hat{V}_{RR-IPW}$ can be derived:

$$\hat{V}_{RR-IPW} = \hat{\tau}_{\text{RR,IPW,n}}^2 \left(\frac{\frac{1}{n}\sum_{T_i=1}\left(\frac{Y_i}{\hat{e}(x_i)}\right)^2}{\left(\frac{1}{n}\sum_{T_i=1}Y_i\right)^2} + \frac{\frac{1}{n}\sum_{T_i=0}\left(\frac{Y_i}{1-\hat{e}(x_i)}\right)^2}{\left(\frac{1}{n}\sum_{T_i=0}Y_i\right)^2}\right) \tag{47}$$

Since we have $\mathbb{E}\left[\left(\frac{TY}{e(X)}\right)^2\right] = \mathbb{E}\left[\frac{(Y^{(1)})^2}{e(X)}\right]$.

**Finite sample bias and variance of the oracle Risk Ratio IPW estimator** Let $T_1(\mathbf{Z}) = \frac{1}{n}\sum_{i=1}^{n}\frac{T_iY_i}{e(X_i)}$ and $T_0(\mathbf{Z}) = \frac{1}{n}\sum_{i=1}^{n}\frac{(1-T_i)Y_i}{1-e(X_i)}$ where $\mathbf{Z} = (Z_1, \ldots, Z_n)$. We first show that $\text{Var}(T_1(\mathbf{Z})) = O_p\left(\frac{1}{n}\right)$ and $\text{Var}(T_0(\mathbf{Z})) = O_p\left(\frac{1}{n}\right)$:

$$
\begin{aligned}
\text{Var}(T_1(\mathbf{Z})) &= \frac{1}{n^2}\text{Var}\left(\sum_{i=1}^{n}\frac{T_iY_i}{e(X_i)}\right) \\
&= \frac{1}{n^2}\sum_{i=1}^{n}\text{Var}\left(\frac{T_iY_i}{e(X_i)}\right) && \text{(by i.i.d.)} \\
&= \frac{1}{n}\left(\mathbb{E}\left[\left(\frac{T_iY_i}{e(X_i)}\right)^2\right] - \mathbb{E}\left[\frac{T_iY_i}{e(X_i)}\right]^2\right) && \text{(by law of total expectation)} \\
&= \frac{\mathbb{E}\left[\frac{(Y^{(1)})^2}{e(X_i)}\right] - \mathbb{E}\left[Y^{(1)}\right]^2}{n} \\
&= O_p\left(\frac{1}{n}\right)
\end{aligned}
$$

Similarly, $\text{Var}(T_0(\mathbf{Z})) = O_p\left(\frac{1}{n}\right)$. And we also have:

$$\mathbb{E}[T_1(\mathbf{Z})] = \mathbb{E}\left[\frac{T_iY_i}{e(X_i)}\right] = \mathbb{E}\left[Y^{(1)}\right]$$

$$\mathbb{E}\left[T_0(\boldsymbol{Z})\right] = \mathbb{E}\left[\frac{(1 - T_i)Y_i}{1 - e(X_i)}\right] = \mathbb{E}\left[Y^{(0)}\right]$$

Therefore, we showed that $T_1(\boldsymbol{Z})$ and $T_0(\boldsymbol{Z})$ are respectively unbiased estimators of $\mathbb{E}\left[Y^{(1)}\right]$ and $\mathbb{E}\left[Y^{(0)}\right] > 0$ such that $\mathrm{Var}(T_1(\boldsymbol{Z})) = O_p\left(\frac{1}{n}\right)$ and $\mathrm{Var}(T_0(\boldsymbol{Z})) = O_p\left(\frac{1}{n}\right)$. By assumption,

$$\frac{m}{1 - \eta} \le T_0(\boldsymbol{Z}) \le \frac{M}{\eta} \quad \text{and} \quad T_1(\boldsymbol{Z}) \le \frac{M}{\eta},$$

thus $T_0(\boldsymbol{Z})$ and $T_1(\boldsymbol{Z})$ are bounded. Applying Theorem 7.3, we obtain

$$\left|\mathbb{E}\left[\hat{\tau}_{\text{RR, IPW, n}}\right] - \tau_{\text{RR}}\right| \le \frac{2M^3(1 - \eta)^3}{nm^3\eta^3},$$

and

$$\left|\mathrm{Var}(\hat{\tau}_{\text{RR, HT, n}}) - V_{\text{RR, HT}}\right| \le \frac{4M^4(1 - \eta)^6}{nm^6\eta^4}.$$

$\square$

### 7.3.2. RISK RATIO INVERSE PROPENSITY WEIGHTING IN LOGISTIC MODELS

*Proof of Proposition 3.5.* The likelihood function $L\left(\boldsymbol{\beta}\right)$ is

$$L\left(\boldsymbol{\beta}\right) = \prod_{i=1}^{n} \mathrm{P}\left(T = t_i \mid X = x_i\right) = \prod_{i=1}^{n} e\left(x_i; \boldsymbol{\beta}\right)^{t_i} \left(1 - e\left(x_i; \boldsymbol{\beta}\right)\right)^{1 - t_i}$$

where we define $e\left(X; \boldsymbol{\beta}\right) = \{1 + \exp(-X^\top \beta_1 - \beta_0)\}^{-1}$. Now, taking minus the logarithm of this expression, the log likelihood function, denoted $\ln L\left(\boldsymbol{\beta}\right)$, we obtain,

$$-\ln L\left(\boldsymbol{\beta}\right) = -\sum_{i=1}^{n} t_i \log(e(X_i; \boldsymbol{\beta})) + (1 - t_i)\log(1 - e(X_i; \boldsymbol{\beta})).$$

The minimization of this quantity can be obtained when looking for the root of the derivative, so that we obtain the following expression,

$$-\frac{\partial}{\partial \boldsymbol{\beta_0}}\ln L\left(\boldsymbol{\beta}\right) = -\sum_{i=1}^{n} \frac{T_i - e\left(X_i; \boldsymbol{\beta}\right)}{e\left(X_i; \boldsymbol{\beta}\right)\left(1 - e\left(X_i; \boldsymbol{\beta}\right)\right)} \frac{\partial}{\partial \boldsymbol{\beta_0}} e\left(X_i; \boldsymbol{\beta}\right).$$

$$-\frac{\partial}{\partial \boldsymbol{\beta_1}}\ln L\left(\boldsymbol{\beta}\right) = -\sum_{i=1}^{n} \frac{T_i - e\left(X_i; \boldsymbol{\beta}\right)}{e\left(X_i; \boldsymbol{\beta}\right)\left(1 - e\left(X_i; \boldsymbol{\beta}\right)\right)} \frac{\partial}{\partial \boldsymbol{\beta_1}} e\left(X_i; \boldsymbol{\beta}\right).$$

Therefore,

$$\frac{\partial}{\partial \boldsymbol{\beta_0}}\ln L\left(\boldsymbol{\beta}\right) = -\sum_{i=1}^{n}\left(T_i - e\left(X_i; \boldsymbol{\beta}\right)\right) \quad \text{and} \quad \frac{\partial}{\partial \boldsymbol{\beta_1}}\ln L\left(\boldsymbol{\beta}\right) = -\sum_{i=1}^{n} X_i\left(T_i - e\left(X_i; \boldsymbol{\beta}\right)\right) \tag{48}$$

In particular, if we apply 48 for the maximum likelihood estimator $\hat{\boldsymbol{\beta}}_n$, and if we define $\widetilde{X} := (1, X)$, we have:

$$\frac{\partial}{\partial \boldsymbol{\beta}} \ln L\left(\boldsymbol{\beta}\right)\bigg|_{\boldsymbol{\beta}=\hat{\boldsymbol{\beta}}_n} = 0 \iff \sum_{i=1}^{n} \widetilde{X}_i\left(T_i - e\left(X_i; \hat{\boldsymbol{\beta}}_n\right)\right) = 0.$$

Let $Z = (X, T, Y)$ and $\boldsymbol{\theta} = (\boldsymbol{\beta}, \theta_0, \theta_1, \theta_2)$, we define $\psi$ and $\hat{\boldsymbol{\theta}}_n$ as:

$$\psi(Z, \boldsymbol{\theta}) = \begin{pmatrix} \widetilde{X}(T - e(X, \boldsymbol{\beta})) \\ \frac{(1-T)Y}{1-e(X,\boldsymbol{\beta})} - \theta_0 \\ \frac{TY}{e(X,\boldsymbol{\beta})} - \theta_1 \\ \theta_1 - \theta_2\theta_0 \end{pmatrix} \quad \text{and} \quad \hat{\boldsymbol{\theta}}_n = \begin{pmatrix} \hat{\boldsymbol{\beta}}_n \\ \hat{\tau}_{\text{IPW},0} \\ \hat{\tau}_{\text{IPW},1} \\ \hat{\tau}_{\text{IPW},1}/\hat{\tau}_{\text{IPW},0} \end{pmatrix}$$

where $\hat{\tau}_{\text{IPW},1} = \frac{1}{n}\sum_{i=1}^{n} \frac{TY}{e(X,\hat{\boldsymbol{\beta}}_n)}$ and $\hat{\tau}_{\text{IPW},0} = \frac{1}{n}\sum_{i=1}^{n} \frac{(1-T)Y}{1-e(X,\hat{\boldsymbol{\beta}}_n)}$. One can note that:

$$\sum_{i=1}^{n} \psi_1(Z_i, \hat{\boldsymbol{\theta}}_n) = \sum_{i=1}^{n} \widetilde{X}_i\left(T_i - e\left(X_i; \hat{\boldsymbol{\beta}}_n\right)\right) = 0.$$

$$\sum_{i=1}^{n} \psi_2(Z_i, \hat{\boldsymbol{\theta}}_n) = \sum_{i=1}^{n}\left(\frac{(1-T_i)Y_i}{1-e(X_i, \hat{\boldsymbol{\beta}}_n)} - \frac{1}{n}\sum_{j=1}^{n}\frac{(1-T_j)Y_j}{1-e(X_j, \hat{\boldsymbol{\beta}}_n)}\right) = 0.$$

$$\sum_{i=1}^{n} \psi_3(Z_i, \hat{\boldsymbol{\theta}}_n) = \sum_{i=1}^{n}\left(\frac{T_i Y_i}{e(X_i, \hat{\boldsymbol{\beta}}_n)} - \frac{1}{n}\sum_{j=1}^{n}\frac{T_j Y_j}{e(X_j, \hat{\boldsymbol{\beta}}_n)}\right) = 0.$$

$$\sum_{i=1}^{n} \psi_4(Z_i, \hat{\boldsymbol{\theta}}_n) = \sum_{i=1}^{n} \hat{\tau}_{\text{IPW},1} - \underbrace{\frac{\hat{\tau}_{\text{IPW},1}}{\hat{\tau}_{\text{IPW},0}}\hat{\tau}_{\text{IPW},0}}_{\hat{\tau}_{\text{IPW},1}} = 0.$$

Gathering the three previous equalities, we obtain

$$\sum_{i=1}^{n} \psi(Z_i, \hat{\boldsymbol{\theta}}_n) = 0, \tag{49}$$

which proves that $\hat{\boldsymbol{\theta}}_n$ is an M-estimator of type $\psi$ (see Stefanski & Boos, 2002). Furthermore, letting $\boldsymbol{\theta}_\infty = (\boldsymbol{\beta}_\infty, \mathbb{E}\left[Y^{(0)}\right], \mathbb{E}\left[Y^{(1)}\right], \mathbb{E}\left[Y^{(1)}\right]/\mathbb{E}\left[Y^{(0)}\right])$, we can compute the following quantities:

$$\begin{aligned}
\mathbb{E}\left[\psi_1(Z, \boldsymbol{\theta}_\infty)\right] &= \mathbb{E}\left[\widetilde{X}(T - e(X))\right] \\
&= \mathbb{E}\left[\widetilde{X} \cdot \mathbb{E}\left[T - e(X) \mid X\right]\right] \quad \text{(Law of Total Probability)} \\
&= \mathbb{E}\left[\widetilde{X} \cdot (\mathbb{E}\left[T \mid X\right] - e(X))\right] \quad (e(X) \text{ is a function of } X) \\
&= 0 \quad \text{(Definition of } e(X))
\end{aligned}$$

Furthermore, note that:

$$\mathbb{E}\left[\psi_2(Z, \boldsymbol{\theta}_\infty)\right] = \mathbb{E}\left[\frac{1}{n}\sum_{i=1}^{n}\left(\frac{T_i Y_i}{e(X_i)}\right)\right] - \mathbb{E}[Y^{(1)}],$$

and the following holds:

$$\mathbb{E}\left[\frac{1}{n}\sum_{i=1}^{n}\left(\frac{T_i Y_i}{e(X_i)}\right)\right] = \frac{1}{n}\sum_{i=1}^{n}\mathbb{E}\left[\frac{T_i Y_i}{e(X_i)}\right] \qquad \text{(Linearity of expectation)}$$

$$= \mathbb{E}\left[\frac{TY^{(1)}}{e(X)}\right] \qquad \text{(Independence and consistency)}$$

$$= \mathbb{E}\left[\mathbb{E}\left[\frac{TY^{(1)}}{e(X)} \mid X\right]\right] \qquad \text{(Law of Total Probability)}$$

$$= \mathbb{E}\left[\frac{1}{e(X)}\mathbb{E}\left[TY^{(1)} \mid X\right]\right] \qquad (e(X) \text{ depends on } X)$$

$$= \mathbb{E}\left[\frac{1}{e(X)}\mathbb{E}\left[Y^{(1)} \mid X\right]\mathbb{E}\left[T \mid X\right]\right] \qquad \text{(No confounding assumption)}$$

$$= \mathbb{E}\left[\mathbb{E}\left[Y^{(1)} \mid X\right]\right] \qquad \text{(Definition of } e(X))$$

$$= \mathbb{E}[Y^{(1)}].$$

This shows that $\mathbb{E}\left[\psi_2(Z, \boldsymbol{\theta}_\infty)\right] = 0$. Similarly, one can show that $\mathbb{E}\left[\psi_3(Z, \boldsymbol{\theta}_\infty)\right] = 0$. Finally, we also have $\psi_4(Z, \boldsymbol{\theta}_\infty) = 0$. Therefore,

$$\mathbb{E}\left[\psi(Z, \boldsymbol{\theta}_\infty)\right] = 0. \tag{50}$$

We now show that $\boldsymbol{\theta}_\infty$ defined above is the unique value that satisfies 50. Let

$$L(\beta) = -\mathbb{E}\left[T \ln\big(e(X, \beta)\big) + \big(1 - T\big)\ln\big(1 - e(X, \beta)\big)\right].$$

A direct calculation shows that

$$\nabla_\beta L(\beta) = \mathbb{E}\left[\widetilde{X}\left(e(X, \beta) - T\right)\right] \quad \text{and} \quad \nabla_\beta^2 L(\beta) = \mathbb{E}\left[\widetilde{X}\,\widetilde{X}^\top\, e(X, \beta)\left(1 - e(X, \beta)\right)\right]$$

Since $\mathbb{E}[T \mid X] = e(X, \beta_\infty)$, so at $\beta = \beta_\infty$,

$$\nabla_\beta L(\beta_\infty) = \mathbb{E}\left[\widetilde{X}\left(e(X, \beta_\infty) - T\right)\right] = 0$$

making $\beta_\infty$ a stationary point. Furthermore, using overlap we have $e(X, \beta)\big(1 - e(X, \beta)\big) \geq \eta^2$ therefore $\forall v \in \mathbb{R}^{p+1}$:

$$v^\top \nabla_\beta^2 L(\beta) v = \mathbb{E}\left[||\widetilde{X}^\top v||_2^2\, e(X, \beta)\left(1 - e(X, \beta)\right)\right]$$

$$\geq \eta^2 \mathbb{E}\left[||\widetilde{X}^\top v||_2^2\right]$$

$$\geq \eta^2 v^\top \mathbb{E}\left[\widetilde{X}\,\widetilde{X}^\top\right] v.$$

Since we assumed that $\mathbb{E}\left[X X^\top\right]$ is positive definite, the Hessian $\nabla_\beta^2 L(\beta)$ is positive definite, so $L(\beta)$ is strictly convex. Hence there is a unique global minimizer of $L(\beta)$; since $\beta_\infty$ is a critical point, it must be that unique minimizer. Consequently, any solution to

$$\mathbb{E}\left[\widetilde{X}\left(e(X, \beta) - T\right)\right] = 0$$

must equal $\beta_\infty$. Since the second and third two components of $\psi$ are linear with respect to $\theta_0$ and $\theta_1$ and since the forth component is linear with respect to $\theta_2$, $\boldsymbol{\theta}_\infty$ is the only value satisfying (50).

We want to show that for every $\boldsymbol{\theta}$ in a neighborhood of $\boldsymbol{\theta}_\infty$, all the components of the second derivatives

$$\left| \frac{\partial^2}{\partial^2 \boldsymbol{\theta}} \psi_k(z, \boldsymbol{\theta}) \right|$$

are integrable for all $k \in \{1, 4\}$. Since $\boldsymbol{\theta} = (\boldsymbol{\beta}, \theta_0, \theta_1, \theta_2)$, we need to show that for $k \in \{1, 4\}$ and $i, j \in \{0, 2\}$ the following quantities are integrable

$$\left| \frac{\partial^2}{\partial \theta_i \partial \theta_j} \psi_k(z, \boldsymbol{\theta}) \right| \qquad \left| \frac{\partial^2}{\partial \theta_i \partial \boldsymbol{\beta}} \psi_k(z, \boldsymbol{\theta}) \right| \qquad \left| \frac{\partial^2}{\partial \boldsymbol{\beta} \partial \theta_i} \psi_k(z, \boldsymbol{\theta}) \right| \qquad \left| \frac{\partial^2}{\partial \boldsymbol{\beta} \partial \boldsymbol{\beta}} \psi_k(z, \boldsymbol{\theta}) \right|$$

One can note that the first three quantities are bounded by 1 and therefore integrable. Hence, it suffices to consider

$$\left| \frac{\partial^2}{\partial^2 \boldsymbol{\beta}} \psi_k(z, \boldsymbol{\theta}) \right|,$$

where $k \in \{1, 2, 3\}$, since $\psi_4$ does not depend on $\theta_1$. For $k = 1$, a direct calculation yields

$$\left| \frac{\partial^2}{\partial^2 \boldsymbol{\beta}} \psi_1(z, \boldsymbol{\theta}) \right| = \left| -\widetilde{X_k} \, \widetilde{X_l} \, \widetilde{X_m} \, e(X, \boldsymbol{\beta}) \big(1 - e(X, \boldsymbol{\beta})\big)\big(1 - 2e(X, \boldsymbol{\beta})\big) \right| \le \left| \widetilde{X_k} \, \widetilde{X_l} \, \widetilde{X_m} \right|.$$

By Cauchy–Schwarz or Hölder's inequality,

$$\mathbb{E}\big[ \big| \widetilde{X_k} \, \widetilde{X_l} \, \widetilde{X_m} \big| \big] \; \le \; \mathbb{E}\big[ (\widetilde{X_k})^2 \big]^{1/2} \mathbb{E}\big[ (\widetilde{X_l} \, \widetilde{X_m})^2 \big]^{1/2} \; \le \; \mathbb{E}\big[ (\widetilde{X_k})^2 \big]^{1/2} \mathbb{E}\big[ (\widetilde{X_l})^4 \big]^{1/4} \mathbb{E}\big[ (\widetilde{X_m})^4 \big]^{1/4}.$$

Since $\widetilde{X}$ is sub-Gaussian, it has finite moments of all orders, implying integrability of $\big| \widetilde{X_k} \, \widetilde{X_l} \, \widetilde{X_m} \big|$.

For $k = 2$ or $k = 3$, we similarly get

$$\left| \frac{\partial^2}{\partial^2 \boldsymbol{\beta}} \psi_2(z, \boldsymbol{\theta}) \right| \le \left| Y \, \exp(2 \, \widetilde{X}^\top \boldsymbol{\beta}) \, \widetilde{X} \, \widetilde{X}^\top \right|,$$

and

$$\left| \frac{\partial^2}{\partial^2 \boldsymbol{\theta}} \psi_3(z, \boldsymbol{\theta}) \right| \le \left| Y \, \exp(2 \, \widetilde{X}^\top \boldsymbol{\beta}) \, \widetilde{X} \, \widetilde{X}^\top \right|.$$

Hence, it remains to show that

$$\mathbb{E}\Big[ \big| Y \, \exp(2 \, \widetilde{X}^\top \boldsymbol{\beta}) \, \widetilde{X}_k \, \widetilde{X}_l \big| \Big]$$

is finite. By Cauchy–Schwarz,

$$\mathbb{E}\Big[ \big| Y \, \exp(2 \, \widetilde{X}^\top \boldsymbol{\beta}) \, \widetilde{X}_k \, \widetilde{X}_l \big| \Big] \; \le \; \sqrt{\mathbb{E}\big[ Y^2 \big]} \sqrt{\mathbb{E}\Big[ \exp\big(4 \, \widetilde{X}^\top \boldsymbol{\beta}\big) (\widetilde{X}_k \, \widetilde{X}_l)^2 \Big]}.$$

Since $\widetilde{X}$ is sub-Gaussian, its exponential moments are finite. Specifically, for some $\sigma > 0$,

$$\mathbb{E}\big[ \exp\big(\lambda \, \boldsymbol{v}^\top \widetilde{X}\big) \big] \le \exp\Big( \tfrac{\lambda^2 \|\boldsymbol{v}\|_2^2 \sigma^2}{2} \Big) \quad \forall \lambda \in \mathbb{R}, \; \boldsymbol{v} \in \mathbb{R}^d,$$

which guarantees $\mathbb{E}[\exp(8 \, \widetilde{X}^\top \boldsymbol{\beta})]$ is finite. Moreover, sub-Gaussian random variables have finite polynomial moments, so $\mathbb{E}[(\widetilde{X}_k \, \widetilde{X}_l)^4]$ is also finite. Therefore,

$$\big| Y \, \exp(2 \, \widetilde{X}^\top \boldsymbol{\beta}) \, \widetilde{X}_k \, \widetilde{X}_l \big|$$

is integrable.

Collecting these results, we conclude that every second derivative $\left| \frac{\partial^2}{\partial^2 \boldsymbol{\theta}} \psi_k(z, \boldsymbol{\theta}) \right|$ is integrable for all $k \in \{1, 4\}$ in a neighborhood of $\boldsymbol{\theta}_\infty$.

Define

$$A\left(\theta_\infty\right) = \mathbb{E}\left[\left.\frac{\partial\psi}{\partial\theta}\right|_{\theta=\theta_\infty}\right] \quad \text{and} \quad B(\theta_\infty) = \mathbb{E}\left[\psi(Z,\theta_\infty)\psi(Z,\theta_\infty)^T\right].$$

Next, we verify the conditions of Theorem 7.2 in Stefanski & Boos (2002). To do so, we compute $A\left(\theta_\infty\right)$ and $B\left(\theta_\infty\right)$. Since

$$\frac{\partial\psi}{\partial\theta}(Z,\theta) = \begin{pmatrix} -e(X,\boldsymbol{\beta})\left(1-e(X,\boldsymbol{\beta})\right)\widetilde{X}\widetilde{X}^\top & 0 & 0 & 0 \\ \frac{(1-T)Ye(X,\boldsymbol{\beta})}{1-e(X,\boldsymbol{\beta})}\widetilde{X}^\top & -1 & 0 & 0 \\ -\frac{TY(1-e(X,\boldsymbol{\beta}))}{e(X,\boldsymbol{\beta})}\widetilde{X}^\top & 0 & -1 & 0 \\ 0 & -\theta_2 & 1 & -\theta_0 \end{pmatrix}, \tag{51}$$

We obtain

$$A\left(\boldsymbol{\theta_\infty}\right) = \begin{pmatrix} -Q & 0 & 0 & 0 \\ c_{10}^\top & -1 & 0 & 0 \\ -c_{01}^\top & 0 & -1 & 0 \\ 0 & -\frac{\mathbb{E}\left[Y^{(1)}\right]}{\mathbb{E}\left[Y^{(0)}\right]} & 1 & -\mathbb{E}\left[Y^{(0)}\right] \end{pmatrix},$$

where:

- $Q = \mathbb{E}\left[e(X)(1-e(X))\widetilde{X}\widetilde{X}^\top\right]$,

- $c_{10} = \mathbb{E}[\widetilde{X}\,e(X)Y^{(0)}]$ and $c_{01} = \mathbb{E}[\widetilde{X}(1-e(X))Y^{(1)}]$,

which using Schur complement leads to:

$$A^{-1}\left(\boldsymbol{\theta_\infty}\right) = \begin{pmatrix} -Q^{-1} & 0 & 0 & 0 \\ -c_{10}^\top Q^{-1} & -1 & 0 & 0 \\ c_{01}^\top Q^{-1} & 0 & -1 & 0 \\ \left(\frac{\mathbb{E}[Y(1)]}{\mathbb{E}[Y(0)]^2}c_{10}^\top + \frac{1}{\mathbb{E}[Y^{(0)}]}c_{01}^\top\right)Q^{-1} & \frac{\mathbb{E}[Y(1)]}{\mathbb{E}[Y(0)]^2} & \frac{-1}{\mathbb{E}[Y^{(0)}]} & \frac{-1}{\mathbb{E}[Y^{(0)}]} \end{pmatrix}.$$

Regarding $B\left(\theta_\infty\right)$, elementary calculations show that

$$B(\theta_\infty) = \begin{pmatrix} Q & -c_{10} & c_{01} & 0 \\ -c_{10}^\top & \text{Var}\left(\frac{(1-T)Y}{1-e(X)}\right) & -\mathbb{E}\left[Y^{(1)}\right]\mathbb{E}\left[Y^{(0)}\right] & 0 \\ c_{01}^\top & -\mathbb{E}\left[Y^{(1)}\right]\mathbb{E}\left[Y^{(0)}\right] & \text{Var}\left(\frac{TY}{e(X)}\right) & 0 \\ 0 & 0 & 0 & 0 \end{pmatrix},$$

Based on the previous calculations, we have

- $\psi(z,\boldsymbol{\theta})$ and its first two partial derivatives with respect to $\boldsymbol{\theta}$ exist for all $z$ and for all $\boldsymbol{\theta}$ in the neighborhood of $\boldsymbol{\theta_\infty}$.

- For each $\boldsymbol{\theta}$ in the neighborhood of $\boldsymbol{\theta_\infty}$, we have for all $k \in \{1,4\}\left|\frac{\partial^2}{\partial^2\boldsymbol{\theta}}\psi_k(z,\boldsymbol{\theta})\right|$ is integrable.

- $A(\theta_\infty)$ exists and is nonsingular.

- $B(\theta_\infty)$ exists and is finite.

Since we have:

$$\sum_{i=1}^{n} \psi(T_i, Y_i, \hat{\boldsymbol{\theta}}_n) = 0 \quad \text{and} \quad \hat{\boldsymbol{\theta}}_n \xrightarrow{p} \theta_\infty.$$

Then the conditions of Theorem 7.2 in Stefanski & Boos (2002) are satisfied, we have

$$\sqrt{n}\left(\hat{\boldsymbol{\theta}}_n - \theta_\infty\right) \xrightarrow{d} \mathcal{N}\left(0, A(\theta_\infty)^{-1}B(\theta_\infty)(A(\theta_\infty)^{-1})^\top\right),$$

Since we are only interested in the bottom right term of the sandwich term , we only need to compute $u_{d+3}^T A(\theta_\infty)^{-1}B(\theta_\infty)(A(\theta_\infty)^{-1})^\top u_{d+3}$ where $u_{d+3}$ is the last vector canonical basis vector of $\mathbb{R}^{d+3}$. Hence,

$$\begin{aligned}\left[A(\theta_\infty)^{-1}B(\theta_\infty)(A(\theta_\infty)^{-1})^\top\right]_{d+3,d+3} &= u_{d+3}^T A(\theta_\infty)^{-1}B(\theta_\infty)(A(\theta_\infty)^{-1})^\top u_{d+3} \\ &= u_{d+3}^T A(\theta_\infty)^{-1}B(\theta_\infty)(u_{d+3}^T A(\theta_\infty)^{-1})^\top\end{aligned}$$

Noting that $u_{d+3}^\top A(\theta_\infty)^{-1} = \left(\left(\frac{\mathbb{E}[Y(1)]}{\mathbb{E}[Y(0)]^2}c_{10}^\top + \frac{1}{\mathbb{E}[Y^{(0)}]}c_{01}^\top\right)Q^{-1}, \frac{\mathbb{E}[Y(1)]}{\mathbb{E}[Y(0)]^2}, \frac{-1}{\mathbb{E}[Y(0)]}, \frac{-1}{\mathbb{E}[Y(0)]}\right)$ where we used that $(Q^{-1})^\top = Q^{-1}$ since $Q$ is symmetric. We defining

$$u_{d+3}^\top A(\theta_\infty)^{-1}B(\theta_\infty) := W = (W_1, W_2, W_3, W_4)$$

where:

$$W_1 = \left[\frac{\mathbb{E}[Y(1)]}{\mathbb{E}[Y(0)]^2}c_{10}^\top + \frac{1}{\mathbb{E}[Y(0)]}c_{01}^\top\right]\underbrace{Q^{-1}Q}_{I_d} - c_{10}^\top\frac{\mathbb{E}[Y(1)]}{\mathbb{E}[Y(0)]^2} - c_{01}^\top\frac{1}{\mathbb{E}[Y(0)]} = 0,$$

$$W_2 = -\left[\frac{\mathbb{E}[Y(1)]}{\mathbb{E}[Y(0)]^2}c_{10}^\top + \frac{1}{\mathbb{E}[Y(0)]}c_{01}^\top\right]Q^{-1}c_{10} + \text{Var}\left(\frac{(1-T)Y}{1-e(X)}\right)\frac{\mathbb{E}[Y(1)]}{\mathbb{E}[Y(0)]^2} + \mathbb{E}\left[Y^{(1)}\right],$$

$$W_3 = \left[\frac{\mathbb{E}[Y(1)]}{\mathbb{E}[Y(0)]^2}c_{10}^\top + \frac{1}{\mathbb{E}[Y(0)]}c_{01}^\top\right]Q^{-1}c_{01} - \frac{\mathbb{E}[Y(1)]^2}{\mathbb{E}[Y(0)]} - \frac{\text{Var}\left(\frac{TY}{e(X)}\right)}{\mathbb{E}[Y(0)]},$$

and $W_4 = 0$. Finally, gathering all the terms we have:

$$\begin{aligned}\left[A(\theta_\infty)^{-1}B(\theta_\infty)(A(\theta_\infty)^{-1})^\top\right]_{d+3,d+3} &= W_2\frac{\mathbb{E}[Y(1)]}{\mathbb{E}[Y(0)]^2} - W_3\frac{1}{\mathbb{E}[Y(0)]} \\ &= -\left[\frac{\mathbb{E}[Y(1)]}{\mathbb{E}[Y(0)]^2}c_{10}^\top + \frac{1}{\mathbb{E}[Y(0)]}c_{01}^\top\right]Q^{-1}c_{10}\frac{\mathbb{E}[Y(1)]}{\mathbb{E}[Y(0)]^2} \\ &\quad - \left[\frac{\mathbb{E}[Y(1)]}{\mathbb{E}[Y(0)]^2}c_{10}^\top + \frac{1}{\mathbb{E}[Y(0)]}c_{01}^\top\right]Q^{-1}c_{01}\frac{1}{\mathbb{E}[Y(0)]} \\ &\quad + \text{Var}\left(\frac{(1-T)Y}{1-e(X)}\right)\left(\frac{\mathbb{E}[Y(1)]}{\mathbb{E}[Y(0)]^2}\right)^2 + \frac{\text{Var}\left(\frac{TY}{e(X)}\right)}{(\mathbb{E}[Y(0)])^2} \\ &\quad + 2\frac{\mathbb{E}[Y(1)]^2}{\mathbb{E}[Y(0)]^2}.\end{aligned}$$

One can note that:

$$\mathrm{Var}\left(\frac{(1-T)Y}{1-e(X)}\right)\left(\frac{\mathbb{E}[Y(1)]}{\mathbb{E}[Y(0)]^2}\right)^2 + \frac{\mathrm{Var}\left(\frac{TY}{e(X)}\right)}{(\mathbb{E}[Y(0)])^2} + 2\frac{\mathbb{E}[Y(1)]^2}{\mathbb{E}[Y(0)]^2} = \underbrace{\left(\frac{\mathbb{E}[Y(1)]}{\mathbb{E}[Y(0)]}\right)^2}_{\tau_{\mathrm{RR}}^2}\left(\frac{\mathrm{Var}\left(\frac{(1-T)Y}{1-e(X)}\right)}{\mathbb{E}[Y(0)]^2} + \frac{\mathrm{Var}\left(\frac{TY}{e(X)}\right)}{\mathbb{E}[Y(1)]^2} + 2\right)$$

$$= \tau_{\mathrm{RR}}^2\left(\frac{\mathbb{E}\left[\frac{(Y^{(1)})^2}{e(X)}\right]}{\mathbb{E}\left[Y^{(1)}\right]^2} + \frac{\mathbb{E}\left[\frac{(Y^{(0)})^2}{1-e(X)}\right]}{\mathbb{E}\left[Y^{(0)}\right]^2}\right)$$

where for the last equality we used that:

$$\mathrm{Var}\left(\frac{TY}{e(X)}\right) = \mathbb{E}\left[\frac{(Y^{(1)})^2}{e(X)}\right] - \mathbb{E}\left[Y^{(1)}\right]^2 \quad \text{and} \quad \mathrm{Var}\left(\frac{(1-T)Y}{1-e(X)}\right) = \mathbb{E}\left[\frac{(Y^{(0)})^2}{1-e(X)}\right] - \mathbb{E}\left[Y^{(0)}\right]^2$$

Finally using calculation we did previously we get that:

$$V_{\mathrm{RR\text{-}MLE}} = \underbrace{\tau_{\mathrm{RR}}^2\left(\frac{\mathbb{E}\left[\frac{(Y^{(1)})^2}{e(X)}\right]}{\mathbb{E}\left[Y^{(1)}\right]^2} + \frac{\mathbb{E}\left[\frac{(Y^{(0)})^2}{1-e(X)}\right]}{\mathbb{E}\left[Y^{(0)}\right]^2}\right)}_{V_{\mathrm{RR\text{-}IPW}}} - \tau_{\mathrm{RR}}^2\left\|\frac{c_{10}}{\mathbb{E}[Y(0)]} + \frac{c_{01}}{\mathbb{E}[Y(1)]}\right\|_{Q^{-1}}^2$$

□

### 7.3.3. RISK RATIO G FORMULA ESTIMATOR

*Proof of Proposition 3.7.*
**Asymptotic bias and variance of the oracle risk ratio G formula estimator** Recall that the oracle risk ratio G formula is defined as

$$\tau_{\mathrm{RR,G,n}}^{\star} = \frac{\sum_{i=1}^{n}\mu_{(1)}(X_i)}{\sum_{i=1}^{n}\mu_{(0)}(X_i)},$$

where the response surfaces $\mu_{(0)}$ and $\mu_{(1)}$ are assumed to be known. Let us define $g_1(Z) = \mu_{(1)}(X_i)$ and $g_0(Z) = \mu_{(0)}(X_i)$ with $Z = X$. Since $g_1(Z)$ and $g_0(Z)$ are bounded, they are square integrable. We also have that $\mathbb{E}\left[g_0(Z_i)\right] = \mathbb{E}\left[Y^{(0)}\right]$ and $\mathbb{E}\left[g_1(Z_i)\right] = \mathbb{E}\left[Y^{(1)}\right]$. We can therefore apply Theorem 7.3 and conclude that

$$\sqrt{n}(\tau_{\mathrm{RR,G,n}}^{\star} - \tau_{\mathrm{RR}}) \to \mathcal{N}(0, V_{\mathrm{RR,G}}),$$

where $V_{\mathrm{RR,G}} = \tau_{\mathrm{RR}}^2\,\mathrm{Var}\left(\frac{\mu_1^{\star}(X)}{\mathbb{E}\left[Y^{(1)}\right]} - \frac{\mu_0^{\star}(X)}{\mathbb{E}\left[Y^{(0)}\right]}\right)$. As a consequence an estimator $\hat{V}_{RR,G}$ can be derived:

$$\hat{V}_{RR,G} = \frac{\hat{\tau}_{\mathrm{RR,G,n}}^2}{n}\sum_{i=1}^{n}\left(\frac{\hat{\mu}_1(X_i)}{\frac{1}{n}\sum_{T_i=1}Y_i} - \frac{\hat{\mu}_0(X_i)}{\frac{1}{n}\sum_{T_i=0}Y_i} - \frac{1}{n}\sum_{i=1}^{n}\frac{\hat{\mu}_1(X_i)}{\frac{1}{n}\sum_{T_i=1}Y_i} - \frac{\hat{\mu}_0(X_i)}{\frac{1}{n}\sum_{T_i=0}Y_i}\right)^2 \tag{52}$$

**Finite sample bias and variance of the oracle ratio G formula estimator** Let $T_1(\boldsymbol{Z}) = \frac{1}{n}\sum_{i=1}^{n}\mu_{(1)}(X_i)$ and $T_0(\boldsymbol{Z}) =$

$\frac{1}{n}\sum_{i=1}^{n}\mu_{(0)}(X_i)$ where $\boldsymbol{Z} = (X_1, \ldots, X_n)$. We first show that $\mathrm{Var}(T_1(\boldsymbol{Z})) = O_p\left(\frac{1}{n}\right)$ and $\mathrm{Var}(T_0(\boldsymbol{Z})) = O_p\left(\frac{1}{n}\right)$:

$$
\begin{aligned}
\mathrm{Var}(T_1(\boldsymbol{Z})) &= \frac{1}{n^2}\,\mathrm{Var}\left(\sum_{i=1}^{n}\mu_{(1)}(X_i)\right) \\
&= \frac{1}{n^2}\sum_{i=1}^{n}\mathrm{Var}(\mu_{(1)}(X_i)) && \text{(by i.i.d.)} \\
&= \frac{1}{n}\left(\mathbb{E}\left[(\mu_{(1)}(X_i))^2\right] - \mathbb{E}\left[Y^{(1)}\right]^2\right) && \text{(by law of total expectation)} \\
&\leq \frac{M^2 - \mathbb{E}\left[Y^{(1)}\right]^2}{n} && (\mu_{(1)}(X_i)) \leq M) \\
&= O_p\left(\frac{1}{n}\right).
\end{aligned}
$$

Similarly, $\mathrm{Var}(T_0(\boldsymbol{Z})) = O_p(1/n)$. Since we also have that

$$
\mathbb{E}[T_1(\boldsymbol{Z})] = \mathbb{E}\left[Y^{(1)}\right] \quad \mathbb{E}[T_0(\boldsymbol{Z})] = \mathbb{E}\left[Y^{(0)}\right]
$$

Therefore, we showed that $T_1(\boldsymbol{Z})$ and $T_0(\boldsymbol{Z})$ are unbiased estimators of $\mathbb{E}\left[Y^{(1)}\right]$ and $\mathbb{E}\left[Y^{(0)}\right] > 0$ such that $\mathrm{Var}(T_1(\boldsymbol{Z})) = O_p(1/n)$ and $\mathrm{Var}(T_0(\boldsymbol{Z})) = O_p(1/n)$. We also have that $T_0(\boldsymbol{Z})$ and $T_1(\boldsymbol{Z})$ are bounded:

$$
m_0 \leq T_0(\boldsymbol{Z}) \leq M_0 \quad \text{and} \quad T_1(\boldsymbol{Z}) \leq M_1
$$

Applying Theorem 7.3, under Assumption 7.2 we obtain:

$$
|\mathbb{E}[\hat{\tau}_{\mathrm{RR,\,HT,\,n}}] - \tau_{\mathrm{RR}}| \leq \frac{2M_1M_0^2}{nm_0^3} \quad \text{and} \quad |\mathrm{Var}(\hat{\tau}_{\mathrm{RR,\,HT,\,n}}) - V_{\mathrm{RR,\,HT}}| \leq \frac{2M_0^2M_1(M_1+M_0)}{m_0^6}
$$

$\square$

### 7.3.4. RISK RATIO G-FORMULA IN LINEAR MODELS

**Lemma 7.6** (see, e.g. (Seber & Lee, 2012)). *Grant Assumption 3.8 linear model. Let $\gamma_{(t)} = (c_{(t)}, \beta_{(t)}) \in \mathbb{R}^{d+1}$ and $Z = (1, X)$. We rearrange the $Y_i$ and $Z_i$ so that the first $n_1$ observations correspond to $T = 1$. We then define $\mathbf{Y}_1 = (Y_1, \ldots, Y_{n_1})^\top$ and $\mathbf{Y}_0 = (Y_{n_1+1}, \ldots, Y_n)^\top$, as well as $\mathbf{Z}_1 = (Z_1, \ldots, Z_{n_1})^\top$ and $\mathbf{Z}_0 = (Z_{n_1+1}, \ldots, Z_n)^\top$. Then for $t \in \{0, 1\}$, the linear model can be formulated as:*

$$
Y^{(t)} = Z^\top\gamma_{(t)} + \varepsilon_{(t)}, \quad \mathbb{E}[\varepsilon_{(t)}|Z] = 0, \quad Var[\varepsilon_{(t)}|Z] = \sigma^2,
$$

*and the least square estimator is given as*

$$
\hat{\gamma}_{(t)} = \left(\frac{1}{n_t}\mathbf{Z}_t^\top\mathbf{Z}_t\right)^{-1}\frac{1}{n_t}\mathbf{Z}_t^\top\mathbf{Y}_t
$$

**Proposition 7.7.** *Grant Assumption 3.8. Let $\hat{e} = \left(\sum_{i=1}^{n}T_i\right)/n$ and for all $t \in \{0, 1\}$,*

$$
\bar{Z}_{(t)} = \frac{1}{\sum_{i=1}^{n}\mathbb{1}_{T_i=t}}\sum_{i=1}^{n}\mathbb{1}_{T_i=t}Z_i. \tag{53}
$$

*Defining $\nu_t = \mathbb{E}[X|T = t]$ and $\Sigma_t = Var(X|T = t)$, we have*

$$
\sqrt{n}(\hat{\boldsymbol{\theta}}_n - \boldsymbol{\theta}_\infty) \xrightarrow{d} \mathcal{N}(0, \Sigma),
$$

*where*

$$\boldsymbol{\theta}_n = \begin{pmatrix} \bar{Z}_{(0)} \\ \bar{Z}_{(1)} \\ \hat{\gamma}_{(0)} \\ \hat{\gamma}_{(1)} \\ \hat{e} \end{pmatrix}, \quad \boldsymbol{\theta}_\infty = \begin{pmatrix} E[Z|T=0] \\ E[Z|T=1] \\ \gamma_{(0)} \\ \gamma_{(1)} \\ e \end{pmatrix}, \quad \Sigma = \begin{pmatrix} \frac{\mathrm{Var}[Z|T=0]}{(1-e)} & 0 & 0 & 0 & 0 \\ 0 & \frac{\mathrm{Var}[Z|T=1]}{e} & 0 & 0 & 0 \\ 0 & 0 & \frac{\sigma^2 Q_0^{-1}}{1-e} & 0 & 0 \\ 0 & 0 & 0 & \frac{\sigma^2 Q_1^{-1}}{e} & 0 \\ 0 & 0 & 0 & 0 & e(1-e) \end{pmatrix},$$

*with* $Q_t^{-1} = \begin{bmatrix} 1 + \nu_t^T \Sigma_t^{-1} \nu_t & -\nu_t^T \Sigma_t^{-1} \\ -\Sigma_t^{-1} \nu_t & \Sigma_t^{-1} \end{bmatrix}.$

*Proof.* Using M-estimation theory to prove asymptotic normality of the $\theta_n$, we first define the following:

$$\psi(T, Z, \boldsymbol{\theta}) = \begin{pmatrix} \psi_0(T, Z, \boldsymbol{\theta}) \\ \psi_1(T, Z, \boldsymbol{\theta}) \\ \psi_2(T, Z, \boldsymbol{\theta}) \\ \psi_3(T, Z, \boldsymbol{\theta}) \\ \psi_4(T, Z, \boldsymbol{\theta}) \end{pmatrix} := \begin{pmatrix} (1-T)(Z - \theta_0) \\ T(Z - \theta_1) \\ (1-T)\left(Z\epsilon(0) - ZZ^\top \left(\theta_2 - \gamma_{(0)}\right)\right) \\ T\left(Z\epsilon(1) - ZZ^\top \left(\theta_3 - \gamma(1)\right)\right) \\ T - \theta_4 \end{pmatrix}$$

where $\boldsymbol{\theta} = (\theta_0, \theta_1, \theta_2, \theta_3, \theta_4)$. We still have that $\hat{\boldsymbol{\theta}}_n = (\bar{Z}_{(0)}, \bar{Z}_{(1)}, \hat{\gamma}_{(0)}, \hat{\gamma}_{(1)}, \hat{e})$ is an M-estimator of type $\psi$ (see Stefanski & Boos, 2002) since

$$\sum_{i=1}^n \psi(T_i, Z_i, \hat{\boldsymbol{\theta}}_n) = 0.$$

We now demonstrate that $\mathbb{E}\left[\psi(T, Y, \boldsymbol{\theta}_\infty)\right] = 0$. We directly have that $\mathbb{E}\left[\psi_4(T, Y, \boldsymbol{\theta}_\infty)\right] = 0$. For the other terms we have:

$$\begin{aligned} \mathbb{E}\left[\psi_1(T, Z, \boldsymbol{\theta}_\infty)\right] &= \mathbb{E}\left[T\left(Z - \mathbb{E}[Z|T=1]\right)\right] \\ &= \mathbb{E}\left[\mathbb{E}\left[T\left(Z - \mathbb{E}[Z|T=1]\right)|T\right]\right] \\ &= \mathbb{E}\left[T\left(\mathbb{E}\left[Z|T\right] - \mathbb{E}[Z|T=1]\right)\right] \\ &= \mathbb{E}\left[T\left(\mathbb{E}\left[Z|T\right] - \mathbb{E}[Z|T=1]\right)\right] \\ &= \mathbb{P}\left[T=1\right]\left(\mathbb{E}\left[Z|T=1\right] - \mathbb{E}[Z|T=1]\right) \\ &= 0 \end{aligned}$$

We also have that:

$$\begin{aligned} \mathbb{E}\left[\psi_3(T, Z, \boldsymbol{\theta}_\infty)\right] &= \mathbb{E}\left[TZ\epsilon_{(1)}\right] \\ &= \mathbb{E}\left[Z\mathbb{E}\left[T\epsilon_{(1)}|Z\right]\right] \\ &= \mathbb{E}\left[Z\mathbb{E}\left[\epsilon_{(1)}|Z, T=1\right]\right] \\ &= 0. \end{aligned}$$

Similarly, we can show:

$$\mathbb{E}\left[\psi_0(T, Z, \boldsymbol{\theta}_\infty)\right] = 0 \quad \text{and} \quad \mathbb{E}\left[\psi_2(T, Z, \boldsymbol{\theta}_\infty)\right] = 0.$$

At this point, we note that since $\psi(T, Z, \boldsymbol{\theta})$ is a linear function of $\boldsymbol{\theta}$, $\theta_\infty$ is the only value of $\boldsymbol{\theta}$ such that $\mathbb{E}\left[\psi(T, Z, \boldsymbol{\theta})\right] = 0$
We proceed by defining:

$$A\left(\theta_\infty\right) = \mathbb{E}\left[\left.\frac{\partial \psi}{\partial \theta}\right|_{\theta=\theta_\infty}\right] \quad \text{and} \quad B(\theta_\infty) = \mathbb{E}\left[\psi(T, Z, \theta_\infty)\psi(T, Z, \theta_\infty)^T\right].$$

Next, we check the conditions of Theorem 7.2 in Stefanski & Boos (2002). First, we compute $A\left(\theta_\infty\right)$ and $B\left(\theta_\infty\right)$. Since:

$$\frac{\partial \psi}{\partial \theta}(T, Z, \theta) = \begin{pmatrix} -(1-T) & 0 & 0 & 0 & 0 \\ 0 & -T & 0 & 0 & 0 \\ 0 & 0 & -(1-T)ZZ^\top & 0 & 0 \\ 0 & 0 & 0 & -TZZ^\top & 0 \\ 0 & 0 & 0 & 0 & -1 \end{pmatrix},$$

we obtain:

$$A\left(\boldsymbol{\theta}_{\infty}\right) = \begin{pmatrix} -(1-e) & 0 & 0 & 0 & 0 \\ 0 & -e & 0 & 0 & 0 \\ 0 & 0 & -(1-e)Q_0 & 0 & 0 \\ 0 & 0 & 0 & -eQ_1 & 0 \\ 0 & 0 & 0 & 0 & -1 \end{pmatrix}, \quad \text{where} \quad Q_t = \mathbb{E}\left[ZZ^{\top}|T=t\right].$$

which leads to:

$$A^{-1}\left(\boldsymbol{\theta}_{\infty}\right) = \begin{pmatrix} -\frac{1}{1-e} & 0 & 0 & 0 & 0 \\ 0 & -\frac{1}{e} & 0 & 0 & 0 \\ 0 & 0 & -\frac{Q_0^{-1}}{1-e} & 0 & 0 \\ 0 & 0 & 0 & -\frac{Q_1^{-1}}{e} & 0 \\ 0 & 0 & 0 & 0 & -1 \end{pmatrix}.$$

Regarding $B(\boldsymbol{\theta}_{\infty})$, since we have $T(1-T) = 0$, elementary calculations show that:

$$\begin{array}{ll} B(\boldsymbol{\theta}_{\infty})_{1,2} = B(\boldsymbol{\theta}_{\infty})_{2,1} = 0 & \\ B(\boldsymbol{\theta}_{\infty})_{3,4} = B(\boldsymbol{\theta}_{\infty})_{4,3} = 0 & \text{and} \end{array} \quad \begin{array}{l} B(\boldsymbol{\theta}_{\infty})_{1,4} = B(\boldsymbol{\theta}_{\infty})_{4,1} = 0 \\ B(\boldsymbol{\theta}_{\infty})_{2,3} = B(\boldsymbol{\theta}_{\infty})_{3,2} = 0. \end{array}$$

Besides

$$\begin{aligned} B(\boldsymbol{\theta}_{\infty})_{2,2} &= E\left[T^2(Z - E\left[Z|T=1\right])(Z - E\left[Z|T=1\right])^{\top}\right] \\ &= E\left[T(Z - E\left[Z|T=1\right])(Z - E\left[Z|T=1\right])^{\top}\right] \\ &= E\left[TE\left[(Z - E\left[Z|T=1\right])(Z - E\left[Z|T=1\right])^{\top}|T\right]\right] \\ &= \mathbb{P}\left[T=1\right]E\left[(Z - E\left[Z|T=1\right])(Z - E\left[Z|T=1\right])^{\top}|T=1\right] \\ &= e\,\text{Var}\left[Z|T=1\right], \end{aligned}$$

and similarly,

$$B(\boldsymbol{\theta}_{\infty})_{1,1} = (1-e)\,\text{Var}\left[Z|T=0\right].$$

We can also note that:

$$\begin{aligned} B(\boldsymbol{\theta}_{\infty})_{4,4} &= E\left[T^2 ZZ^{\top}\epsilon_{(1)}^2\right] = E\left[TZZ^{\top}\epsilon_{(1)}^2\right] \\ &= E\left[TE\left[ZZ^{\top}\epsilon_{(1)}^2|T\right]\right] \\ &= \mathbb{P}\left[T=1\right]E\left[ZZ^{\top}\epsilon_{(1)}^2|T=1\right] \\ &= eE\left[ZZ^{\top}E\left[\epsilon_{(1)}^2|T=1,Z\right]|T=1\right] \\ &= e\sigma^2 E\left[ZZ^{\top}|T=1\right] := e\sigma^2 Q_1, \end{aligned}$$

and similarly,

$$B(\boldsymbol{\theta}_{\infty})_{3,3} = (1-e)\sigma^2 Q_0.$$

Finally,

$$\begin{aligned} B(\boldsymbol{\theta}_{\infty})_{2,4} = B(\boldsymbol{\theta}_{\infty})_{4,2} &= E\left[T^2(Z - E\left[Z|T=1\right])Z^{\top}\epsilon_{(1)}\right] \\ &= E\left[T(Z - E\left[Z|T=1\right])Z^{\top}\epsilon_{(1)}\right] \\ &= \mathbb{P}\left[T=1\right]E\left[(Z - E\left[Z|T=1\right])Z^{\top}\epsilon_{(1)}|T=1\right] \\ &= eE\left[(Z - E\left[Z|T=1\right])Z^{\top}E\left[\epsilon_{(1)}|T=1,Z\right]|T=1\right] \\ &= 0, \end{aligned}$$

and similarly,

$$B(\boldsymbol{\theta}_\infty)_{1,3} = B(\boldsymbol{\theta}_\infty)_{3,1} = 0.$$

We also have that:

$$
\begin{aligned}
B(\boldsymbol{\theta}_\infty)_{2,5} = B(\boldsymbol{\theta}_\infty)_{5,2} &= E\left[T(Z - E\left[Z|T=1\right])(T-e)\right] \\
&= E\left[T^2 Z - T^2 E\left[Z|T=1\right] - eTZ + eTE\left[Z|T=1\right]\right] \\
&= E\left[T^2 Z - T^2 E\left[Z|T=1\right] - eTZ + eTE\left[Z|T=1\right]\right] \\
&= eE\left[Z|T=1\right] - eE\left[Z|T=1\right] - e^2 E\left[Z|T=1\right] + e^2 E\left[Z|T=1\right] \\
&= 0
\end{aligned}
$$

and similarly,

$$B(\boldsymbol{\theta}_\infty)_{1,5} = B(\boldsymbol{\theta}_\infty)_{5,1} = 0.$$

We also have that :

$$
\begin{aligned}
B(\boldsymbol{\theta}_\infty)_{4,5} = B(\boldsymbol{\theta}_\infty)_{5,4} &= E\left[(T-e)TZ\epsilon_{(0)}\right] \\
&= E\left[(TZ\epsilon_{(0)}\right] - eE\left[(TZ\epsilon_{(0)}\right] \\
&= (1-e)E\left[TZ\epsilon_{(0)}\right] \\
&= (1-e)\mathbb{E}\left[Z\mathbb{E}\left[T\epsilon_{(0)}|Z\right]\right] \\
&= (1-e)\mathbb{E}\left[Z\mathbb{E}\left[\epsilon_{(0)}|Z,T=1\right]\right] \\
&= 0
\end{aligned}
$$

and similarly,

$$B(\boldsymbol{\theta}_\infty)_{3,5} = B(\boldsymbol{\theta}_\infty)_{5,3} = 0.$$

Gathering all calculations, and since $B(\boldsymbol{\theta}_\infty)_{5,5} = e(1-e)$, we have

$$
B(\boldsymbol{\theta}_\infty) = \begin{pmatrix}
(1-e)\operatorname{Var}\left[Z|T=0\right] & 0 & 0 & 0 & 0 \\
0 & e\operatorname{Var}\left[Z|T=1\right] & 0 & 0 & 0 \\
0 & 0 & (1-e)\sigma^2 Q_0 & 0 & 0 \\
0 & 0 & 0 & e\sigma^2 Q_1 & 0 \\
0 & 0 & 0 & 0 & e(1-e)
\end{pmatrix},
$$

Based on the previous calculations, we have:

- $\psi(z,\boldsymbol{\theta})$ and its first two partial derivatives with respect to $\boldsymbol{\theta}$ exist for all $z$ and for all $\boldsymbol{\theta}$ in the neighborhood of $\boldsymbol{\theta}_\infty$.

- For each $\boldsymbol{\theta}$ in the neighborhood of $\boldsymbol{\theta}_\infty$, we have for all $i,j,k \in \{0,2\}$:

$$\left|\frac{\partial^2}{\partial\theta_i \partial\theta_j}\psi_k(z,\boldsymbol{\theta})\right| \le 1$$

  and 1 is integrable.

- $A(\theta_\infty)$ exists and is nonsingular.

- $B(\theta_\infty)$ exists and is finite.

Since we have:

$$\sum_{i=1}^n \psi(T_i, Z_i, \hat{\boldsymbol{\theta}}_n) = 0 \quad \text{and} \quad \hat{\boldsymbol{\theta}}_n \xrightarrow{p} \theta_\infty.$$

Then, the conditions of Theorem 7.2 in Stefanski & Boos (2002) are satisfied, we have:

$$\sqrt{n}\left(\hat{\boldsymbol{\theta}}_n - \theta_\infty\right) \xrightarrow{d} \mathcal{N}\left(0, A(\theta_\infty)^{-1}B(\theta_\infty)(A(\theta_\infty)^{-1})^\top\right),$$

where:

$$A(\theta_\infty)^{-1}B(\theta_\infty)(A(\theta_\infty)^{-1})^\top = \begin{pmatrix} \frac{\mathrm{Var}[Z|T=0]}{(1-e)} & 0 & 0 & 0 & 0 \\ 0 & \frac{\mathrm{Var}[Z|T=1]}{e} & 0 & 0 & 0 \\ 0 & 0 & \frac{\sigma^2 Q_0^{-1}}{1-e} & 0 & 0 \\ 0 & 0 & 0 & \frac{\sigma^2 Q_1^{-1}}{e} & 0 \\ 0 & 0 & 0 & 0 & e(1-e) \end{pmatrix},$$

$\square$

**Proposition 7.8** (asymptotical normality of $\hat{\tau}_{\mathrm{RR,OLS}}$). *Assume we have linear model then we have:*

$$\sqrt{n}(\hat{\tau}_{RR,OLS} - \tau_{RR}) \xrightarrow{d} \mathcal{N}(0, V_{RR\text{-}OLS})$$

*with*

$$\frac{V_{RR\text{-}OLS}}{\tau_{RR}^2} = \left\| \frac{\beta_{(1)}}{\mathbb{E}\left[Y^{(1)}\right]} - \frac{\beta_{(0)}}{\mathbb{E}\left[Y^{(0)}\right]} \right\|_\Sigma^2 + \sigma^2 \left( \frac{1 + (1-e)^2 \|\nu_1 - \nu_0\|_{\Sigma_1^{-1}}^2}{e\mathbb{E}\left[Y^{(1)}\right]^2} + \frac{1 + e^2 \|\nu_1 - \nu_0\|_{\Sigma_0^{-1}}^2}{(1-e)\mathbb{E}\left[Y^{(0)}\right]^2} \right).$$

*Proof.* Let $\hat{\beta}_{(1)}$ and $\hat{c}_{(1)}$ be the parameters obtained via fitting an ordinary least square method on the treated individuals only, that is

$$(\hat{\beta}_{(1)}, \hat{c}_{(1)}) \in \arg\min_{c_{(1)}, \beta_{(1)}} \sum_{i=1}^n (Y_i^{(1)} - c_{(1)} - \beta_{(1)} X_i)^2 \mathbb{1}_{T_i=1}. \tag{54}$$

Similarly, let $\hat{\beta}_{(0)}$ and $\hat{c}_{(0)}$ be the parameters obtained via fitting an ordinary least square method on the control individuals only, that is

$$(\hat{\beta}_{(0)}, \hat{c}_{(0)}) \in \arg\min_{c_{(0)}, \beta_{(0)}} \sum_{i=1}^n (Y_i^{(0)} - c_{(0)} - \beta_{(0)} X_i)^2 \mathbb{1}_{T_i=0}. \tag{55}$$

An estimator of the RR using the G-formula approach is thus given by

$$\hat{\tau}_{\mathrm{RR,OLS}} = \frac{\sum_{i=1}^n \left( \hat{c}_{(1)} + X_i^\top \hat{\beta}_{(1)} \right)}{\sum_{i=1}^n \left( \hat{c}_{(0)} + X_i^\top \hat{\beta}_{(0)} \right)} \tag{56}$$

$$= \frac{\hat{c}_{(1)} + \bar{X}^\top \hat{\beta}_{(1)}}{\hat{c}_{(0)} + \bar{X}^\top \hat{\beta}_{(0)}}. \tag{57}$$

Besides, note that assuming a linear model implies that

$$\hat{\tau}_{\mathrm{RR,OLS}} = \frac{c_{(1)} + \mathbb{E}[X]^\top \beta_{(1)}}{c_{(0)} + \mathbb{E}[X]^\top \beta_{(0)}}. \tag{58}$$

Let, for all $i$, $Z_i = (1, X_i)$ and $\gamma_{(j)} = (c_{(j)}, \beta_{(j)})$ for all $j \in \{0, 1\}$. Expanding the following difference, we have:

$$\sqrt{n}(\tau_{\text{RR,OLS}} - \tau_{\text{RR}}) = \sqrt{n} \left( \frac{\hat{c}_{(1)} + \bar{X}^\top \hat{\beta}_{(1)}}{\hat{c}_{(0)} + \bar{X}^\top \hat{\beta}_{(0)}} - \frac{c_{(1)} + \mathbb{E}[X]^\top \beta_{(1)}}{c_{(0)} + \mathbb{E}[X]^\top \beta_{(0)}} \right) \tag{59}$$

$$= \sqrt{n} \left( \hat{c}_{(1)} + \bar{X}^\top \hat{\beta}_{(1)} \right) \left( \frac{1}{\hat{c}_{(0)} + \bar{X}^\top \hat{\beta}_{(0)}} - \frac{1}{c_{(0)} + \mathbb{E}[X]^\top \beta_{(0)}} \right) \tag{60}$$

$$+ \frac{\sqrt{n}}{c_{(0)} + \mathbb{E}[X]^\top \beta_{(0)}} \left( \hat{c}_{(1)} + \bar{X}^\top \hat{\beta}_{(1)} - c_{(1)} - \mathbb{E}[X]^\top \beta_{(1)} \right) \tag{61}$$

$$= \sqrt{n} \left( \bar{Z}^\top \hat{\gamma}_{(1)} \right) \left( \frac{1}{\bar{Z}^\top \hat{\gamma}_{(0)}} - \frac{1}{\mathbb{E}[Z]^\top \gamma_{(0)}} \right) \tag{62}$$

$$+ \frac{\sqrt{n}}{\mathbb{E}[Z]^\top \gamma_{(0)}} \left( \bar{Z}^\top \hat{\gamma}_{(1)} - \mathbb{E}[Z]^\top \gamma_{(1)} \right) \tag{63}$$

$$= \sqrt{n} \frac{\bar{Z}^\top \hat{\gamma}_{(1)}}{\bar{Z}^\top \hat{\gamma}_{(0)} \mathbb{E}[Z]^\top \gamma_{(0)}} \left( \mathbb{E}[Z]^\top \gamma_{(0)} - \bar{Z}^\top \hat{\gamma}_{(0)} \right) \tag{64}$$

$$+ \frac{\sqrt{n}}{\mathbb{E}[Z]^\top \gamma_{(0)}} \left( \bar{Z}^\top \hat{\gamma}_{(1)} - \mathbb{E}[Z]^\top \gamma_{(1)} \right). \tag{65}$$

Besides, we have

$$\bar{Z} - \mathbb{E}[Z] = \hat{e}\bar{Z}_{(1)} + (1 - \hat{e})\bar{Z}_{(0)} - e\mathbb{E}[Z|T = 1] - (1 - e)\mathbb{E}[Z|T = 0]$$
$$= (1 - e) \left( \bar{Z}_{(0)} - \mathbb{E}[Z|T = 0] \right) + e \left( \bar{Z}_{(1)} - \mathbb{E}[Z|T = 1] \right) + \left( \bar{Z}_{(1)} - \bar{Z}_{(0)} \right) (\hat{e} - e)$$
$$= \zeta(\theta_n - \theta_\infty),$$

where $\zeta = \left[ (1 - e)I_{d+1}, \ eI_{d+1}, \ 0_{d+1}, \ 0_{d+1}, \ \left( \bar{Z}_{(1)} - \bar{Z}_{(0)} \right) \right] \in \mathbb{R}^{(d+1) \times 4(d+1)+1}$ and

$$\boldsymbol{\theta}_n = \begin{pmatrix} \bar{Z}_{(0)} \\ \bar{Z}_{(1)} \\ \hat{\gamma}_{(0)} \\ \hat{\gamma}_{(1)} \\ \hat{e} \end{pmatrix}, \quad \boldsymbol{\theta}_\infty = \begin{pmatrix} E[Z|T = 0] \\ E[Z|T = 1] \\ \gamma_{(0)} \\ \gamma_{(1)} \\ e \end{pmatrix}.$$

Note that for all $t \in \{0, 1\}$,

$$\bar{Z}^\top \hat{\gamma}_{(t)} - \mathbb{E}[Z]^\top \gamma_{(t)} = \hat{\gamma}_{(t)}^\top \left( \bar{Z} - \mathbb{E}[Z] \right) + \mathbb{E}[Z]^\top \left( \hat{\gamma}_{(t)} - \gamma_{(t)} \right)$$
$$= \hat{\gamma}_{(t)}^\top \zeta(\theta_n - \theta_\infty) + \mathbb{E}[Z]^\top \left( \hat{\gamma}_{(t)} - \gamma_{(t)} \right)$$
$$= \hat{\alpha}_{(t)}^\top (\theta_n - \theta_\infty),$$

with

$$\hat{\alpha}_{(t)} = \begin{pmatrix} (1 - e)\hat{\gamma}_{(t)} \\ e\hat{\gamma}_{(t)} \\ \mathbb{1}_{t=0}\mathbb{E}[Z] \\ \mathbb{1}_{t=1}\mathbb{E}[Z] \\ \left( \bar{Z}_{(1)} - \bar{Z}_{(0)} \right)^\top \hat{\gamma}_{(t)}. \end{pmatrix}$$

. Therefore

$$\sqrt{n}(\tau_{\text{RR,OLS}} - \tau_{\text{RR}}) = \sqrt{n}\frac{\bar{Z}^\top\hat{\gamma}_{(1)}}{\bar{Z}^\top\hat{\gamma}_{(0)}\mathbb{E}[Z]^\top\gamma_{(0)}}\left(\mathbb{E}[Z]^\top\gamma_{(0)} - \bar{Z}^\top\hat{\gamma}_{(0)}\right)$$

$$+ \frac{\sqrt{n}}{\mathbb{E}[Z]^\top\gamma_{(0)}}\left(\bar{Z}^\top\hat{\gamma}_{(1)} - \mathbb{E}[Z]^\top\gamma_{(1)}\right)$$

$$= \frac{\sqrt{n}}{\mathbb{E}[Z]^\top\gamma_{(0)}}\hat{\alpha}_{(1)}^\top(\theta_n - \theta_\infty)$$

$$- \sqrt{n}\frac{\bar{Z}^\top\hat{\gamma}_{(1)}}{\bar{Z}^\top\hat{\gamma}_{(0)}\mathbb{E}[Z]^\top\gamma_{(0)}}\hat{\alpha}_{(0)}^\top(\theta_n - \theta_\infty)$$

Therefore, we get that

$$\sqrt{n}(\tau_{\text{RR,OLS}} - \tau_{\text{RR}}) = \sqrt{n}\left(\frac{1}{\mathbb{E}[Z]^\top\gamma_{(0)}}\hat{\alpha}_{(1)} - \frac{\bar{Z}^\top\hat{\gamma}_{(1)}}{\bar{Z}^\top\hat{\gamma}_{(0)}\mathbb{E}[Z]^\top\gamma_{(0)}}\hat{\alpha}_{(0)}\right)^\top(\theta_n - \theta_\infty).$$

According to the Law of Large Numbers,

$$\frac{1}{\mathbb{E}[Z]^\top\gamma_{(0)}}\hat{\alpha}_{(1)} - \frac{\bar{Z}^\top\hat{\gamma}_{(1)}}{\bar{Z}^\top\hat{\gamma}_{(0)}\mathbb{E}[Z]^\top\gamma_{(0)}}\hat{\alpha}_{(0)} \xrightarrow{p} \frac{\mathbb{E}[Z]^\top\gamma_{(1)}}{\mathbb{E}[Z]^\top\gamma_{(0)}}\left(\frac{\alpha_{(1)}}{\mathbb{E}[Z]^\top\gamma_{(1)}} - \frac{\alpha_{(0)}}{\mathbb{E}[Z]^\top\gamma_{(0)}}\right) := \alpha_\infty,$$

with, for all $t \in \{0, 1\}$,

$$\alpha_{(t)} = \begin{pmatrix} (1-e)\gamma_{(t)} \\ e\gamma_{(t)} \\ \mathbb{1}_{t=0}\mathbb{E}[Z] \\ \mathbb{1}_{t=1}\mathbb{E}[Z] \\ (\mathbb{E}[Z|T=1] - \mathbb{E}[Z|T=0])^\top\gamma_{(t)} \end{pmatrix}$$

and

$$\alpha_\infty = \frac{\mathbb{E}[Z]^\top\gamma_{(1)}}{\mathbb{E}[Z]^\top\gamma_{(0)}}\begin{pmatrix} \frac{(1-e)\gamma_{(1)}}{\mathbb{E}[Z]^\top\gamma_{(1)}} - \frac{(1-e)\gamma_{(0)}}{\mathbb{E}[Z]^\top\gamma_{(0)}} \\ \frac{e\gamma_{(1)}}{\mathbb{E}[Z]^\top\gamma_{(1)}} - \frac{e\gamma_{(0)}}{\mathbb{E}[Z]^\top\gamma_{(0)}} \\ -\frac{\mathbb{E}[Z]}{\mathbb{E}[Z]^\top\gamma_{(0)}} \\ \frac{\mathbb{E}[Z]}{\mathbb{E}[Z]^\top\gamma_{(1)}} \\ \frac{\gamma_{(0)}^\top(\mathbb{E}[Z|T=1]-\mathbb{E}[Z|T=0])}{\mathbb{E}[Z]^\top\gamma_{(0)}} - \frac{\gamma_{(1)}^\top(\mathbb{E}[Z|T=1]-\mathbb{E}[Z|T=0])}{\mathbb{E}[Z]^\top\gamma_{(1)}} \end{pmatrix}. \tag{66}$$

According to Proposition 7.7, letting $Q_t = \mathbb{E}\left[ZZ^\top|T=t\right]$ for all $t \in \{0, 1\}$, we have

$$\sqrt{n}(\theta_n - \theta_\infty) \xrightarrow{d} \mathcal{N}(0, \Sigma) \quad \text{where} \quad \Sigma = \begin{pmatrix} \frac{\text{Var}[Z|T=0]}{(1-e)} & 0 & 0 & 0 & 0 \\ 0 & \frac{\text{Var}[Z|T=1]}{e} & 0 & 0 & 0 \\ 0 & 0 & \frac{\sigma^2 Q_0^{-1}}{1-e} & 0 & 0 \\ 0 & 0 & 0 & \frac{\sigma^2 Q_1^{-1}}{e} & 0 \\ 0 & 0 & 0 & 0 & e(1-e) \end{pmatrix}.$$

By Slutsky's theorem,

$$\sqrt{n}\left(\frac{1}{\mathbb{E}[Z]^\top\gamma_{(0)}}\hat{\alpha}_{(1)} - \frac{\bar{Z}^\top\hat{\gamma}_{(1)}}{\bar{Z}^\top\hat{\gamma}_{(0)}\mathbb{E}[Z]^\top\gamma_{(0)}}\hat{\alpha}_{(0)}\right)^\top(\theta_n - \theta_\infty) \xrightarrow{d} \mathcal{N}\left(0, \alpha_\infty^\top\Sigma\alpha_\infty\right). \tag{67}$$

We now compute the covariance matrix

$$\frac{\alpha_\infty^\top \Sigma \alpha_\infty}{\left(\frac{\mathbb{E}[Z]^\top \gamma_{(1)}}{\mathbb{E}[Z]^\top \gamma_{(0)}}\right)^2} = (1-e)\left\|\frac{\gamma_{(1)}}{\mathbb{E}[Z]^\top \gamma_{(1)}} - \frac{\gamma_{(0)}}{\mathbb{E}[Z]^\top \gamma_{(0)}}\right\|^2_{\mathrm{Var}[Z|T=0]} + e\left\|\frac{\gamma_{(1)}}{\mathbb{E}[Z]^\top \gamma_{(1)}} - \frac{\gamma_{(0)}}{\mathbb{E}[Z]^\top \gamma_{(0)}}\right\|^2_{\mathrm{Var}[Z|T=1]} \tag{68}$$

$$+ \frac{\sigma^2}{1-e}\left\|\frac{\mathbb{E}[Z]}{\mathbb{E}[Z]^\top \gamma_{(0)}}\right\|^2_{Q_0^{-1}} + \frac{\sigma^2}{e}\left\|\frac{\mathbb{E}[Z]}{\mathbb{E}[Z]^\top \gamma_{(1)}}\right\|^2_{Q_1^{-1}} + e(1-e)\left\|\frac{\gamma_{(1)}}{\mathbb{E}[Z]^\top \gamma_{(1)}} - \frac{\gamma_{(0)}}{\mathbb{E}[Z]^\top \gamma_{(0)}}\right\|^2_{\Delta\Delta^\top}, \tag{69}$$

where $\Delta = \mathbb{E}[Z \mid T = 1] - \mathbb{E}[Z \mid T = 0]$. This variance can be rewritten as follows. Summing the first two terms and the last term in (69) leads to

$$\left\|\frac{\gamma_{(1)}}{\mathbb{E}[Z]^\top \gamma_{(1)}} - \frac{\gamma_{(0)}}{\mathbb{E}[Z]^\top \gamma_{(0)}}\right\|^2_J,$$

where $J = (1-e)\mathrm{Var}(Z \mid T = 0) + e\,\mathrm{Var}(Z \mid T = 1) + e(1-e)\Delta\Delta^\top$. Let us prove that $J = \mathrm{Var}(Z)$. Letting $Z_i$ the components of $Z$ for all $1 \leq i \leq d+1$, by the law of total covariance, we have

$$\mathrm{Cov}[Z_i, Z_j] = \mathbb{E}[\mathrm{Cov}[Z_i, Z_j|T]] + \mathrm{Cov}[\mathbb{E}[Z_i|T], \mathbb{E}[Z_j|T]], \tag{70}$$

with, since $T \in \{0, 1\}$,

$$\mathbb{E}[\mathrm{Cov}[Z_i, Z_j|T]] = e\,\mathrm{Cov}[Z_i, Z_j|T = 1] + (1-e)\,\mathrm{Cov}[Z_i, Z_j|T = 0]. \tag{71}$$

Besides, since $\mathbb{E}[Z] = (1-e)\mathbb{E}[Z \mid T = 0] + e\mathbb{E}[Z \mid T = 1]$, we can compute the deviations from the unconditional mean:

$$\begin{aligned}
\mathbb{E}[Z \mid T = 0] - \mathbb{E}[Z] &= \mathbb{E}[Z \mid T = 0] - ((1-e)\mathbb{E}[Z \mid T = 0] + e\mathbb{E}[Z \mid T = 1]) \\
&= (1 - (1-e))\mathbb{E}[Z \mid T = 0] - e\mathbb{E}[Z \mid T = 1] \\
&= -e\left(\mathbb{E}[Z \mid T = 1] - \mathbb{E}[Z \mid T = 0]\right) \\
&= -e\Delta.
\end{aligned}$$

and

$$\begin{aligned}
\mathbb{E}[Z \mid T = 1] - \mathbb{E}[Z] &= \mathbb{E}[Z \mid T = 1] - ((1-e)\mathbb{E}[Z \mid T = 0] + e\mathbb{E}[Z \mid T = 1]) \\
&= (1-e)\left(\mathbb{E}[Z \mid T = 1] - \mathbb{E}[Z \mid T = 0]\right) \\
&= (1-e)\Delta.
\end{aligned}$$

Now, we can compute the second term in (70)

$$\begin{aligned}
\mathrm{Cov}[\mathbb{E}[Z_i|T], \mathbb{E}[Z_j|T]] &= \mathbb{E}[(\mathbb{E}[Z_i|T] - \mathbb{E}[Z_i])(\mathbb{E}[Z_j|T] - \mathbb{E}[Z_j])] \tag{72} \\
&= e(\mathbb{E}[Z_i|T = 1] - \mathbb{E}[Z_i])(\mathbb{E}[Z_j|T = 1] - \mathbb{E}[Z_j]) \tag{73} \\
&\quad + (1-e)(\mathbb{E}[Z_i|T = 0] - \mathbb{E}[Z_i])(\mathbb{E}[Z_j|T = 0] - \mathbb{E}[Z_j]) \tag{74} \\
&= e(1-e)^2\Delta_i\Delta_j + e^2(1-e)\Delta_i\Delta_j \tag{75} \\
&= e(1-e)\Delta_i\Delta_j. \tag{76}
\end{aligned}$$

Consequently, according to (70),

$$\mathrm{Cov}[Z_i, Z_j] = e\,\mathrm{Cov}[Z_i, Z_j|T = 1] + (1-e)\,\mathrm{Cov}[Z_i, Z_j|T = 0] + e(1-e)\Delta_i\Delta_j, \tag{77}$$

which leads to

$$\mathrm{Var}[Z] = (1-e)\mathrm{Var}(Z \mid T = 0) + e\,\mathrm{Var}(Z \mid T = 1) + e(1-e)\Delta\Delta^\top. \tag{78}$$

Similarly the last two remaining terms we have for $t \in \{0, 1\}$:

$$\begin{aligned}
\left\|\frac{\mathbb{E}[Z]}{\mathbb{E}[Z]^\top \gamma_{(t)}}\right\|^2_{Q_t^{-1}} &= \frac{1}{(\mathbb{E}[Z]^\top \gamma_{(t)})^2}\|\mathbb{E}[Z]\|^2_{Q_t^{-1}} \\
&= \frac{1}{(\mathbb{E}[Z]^\top \gamma_{(t)})^2}\mathbb{E}[Z]^\top Q_t^{-1}\mathbb{E}[Z]
\end{aligned}$$

Note that we have $\mathbb{E}[Z] = e\mathbb{E}[Z|T=1] + (1-e)\mathbb{E}[Z|T=0]$ and that for $t \in \{0,1\}$,

$$\mathbb{E}[Z|T=t]^\top Q_t^{-1}\mathbb{E}[Z|T=t] = \begin{pmatrix}1\\\nu_t\end{pmatrix}^\top \begin{pmatrix}1+\nu_t^\top\Sigma_t^{-1}\nu_t & -\nu_t^\top\Sigma_t^{-1}\\-\Sigma_t^{-1}\nu_t & \Sigma_t^{-1}\end{pmatrix}\begin{pmatrix}1\\\nu_t\end{pmatrix} = 1,$$

and

$$\mathbb{E}[Z|T=1-t]^\top Q_t^{-1}\mathbb{E}[Z|T=1-t] = \begin{pmatrix}1\\\nu_{1-t}\end{pmatrix}^\top \begin{pmatrix}1+\nu_t^\top\Sigma_t^{-1}\nu_t & -\nu_t^\top\Sigma_t^{-1}\\-\Sigma_t^{-1}\nu_t & \Sigma_t^{-1}\end{pmatrix}\begin{pmatrix}1\\\nu_{1-t}\end{pmatrix}$$
$$= 1 + \|\nu_{1-t}-\nu_t\|_{\Sigma_t^{-1}}^2,$$

and

$$\mathbb{E}[Z|T=t]^\top Q_t^{-1}\mathbb{E}[Z|T=1-t] = \begin{pmatrix}1\\\nu_t\end{pmatrix}^\top \begin{pmatrix}1+\nu_t^\top\Sigma_t^{-1}\nu_t & -\mu_t^\top\Sigma_t^{-1}\\-\Sigma_t^{-1}\nu_t & \Sigma_t^{-1}\end{pmatrix}\begin{pmatrix}1\\\nu_{1-t}\end{pmatrix}$$
$$= 1.$$

Therefore, we have

$$\mathbb{E}[Z]^\top Q_0^{-1}\mathbb{E}[Z] = e^2\|\mathbb{E}[Z|T=1]\|_{Q_0^{-1}} + (1-e)^2\|\mathbb{E}[Z|T=0]\|_{Q_0^{-1}} + 2e(1-e)\langle\mathbb{E}[Z|T=0],\mathbb{E}[Z|T=1]\rangle_{Q_0^{-1}}$$
$$= e^2\|\mathbb{E}[Z|T=1]\|_{Q_0^{-1}} + (1-e)^2 + 2e(1-e)\langle\mathbb{E}[Z|T=0],\mathbb{E}[Z|T=1]\rangle_{Q_0^{-1}}$$
$$= (1-e)^2 + e^2\left(1+\|\mu_1-\mu_0\|_{\Sigma_0^{-1}}^2\right) + 2e(1-e)$$
$$= 1 + e^2\|\nu_1-\nu_0\|_{\Sigma_0^{-1}}^2,$$

and similarly $\mathbb{E}[Z]^\top Q_1^{-1}\mathbb{E}[Z] = 1 + (1-e)^2\|\nu_1-\nu_0\|_{\Sigma_1^{-1}}^2$. Finally, noting that for all $t \in \{0,1\}$

$$\mathbb{E}[Z]^\top\gamma_{(t)} = \mathbb{E}\left[Y^{(t)}\right] \quad\text{and}\quad \mathrm{Var}\left[Z\right] = \begin{pmatrix}0 & \cdots & 0\\\vdots & \mathrm{Var}\left[X\right] & \\0 & & \end{pmatrix},$$

we have, letting $\Sigma = \mathrm{Var}\left[X\right]$

$$V_{\mathrm{RR,G,OLS}} = \tau_{\mathrm{RR}}^2\left(\left\|\frac{\beta_{(1)}}{\mathbb{E}\left[Y^{(1)}\right]} - \frac{\beta_{(0)}}{\mathbb{E}\left[Y^{(0)}\right]}\right\|_\Sigma^2 + \sigma^2\left(\frac{1+(1-e)^2\|\nu_1-\nu_0\|_{\Sigma_1^{-1}}^2}{e\mathbb{E}\left[Y^{(1)}\right]^2} + \frac{1+e^2\|\nu_1-\nu_0\|_{\Sigma_0^{-1}}^2}{(1-e)\mathbb{E}\left[Y^{(0)}\right]^2}\right)\right).$$

$\square$

**Lemma 7.9** (**Comparison of the asymptotic variances of $\hat{\tau}_{\mathrm{RR,N}}$ and $\hat{\tau}_{\mathrm{RR,G}}$ under a linear model**). *Grant Assumption 2.2, Assumption 2.3 and Assumption 3.8. Recalling that $V_{RR,G,OLS}$ (resp. $V_{RR,G,OLS}$) is the asymptotic variance of the G-formula when oracle surface responses are used (resp. when they are estimated via OLS), we have*

$$V_{RR,N} = \tau_{RR}^2\left(\frac{\|\beta_{(1)}\|_\Sigma^2 + \sigma^2}{e\mathbb{E}\left[Y^{(1)}\right]^2} + \frac{\|\beta_{(0)}\|_\Sigma^2 + \sigma^2}{(1-e)\mathbb{E}\left[Y^{(0)}\right]^2}\right), \tag{79}$$

$$V_{RR,G,OLS} = \tau_{RR}^2\left(\left\|\frac{\beta_{(1)}}{\mathbb{E}\left[Y^{(1)}\right]} - \frac{\beta_{(0)}}{\mathbb{E}\left[Y^{(0)}\right]}\right\|_\Sigma^2 + \sigma^2\left(\frac{1}{e\mathbb{E}\left[Y^{(1)}\right]^2} + \frac{1}{(1-e)\mathbb{E}\left[Y^{(0)}\right]^2}\right)\right) \tag{80}$$

$$= V_{RR,G} + \tau_{RR}^2\sigma^2\left(\frac{1}{e\mathbb{E}\left[Y^{(1)}\right]^2} + \frac{1}{(1-e)\mathbb{E}\left[Y^{(0)}\right]^2}\right), \tag{81}$$

*and*

$$V_{RR,N} - V_{RR,G,OLS} = \tau_{RR}^2\left(e(1-e)\left\|\frac{\beta_{(1)}}{e\mathbb{E}\left[Y^{(1)}\right]} - \frac{\beta_{(0)}}{(1-e)\mathbb{E}\left[Y^{(0)}\right]}\right\|_\Sigma^2\right) \geq 0. \tag{82}$$

*Proof of Lemma 7.9.*

**First equality** The variance of $Y^{(a)}$ satisfies

$$
\begin{aligned}
\mathrm{Var}[Y^{(a)}] &= \mathrm{Var}[c_{(t)} + X^\top \beta_{(t)} + \varepsilon_{(t)}] \\
&= \mathrm{Var}[X^\top \beta_{(t)} + \varepsilon_{(t)}] && c_{(t)} \text{ is a constant} \\
&= \mathrm{Var}[X^\top \beta_{(t)}] + \mathrm{Var}[\varepsilon_{(t)}] + 2\,\mathrm{Cov}(X^\top \beta_{(t)}, \varepsilon_{(t)}) && \text{Bienaymé's identity} \\
&= ||\beta_{(t)}||_\Sigma + \sigma^2, && \text{(by linear model)}
\end{aligned}
$$

since

$$
\begin{aligned}
\mathrm{Cov}(X^\top \beta_{(t)}, \varepsilon_{(t)}) &= \mathbb{E}[X^\top \beta_{(t)} \varepsilon_{(t)}] - \mathbb{E}[X^\top \beta_{(t)}]\mathbb{E}[\varepsilon_{(t)}] \\
&= \mathbb{E}[X^\top \beta_{(t)} \mathbb{E}[\varepsilon_{(t)}|X]] - \mathbb{E}[X^\top \beta_{(t)}]\mathbb{E}[\mathbb{E}[\varepsilon_{(t)}|X]] && \text{(by total expectation)} \\
&= 0, && \mathbb{E}[\varepsilon_{(t)}|X] = 0,
\end{aligned}
$$

and, using Eve's law, $\mathrm{Var}[\varepsilon_{(t)}] = \mathbb{E}[\mathrm{Var}[\varepsilon_{(t)}|X]] + \mathrm{Var}[\mathbb{E}[\varepsilon_{(t)}|X]] = \sigma^2$. Thus, $V_{\mathrm{RR,N}}$ satisfies

$$
V_{\mathrm{RR,N}} = \tau_{\mathrm{RR}}^2 \left( \frac{\mathrm{Var}(Y^{(1)})}{e\mathbb{E}[Y^{(1)}]^2} + \frac{\mathrm{Var}(Y^{(0)})}{(1-e)\mathbb{E}[Y^{(0)}]^2} \right) \tag{83}
$$

$$
= \tau_{\mathrm{RR}}^2 \left( \frac{||\beta_{(1)}||_\Sigma^2 + \sigma^2}{e\mathbb{E}[Y^{(1)}]^2} + \frac{||\beta_{(0)}||_\Sigma^2 + \sigma^2}{(1-e)\mathbb{E}[Y^{(0)}]^2} \right). \tag{84}
$$

**Second and third equality** According to Proposition 3.9 ,

$$
\frac{V_{\mathrm{RR,G,OLS}}}{\tau_{\mathrm{RR}}^2} = \left\| \frac{\beta_{(1)}}{\mathbb{E}\left[Y^{(1)}\right]} - \frac{\beta_{(0)}}{\mathbb{E}\left[Y^{(0)}\right]} \right\|_\Sigma^2 + \sigma^2 \left( \frac{1 + (1-e)^2 \|\nu_1 - \nu_0\|_{\Sigma_1^{-1}}^2}{e\mathbb{E}\left[Y^{(1)}\right]^2} + \frac{1 + e^2 \|\nu_1 - \nu_0\|_{\Sigma_0^{-1}}^2}{(1-e)\mathbb{E}\left[Y^{(0)}\right]^2} \right)
$$

Since we are in a RCT setting, we have that $\nu_1 = \nu_0$ and $\Sigma_1 = \Sigma_0 = \Sigma$. Therefore

$$
\frac{V_{\mathrm{RR,G,OLS}}}{\tau_{\mathrm{RR}}^2} = \left\| \frac{\beta_{(1)}}{\mathbb{E}\left[Y^{(1)}\right]} - \frac{\beta_{(0)}}{\mathbb{E}\left[Y^{(0)}\right]} \right\|_\Sigma^2 + \sigma^2 \left( \frac{1}{e\mathbb{E}\left[Y^{(1)}\right]^2} + \frac{1}{(1-e)\mathbb{E}\left[Y^{(0)}\right]^2} \right)
$$

The first term corresponds to the Oracle variance of the G-formula. Indeed, for all $t \in \{0,1\}$,

$$
\begin{aligned}
\mathrm{Var}[\mu_{(t)}(X)] &= \mathrm{Var}[\mathbb{E}[Y^{(t)}|X]] \\
&= \mathrm{Var}[\mathbb{E}[c_{(t)}|X] + \mathbb{E}[X^\top \beta_{(t)}|X] + \mathbb{E}[\varepsilon_{(t)}|X]] \\
&= \mathrm{Var}[c_{(t)} + \mathbb{E}[X^\top \beta_{(t)}|X]] \\
&= \mathrm{Var}[\mathbb{E}[X^\top \beta_{(t)}|X]] \\
&= \mathrm{Var}[X^\top \beta_{(t)}] \\
&= ||\beta_{(t)}||_\Sigma^2.
\end{aligned}
$$

Besides, the covariance between $\mu_1(X)$ and $\mu_0(X)$ satisfies

$$
\begin{aligned}
\mathrm{Cov}(\mu_{(1)}(X), \mu_{(0)}(X)) =& \mathbb{E}[\mu_{(1)}(X)\mu_{(0)}(X)] - \mathbb{E}[Y^{(0)}]\mathbb{E}[Y^{(1)}] \\
=& \mathbb{E}[(c_{(1)} + X^\top\beta_{(1)})(c_{(0)} + X^\top\beta_{(0)})] - \mathbb{E}[Y^{(0)}]\mathbb{E}[Y^{(1)}] \\
=& \mathbb{E}[c_{(1)}c_{(0)}] + \mathbb{E}[c_{(1)}X^\top\beta_{(0)}] + \mathbb{E}[c_{(0)}X^\top\beta_{(1)}] \\
& + \mathbb{E}[X^\top\beta_{(0)}X^\top\beta_{(1)}] - \mathbb{E}[Y^{(0)}]\mathbb{E}[Y^{(1)}] \\
=& \mathbb{E}[X^\top\beta_{(0)}X^\top\beta_{(1)}] \\
=& \mathbb{E}\left[\sum_j X_j\beta_{(0),j} \sum_k X_k\beta_{(1),k}\right] \\
=& \sum_j\sum_k \beta_{(0),j}\beta_{(1),k}\mathbb{E}[X_kX_j] \\
=& \langle\beta_{(0)}, \beta_{(1)}\rangle_\Sigma.
\end{aligned}
$$

Therefore,

$$
\begin{aligned}
V_{\mathrm{RR,G}} = \tau_{\mathrm{RR}}^2 \, \mathrm{Var}\left(\frac{\mu_1(X)}{\mathbb{E}[Y^{(1)}]} - \frac{\mu_0(X)}{\mathbb{E}[Y^{(0)}]}\right) &= \tau_{\mathrm{RR}}^2\left(\frac{\mathrm{Var}(\mu_1(X))}{\mathbb{E}[Y^{(1)}]^2} + \frac{\mathrm{Var}(\mu_0(X))}{\mathbb{E}[Y^{(0)}]^2} - 2\frac{\mathrm{Cov}(\mu_0(X), \mu_1(X))}{\mathbb{E}[Y^{(0)}]\mathbb{E}[Y^{(1)}]}\right) \\
&= \tau_{\mathrm{RR}}^2\left(\left\|\frac{\beta_{(1)}}{\mathbb{E}[Y^{(1)}]}\right\|_\Sigma^2 + \left\|\frac{\beta_{(0)}}{\mathbb{E}[Y^{(0)}]}\right\|_\Sigma^2 - 2\frac{\langle\beta_{(0)}, \beta_{(1)}\rangle_\Sigma}{\mathbb{E}[Y^{(0)}]\mathbb{E}[Y^{(1)}]}\right) \\
&= \tau_{\mathrm{RR}}^2\left\|\frac{\beta_{(1)}}{\mathbb{E}[Y^{(1)}]} - \frac{\beta_{(0)}}{\mathbb{E}[Y^{(0)}]}\right\|_\Sigma^2.
\end{aligned}
$$

**Last inequality** A simple computation leads to

$$
\begin{aligned}
\frac{V_{\mathrm{RR,N}} - V_{\mathrm{RR,G}}}{\tau_{\mathrm{RR}}^2} &= \frac{\left\|\beta_{(1)}\right\|_\Sigma^2 + \sigma^2}{e\mathbb{E}[Y^{(1)}]^2} + \frac{\left\|\beta_{(0)}\right\|_\Sigma^2 + \sigma^2}{(1-e)\mathbb{E}[Y^{(0)}]^2} \\
&\quad - \left(\left\|\frac{\beta_{(1)}}{\mathbb{E}[Y^{(1)}]} - \frac{\beta_{(0)}}{\mathbb{E}[Y^{(0)}]}\right\|_\Sigma^2 + \sigma^2\left(\frac{1}{e\mathbb{E}[Y^{(1)}]^2} + \frac{1}{(1-e)\mathbb{E}[Y^{(0)}]^2}\right)\right) \\
&= \left(\frac{1-e}{e}\right)\frac{\left\|\beta_{(1)}\right\|_\Sigma^2}{\mathbb{E}[Y^{(1)}]^2} + \left(\frac{e}{1-e}\right)\frac{\left\|\beta_{(0)}\right\|_\Sigma^2}{\mathbb{E}[Y^{(0)}]^2} + \frac{2\langle\beta_{(1)}, \beta_{(0)}\rangle_\Sigma}{\mathbb{E}[Y^{(1)}]\mathbb{E}[Y^{(0)}]} \\
&= e(1-e)\left\|\frac{\beta_{(1)}}{e\mathbb{E}[Y^{(1)}]} - \frac{\beta_{(0)}}{(1-e)\mathbb{E}[Y^{(0)}]}\right\|_\Sigma^2.
\end{aligned}
$$

$\square$

### 7.3.5. RISK RATIO ONE-STEP ESTIMATOR

*Proof of Definition 3.10.* We will use (Kennedy, 2022; 2015) notation in this proof. If you are not familiar on how to compute an influence function, note that it is very similar to compute the derivative of a function. We define our estimand quantity

$$
\psi = \frac{\mathbb{E}[\mathbb{E}[Y|T=1, X]]}{\mathbb{E}[\mathbb{E}[Y|T=0, X]]} = \frac{\psi_1}{\psi_0}.
$$

We can now compute the influence function $\varphi$ of $\psi$.

$$\varphi = \mathbb{IF}(\psi) = \mathbb{IF}\left(\frac{\psi_1}{\psi_0}\right) = \frac{\mathbb{IF}(\psi_1)\,\psi_0 - \mathbb{IF}(\psi_0)\,\psi_1}{\psi_0^2}$$
$$= \frac{\mathbb{IF}(\psi_1)}{\psi_0} - \psi\frac{\mathbb{IF}(\psi_0)}{\psi_0}.$$

According to Example 2 in (Kennedy, 2022), we have

$$\mathbb{IF}(\psi_1) = \mu_1(X) + T\frac{Y - \mu_1(X)}{e(X)} - \psi_1$$

$$\text{and} \quad \mathbb{IF}(\psi_0) = \mu_0(X) + (1 - T)\frac{Y - \mu_0(X)}{1 - e(X)} - \psi_0.$$

Therefore,

$$\varphi = \frac{\mathbb{IF}(\psi_1)}{\psi_0} - \psi\frac{\mathbb{IF}(\psi_0)}{\psi_0}$$

$$= \frac{\mu_1(X) + T\frac{Y - \mu_1(X)}{e(X)} - \psi_1}{\psi_0} - \psi\frac{\mu_0(X) + (1 - T)\frac{Y - \mu_0(X)}{1 - e(X)} - \psi_0}{\psi_0}$$

$$= \frac{\mu_1(X) + T\frac{Y - \mu_1(X)}{e(X)}}{\psi_0} - \psi - \psi\left(\frac{\mu_0(X) + (1 - T)\frac{Y - \mu_0(X)}{1 - e(X)}}{\psi_0} - 1\right)$$

$$= \frac{\mu_1(X) + T\frac{Y - \mu_1(X)}{e(X)}}{\psi_0} - \psi\frac{\mu_0(X) + (1 - T)\frac{Y - \mu_0(X)}{1 - e(X)}}{\psi_0}.$$

As referenced in (Kennedy, 2022) regarding the semiparametric von Mises expansion, consider the functional $\psi : P \to \mathbb{R}$, where $P$ represents the true data distribution and $\hat{P}$ its estimation. The expansion is formulated as:

$$\psi(\hat{P}) - \psi(P) = \int \varphi(z; \hat{P})d(\hat{P} - P)(z) + R_2(\hat{P}, P), \tag{85}$$

for all distributions $\hat{P}$ and $P$. The influence function $\varphi(z; P)$, associated with $\psi$, is a function with zero mean and finite variance as defined by (Tsiatis, 2006)

$$\int \varphi(z; P)dP(z) = 0 \quad \text{and} \quad \int \varphi(z; P)^2 dP(z) < \infty, \tag{86}$$

and $R_2(\hat{P}, P)$ denotes a second-order remainder term. According to the expansion in (85), most plug-in estimators $\psi(\hat{P})$ are biased to the first order, evidenced by:

$$\psi(P) = \psi(\hat{P}) + \int \varphi(z; \hat{P})dP(z) + R_2(\hat{P}, P),$$

since $\int \varphi(z; \hat{P})d\hat{P}(z) = 0$. Therefore, a first-order approximation of $\psi(P)$ is given by $\psi(\hat{P}) + \int \varphi(z; \hat{P})dP(z)$ which can be estimated via

$$\hat{\tau}_{\text{RR-OS}} = \hat{\psi} + \frac{1}{n}\sum_{i=1}^{n}\varphi(Z_i)$$

$$= \hat{\psi} + \frac{1}{n}\sum_{i=1}^{n}\frac{\mu_1(X_i) + T_i\frac{Y_i - \mu_1(X_i)}{e(X_i)}}{\hat{\psi}_0} - \hat{\psi}\frac{\mu_0(X_i) + (1 - T_i)\frac{Y_i - \mu_0(X_i)}{1 - e(X_i)}}{\hat{\psi}_0}$$

$$= \hat{\psi}\left(1 - \frac{\frac{1}{n}\sum_{i=1}^{n}\mu_0(X_i) + (1 - T_i)\frac{Y_i - \mu_0(X_i)}{1 - e(X_i)}}{\hat{\psi}_0}\right) + \frac{\frac{1}{n}\sum_{i=1}^{n}\mu_1(X_i) + T_i\frac{Y_i - \mu_1(X_i)}{e(X_i)}}{\hat{\psi}_0}$$

$$= \frac{\sum_{i=1}^{n}\hat{\mu}_1(X_i)}{\sum_{i=1}^{n}\hat{\mu}_0(X_i)}\left(1 - \frac{\sum_{i=1}^{n}\hat{\mu}_0(X_i) + \frac{(1 - T_i)(Y_i - \hat{\mu}_0(X_i))}{1 - \hat{e}(X_i)}}{\sum_{i=1}^{n}\hat{\mu}_0(X_i)}\right) + \frac{\sum_{i=1}^{n}\hat{\mu}_1(X_i) + \frac{T_i(Y_i - \hat{\mu}_1(X_i))}{\hat{e}(X_i)}}{\sum_{i=1}^{n}\hat{\mu}_0(X_i)}.$$

□

*Proof of Proposition 3.11.*
**Asymptotic bias and variance of the cross-fitted One-step estimator** Recall that

$$\psi(P) = \frac{\mathbb{E}_P\left[\mathbb{E}_P[Y \mid X, T = 1]\right]}{\mathbb{E}_P\left[\mathbb{E}_P[Y \mid X, T = 0]\right]} = \frac{\psi_1}{\psi_0} \tag{87}$$

$$\psi(\hat{P}) = \frac{\sum_{i=1}^n \hat{\mu}_1(X_i)}{\sum_{i=1}^n \hat{\mu}_0(X_i)} = \frac{\hat{\psi}_1}{\hat{\psi}_0} \tag{88}$$

$$\varphi(Z; \hat{P}) = \frac{\hat{\mu}_1(X_i) + T_i \frac{Y_i - \hat{\mu}_1(X_i)}{\hat{e}(X_i)}}{\hat{\psi}_0} - \hat{\psi}\frac{\hat{\mu}_0(X_i) + (1 - T_i)\frac{Y_i - \hat{\mu}_0(X_i)}{1 - \hat{e}(X_i)}}{\hat{\psi}_0} \tag{89}$$

where $P$ represents the true underlying data distribution and $\hat{P}$ the distribution where oracle quantities have been replaced by plug-in estimates. We express $\psi(P)$ as follows:

$$\psi(P) = \psi(\hat{P}) + \int \varphi(z; \hat{P})dP(z) + R_2(\hat{P}, P),$$

where $R_2$ encapsulates higher order remainder terms.

To elucidate, we rearrange to find $\psi(\hat{P}) - \psi(P)$:

$$\psi(\hat{P}) - \psi(P) = R_2(P, \hat{P}) - \int \varphi(z; \hat{P})dP(z)$$

$$= \frac{1}{n}\sum_{i=1}^n \varphi(Z_i; P) - \frac{1}{n}\sum_{i=1}^n \varphi(Z_i; \hat{P})$$

$$+ \frac{1}{n}\sum_{i=1}^n \left(\varphi(Z_i; \hat{P}) - \varphi(Z_i; P)\right) - \int \left(\varphi(z; \hat{P}) - \varphi(z; P)\right) dP(z)$$

$$+ R_2(P, \hat{P}).$$

Recalling that $\hat{\tau}_{\text{RR-OS}} = \psi(\hat{P}) + \frac{1}{n}\sum_{i=1}^n \varphi(Z_i; \hat{P})$ and $\tau_{\text{RR}} = \psi(P)$, we have

$$\hat{\tau}_{\text{RR-OS}} - \tau_{\text{RR}} = \psi(\hat{P}) + \frac{1}{n}\sum_{i=1}^n \varphi(Z_i; \hat{P}) - \psi(P) \tag{90}$$

$$= \frac{1}{n}\sum_{i=1}^n \varphi(Z_i; P) \tag{91}$$

$$+ \frac{1}{n}\sum_{i=1}^n \left(\varphi(Z_i; \hat{P}) - \varphi(Z_i; P)\right) - \int \left(\varphi(z; \hat{P}) - \varphi(z; P)\right) dP(z) \tag{92}$$

$$+ R_2(P, \hat{P}). \tag{93}$$

The first term is a sample average of centered i.i.d. terms since, by definition (86), $\int \varphi(z; P)dP(z) = 0$. According to the central limit theorem, it converges to a normally distributed random variable with variance $\text{Var}(\varphi(Z))/n$.

Following the work of (Vaart, 1998), we consider the second term in (93), that is

$$\frac{1}{n}\sum_{i=1}^n \left(\varphi(Z_i; \hat{P}) - \varphi(Z_i; P)\right) - \int \left(\varphi(z; \hat{P}) - \varphi(z; P)\right) dP(z).$$

Since our estimator is built on a cross-fitting strategy with $K$ folds $\mathcal{I}_1, \ldots \mathcal{I}_K$, containing respectively $n_1, \ldots, n_K$ observations, the above quantity may be written as

$$\frac{1}{n}\sum_{k=1}^K \sum_{i\in\mathcal{I}_k} \left(\varphi(Z_i; \hat{P}^{-k}) - \varphi(Z_i; P)\right) - \int \left(\varphi(z; \hat{P}) - \varphi(z; P)\right) dP(z),$$

where $\hat{P}^{-k}$ corresponds to a data distribution where oracle quantity are replaced by plug-in estimates built on all observations except those in $\mathcal{I}_k$. We denote this set of observations as $\mathcal{I}_{-k}$. We let $\hat{\varphi}^{-k}(Z) = \varphi(Z; \hat{P}^{-k})$ and

$$U_k = \left( \mathbb{P}_n^{(k)} - P \right) \left( \hat{\varphi}^{-k}(Z) - \varphi(Z) \right), \tag{94}$$

where $\mathbb{P}_n^{(k)}$ is the empirical measure over $\mathcal{I}_k$. The quantity of interest can thus be written as

$$\frac{1}{n} \sum_{k=1}^{K} \sum_{i \in \mathcal{I}_k} \left( \varphi^{-k}(Z_i) - \varphi(Z_i; P) \right) - \int \left( \varphi(z; \hat{P}) - \varphi(z; P) \right) dP(z) = \frac{1}{n} \sum_{k=1}^{K} n_k U_k. \tag{95}$$

The expectation and variance of $U_k$ satisfy

$$\mathbb{E}\left[ U_k \mid \mathcal{I}_{-k} \right] = \mathbb{E}\left[ \left( \mathbb{P}_n^{(k)} - P \right) (\hat{\varphi}^{-k} - \varphi) \mid \mathcal{I}_{-k} \right] \tag{96}$$

$$= \mathbb{E}\left[ \mathbb{P}_n^{(k)}(\hat{\varphi}^{-k} - \varphi) \mid \mathcal{I}_{-k} \right] - \mathbb{E}\left[ P(\hat{\varphi}^{-k} - \varphi) \mid \mathcal{I}_{-k} \right] \tag{97}$$

$$= \mathbb{E}\left[ \hat{\varphi}^{-k}(Z) - \varphi(Z) \right] - \mathbb{E}\left[ \hat{\varphi}^{-k}(Z) - \varphi(Z) \right] \tag{98}$$

$$= 0, \tag{99}$$

and

$$\mathrm{Var}\left[ U_k \mid \mathcal{I}_{-k} \right] = \mathrm{Var}\left[ \left( \mathbb{P}_n^{(k)} - P \right) (\hat{\varphi}^{-k} - \varphi) \mid \mathcal{I}_{-k} \right] \tag{100}$$

$$= \mathrm{Var}\left[ \mathbb{P}_n^{(k)}(\hat{\varphi}^{-k} - \varphi) - P(\hat{\varphi}^{-k} - \varphi) \mid \mathcal{I}_{-k} \right] \tag{101}$$

$$= \mathrm{Var}\left[ \frac{1}{n_k} \sum_{i=1}^{n_k} \left( \hat{\varphi}^{-k}(Z_i) - \varphi(Z_i) \right) \mid \mathcal{I}_{-k} \right] \tag{102}$$

$$= \frac{1}{n_k} \mathrm{Var}\left[ \hat{\varphi}^{-k}(Z) - \varphi(Z) \mid \mathcal{I}_{-k} \right] \tag{103}$$

$$\leq \frac{1}{n_k} \mathbb{E}\left[ (\hat{\varphi}^{-k}(Z) - \varphi(Z))^2 \mid \mathcal{I}_{-k} \right]. \tag{104}$$

Let $a > 0$. Applying Chebyshev's inequality leads to

$$\mathbb{P}\left( \frac{|U_k - \mathbb{E}[U_k \mid \mathcal{I}_{-k}]|}{\sqrt{\mathrm{Var}[U_k \mid \mathcal{I}_{-k}]}} \geq a \mid \mathcal{I}_{-k} \right) \leq \frac{1}{a^2} \tag{105}$$

$$\Longleftrightarrow \mathbb{P}\left( \frac{|U_k|}{\sqrt{\mathrm{Var}[U_k \mid \mathcal{I}_{-k}]}} \geq a \mid \mathcal{I}_{-k} \right) \leq \frac{1}{a^2}. \tag{106}$$

Thus,

$$\mathbb{P}\left( \frac{|U_k| \sqrt{n_k}}{\sqrt{\mathbb{E}\left[ (\hat{\varphi}^{-k}(Z) - \varphi(Z))^2 \mid \mathcal{I}_{-k} \right]}} \geq a \mid \mathcal{I}_{-k} \right) \leq \mathbb{P}\left( \frac{|U_k|}{\sqrt{\mathrm{Var}[U_k \mid \mathcal{I}_{-k}]}} \geq a \mid \mathcal{I}_{-k} \right) \leq \frac{1}{a^2}, \tag{107}$$

which leads to

$$\mathbb{P}\left( |U_k| \sqrt{n_k} \geq a \mid \mathcal{I}_{-k} \right) \leq \frac{\mathbb{E}\left[ (\hat{\varphi}^{-k}(Z) - \varphi(Z))^2 \mid \mathcal{I}_{-k} \right]}{a^2}. \tag{108}$$

Finally, taking the expectation on both sides leads to

$$\mathbb{P}\left( |U_k| \sqrt{n_k} \geq a \right) \leq \frac{\mathbb{E}\left[ (\hat{\varphi}^{-k}(Z) - \varphi(Z))^2 \right]}{a^2}. \tag{109}$$

According to (95), the quantity of interest takes the form

$$\frac{1}{n}\sum_{k=1}^{K} n_k U_k = \sum_{k=1}^{K} \frac{n_k}{n} U_k. \tag{110}$$

Hence,

$$\mathbb{P}\left(\sqrt{n}\frac{n_k}{n}|U_k| \le a\frac{\sqrt{n_k}}{\sqrt{n}}\right) \ge 1 - \frac{\mathbb{E}\left[(\hat{\varphi}^{-k}(Z) - \varphi(Z))^2\right]}{a^2}. \tag{111}$$

Therefore,

$$\mathbb{P}\left(\sqrt{n}\sum_{k=1}^{K}\frac{n_k}{n}|U_k| \le a\sum_{k=1}^{K}\frac{\sqrt{n_k}}{\sqrt{n}}\right) \ge 1 - \sum_{k=1}^{K}\frac{\mathbb{E}\left[(\hat{\varphi}^{-k}(Z) - \varphi(Z))^2\right]}{a^2} \tag{112}$$

$$\Rightarrow \quad \mathbb{P}\left(\sqrt{n}\sum_{k=1}^{K}\frac{n_k}{n}|U_k| \le aK\right) \ge 1 - \sum_{k=1}^{K}\frac{\mathbb{E}\left[(\hat{\varphi}^{-k}(Z) - \varphi(Z))^2\right]}{a^2}, \tag{113}$$

which proves that $\sum_{k=1}^{K}\frac{n_k}{n}U_k = o_P(1/\sqrt{n})$ as $K$ is fixed and $\varphi^{-k}$ is $L^2$ consistent.

Regarding the last term, note that

$$R_2(P, \hat{P}) = \psi(\hat{P}) - \psi(P) + \int \varphi(z; \hat{P})dP(z) \tag{114}$$

$$= \psi(\hat{P}) - \psi(P) + \mathbb{E}[\varphi(Z; \hat{P})] \tag{115}$$

$$= \psi(\hat{P}) - \psi(P) + \mathbb{E}\left[\frac{\hat{\mu}_1(X) + T\frac{Y - \hat{\mu}_1(X)}{\hat{e}(X)}}{\hat{\psi}_0} - \hat{\psi}\frac{\hat{\mu}_0(X) + (1-T)\frac{Y - \hat{\mu}_0(X)}{1 - \hat{e}(X)}}{\hat{\psi}_0}\right] \tag{116}$$

$$= \psi(\hat{P}) - \psi(P) + \frac{\mathbb{E}\left[\hat{\mu}_1(X) + T\frac{Y - \hat{\mu}_1(X)}{\hat{e}(X)}\right]}{\hat{\psi}_0} - \hat{\psi}\frac{\mathbb{E}\left[\hat{\mu}_0(X) + (1-T)\frac{Y - \hat{\mu}_0(X)}{1 - \hat{e}(X)}\right]}{\hat{\psi}_0} \tag{117}$$

$$= \psi(\hat{P}) - \psi(P) + \frac{\mathbb{E}\left[\hat{\mu}_1(X) - \mu_1(X) + T\frac{Y - \hat{\mu}_1(X)}{\hat{e}(X)}\right]}{\hat{\psi}_0} + \frac{\psi_1}{\hat{\psi}_0} \tag{118}$$

$$- \hat{\psi}\frac{\mathbb{E}\left[\hat{\mu}_0(X) - \mu_0(X) + (1-T)\frac{Y - \hat{\mu}_0(X)}{1 - \hat{e}(X)}\right]}{\hat{\psi}_0} - \hat{\psi}\frac{\psi_0}{\hat{\psi}_0}. \tag{119}$$

Note that

$$\mathbb{E}\left[\hat{\mu}_1(X) - \mu_1(X) + T\frac{Y - \hat{\mu}_1(X)}{\hat{e}(X)}\right] = \mathbb{E}\left[\frac{1}{\hat{e}(X)}(\mu_1(X) - \hat{\mu}_1(X))(\hat{e}(X) - e(X))\right] \tag{120}$$

$$\text{Positivity} \quad \le \frac{1}{\eta}\mathbb{E}\left[(\mu_1(X) - \hat{\mu}_1(X))(\hat{e}(X) - e(X))\right] \tag{121}$$

$$\text{Cauchy-Schwarz} \quad \le \frac{1}{\eta}\mathbb{E}\left[(\hat{e}(X) - e(X))^2\right]^{1/2}\mathbb{E}\left[(\hat{\mu}_1(X) - \mu_1(X))^2\right]^{1/2} \tag{122}$$

$$= o_p\left(\frac{1}{\sqrt{n}}\right). \tag{123}$$

Similarly,

$$\mathbb{E}\left[\hat{\mu}_1(X) - \mu_1(X) + T\frac{Y - \hat{\mu}_1(X)}{\hat{e}(X)}\right] = o_p\left(\frac{1}{\sqrt{n}}\right). \tag{124}$$

For the last term in (119), since $\psi = \psi_1/\psi_0$ and $\hat{\psi} = \hat{\psi}_1/\hat{\psi}_0$,

$$
\begin{aligned}
\hat{\psi} - \psi + \frac{\psi_1}{\hat{\psi}_0} - \hat{\psi}\frac{\psi_0}{\hat{\psi}_0} &= \psi_1\left(\frac{1}{\hat{\psi}_0} - \frac{1}{\psi_0}\right) + \hat{\psi}\left(1 - \frac{\psi_0}{\hat{\psi}_0}\right) \\
&= \psi_1\frac{\psi_0 - \hat{\psi}_0}{\psi_0\hat{\psi}_0} + \hat{\psi}\left(\frac{\hat{\psi}_0 - \psi_0}{\hat{\psi}_0}\right) \\
&= \left(\frac{\hat{\psi}_0 - \psi_0}{\hat{\psi}_0}\right)\left(\hat{\psi} - \psi\right) \\
&= \left(\frac{\hat{\psi}_0 - \psi_0}{\hat{\psi}_0}\right)\left(\left(\frac{1}{\hat{\psi}_0} - \frac{1}{\psi_0}\right)\hat{\psi}_1 + \frac{1}{\psi_0}\left(\hat{\psi}_1 - \psi_1\right)\right) \\
&= \frac{1}{\psi_0\hat{\psi}_0}\left((\hat{\psi}_0 - \psi_0)(\hat{\psi}_1 - \psi_1) - \hat{\psi}(\hat{\psi}_0 - \psi_0)(\hat{\psi}_0 - \psi_0)\right).
\end{aligned}
$$

By assumption, we have

$$
(\hat{\psi}_0 - \psi_0)(\hat{\psi}_1 - \psi_1) = \mathbb{E}\left[\hat{\mu}_0(X) - \mu_0(X)\right]\mathbb{E}\left[\hat{\mu}_1(X) - \mu_1(X)\right] \tag{125}
$$

$$
\leq \left(\mathbb{E}\left[(\hat{\mu}_0(X) - \mu_0(X))^2\right]\right)^{1/2}\left(\mathbb{E}\left[(\hat{\mu}_1(X) - \mu_1(X))^2\right]\right)^{1/2} \tag{126}
$$

$$
= o_p\left(\frac{1}{\sqrt{n}}\right) \tag{127}
$$

and

$$
\begin{aligned}
(\hat{\psi}_0 - \psi_0)(\hat{\psi}_0 - \psi_0) &= (\mathbb{E}\left[\hat{\mu}_0(X)\right] - \mathbb{E}\left[\mu_0(X)\right])^2 \\
&= o_p\left(\frac{1}{\sqrt{n}}\right).
\end{aligned}
$$

By assumption, $\mathbb{E}\left[(\hat{\mu}_0(X) - \mu_0(X))^2\right]$ tends to zero. Thus, $\hat{\psi}_0 = \mathbb{E}[\hat{\mu}_0]$ tends to $\psi_0 = \mathbb{E}[\mu_0(X)]$. Thus,

$$
\hat{\psi} - \psi + \frac{\psi_1}{\hat{\psi}_0} - \hat{\psi}\frac{\psi_0}{\hat{\psi}_0} = o_p\left(\frac{1}{\sqrt{n}}\right),
$$

which implies that $R_2(P, \hat{P}) = o_p\left(n^{-1/2}\right)$. Finally,

$$
\sqrt{n}\left(\hat{\tau}_{\text{RR-OS}} - \tau_{\text{RR}}\right) = \frac{1}{\sqrt{n}}\sum_{i=1}^n \varphi(Z_i; P) + o_p\left(\frac{1}{\sqrt{n}}\right)
$$

and thus

$$
\sqrt{n}\left(\hat{\tau}_{\text{RR-OS}} - \tau_{\text{RR}}\right) \xrightarrow{d} \mathcal{N}\left(0, \text{Var}(\varphi)\right),
$$

where

$$
\text{Var}(\varphi) = \text{Var}\left(\frac{\mu_1(X) + T\frac{Y - \mu_1(X)}{e(X)}}{\psi_0} - \psi\frac{\mu_0(X) + (1 - T)\frac{Y - \mu_0(X)}{1 - e(X)}}{\psi_0}\right) \tag{128}
$$

$$
= \psi^2\,\text{Var}\left(\frac{\mu_1(X) + T\frac{Y - \mu_1(X)}{e(X)}}{\psi_1}\frac{\mu_0(X) + (1 - T)\frac{Y - \mu_0(X)}{1 - e(X)}}{\psi_0}\right) \tag{129}
$$

$$
= \tau_{\text{RR}}^2\,\text{Var}\left(\frac{g_1(Z)}{\mathbb{E}\left[Y^{(1)}\right]} - \frac{g_0(Z)}{\mathbb{E}\left[Y^{(0)}\right]}\right). \tag{130}
$$

Using Bienaymé's identity, we get

$$\text{Var}\left(\frac{g_1(Z)}{\mathbb{E}\left[Y^{(1)}\right]} - \frac{g_0(Z)}{\mathbb{E}\left[Y^{(0)}\right]}\right) = \text{Var}\left(\frac{\mu_1(X)}{\mathbb{E}\left[Y^{(1)}\right]} - \frac{\mu_0(X)}{\mathbb{E}\left[Y^{(0)}\right]}\right) + \text{Var}\left(\frac{T(Y - \mu_1(X))}{\mathbb{E}\left[Y^{(1)}\right] e(X)} - \frac{(1 - T)(Y - \mu_0(X))}{\mathbb{E}\left[Y^{(0)}\right] (1 - e(X))}\right)$$
(131)

$$+ 2\,\text{Cov}\left(\frac{\mu_1(X)}{\mathbb{E}\left[Y^{(1)}\right]} - \frac{\mu_0(X)}{\mathbb{E}\left[Y^{(0)}\right]}; \frac{T(Y - \mu_1(X))}{\mathbb{E}\left[Y^{(1)}\right] e(X)} - \frac{(1 - T)(Y - \mu_0(X))}{\mathbb{E}\left[Y^{(0)}\right] (1 - e(X))}\right).$$
(132)

The second term can be rewritten as

$$\text{Var}\left(\frac{T(Y - \mu_1(X))}{\mathbb{E}\left[Y^{(1)}\right] e(X)} - \frac{(1 - T)(Y - \mu_0(X))}{\mathbb{E}\left[Y^{(0)}\right] (1 - e(X))}\right)$$
(133)

$$= \text{Var}\left(\frac{T(Y - \mu_1(X))}{\mathbb{E}\left[Y^{(1)}\right] e(X)}\right) + \text{Var}\left(\frac{(1 - T)(Y - \mu_0(X))}{\mathbb{E}\left[Y^{(0)}\right] (1 - e(X))}\right) - 2\,\text{Cov}\left(\frac{T(Y - \mu_1(X))}{\mathbb{E}\left[Y^{(1)}\right] e(X)}, \frac{(1 - T)(Y - \mu_0(X))}{\mathbb{E}\left[Y^{(0)}\right] (1 - e(X))}\right),$$
(134)

with

$$\text{Var}\left(\frac{T(Y - \mu_1(X))}{\mathbb{E}\left[Y^{(1)}\right] e(X)}\right) = \mathbb{E}\left[\left(\frac{T(Y - \mu_1(X))}{\mathbb{E}\left[Y^{(1)}\right] e(X)}\right)^2\right] - \mathbb{E}\left[\frac{T(Y - \mu_1(X))}{\mathbb{E}\left[Y^{(1)}\right] e(X)}\right]^2.$$
(135)

For the first term in (135),

$$\mathbb{E}\left[\left(T\frac{Y - \mu_1(X)}{e(X)\mathbb{E}\left[Y^{(1)}\right]}\right)^2\right]$$

$$= \mathbb{E}\left[\left(T\frac{Y^{(1)} - \mu_1(X)}{e(X)\mathbb{E}\left[Y^{(1)}\right]}\right)^2\right] \qquad\qquad \text{Consistency}$$

$$= \mathbb{E}\left[\mathbb{E}\left[\left(T\frac{Y^{(1)} - \mu_1(X)}{e(X)\mathbb{E}\left[Y^{(1)}\right]}\right)^2 \mid X\right]\right] \qquad\qquad \text{Total expectation}$$

$$= \mathbb{E}\left[\mathbb{E}\left[T\left(\frac{Y^{(1)} - \mu_1(X)}{e(X)\mathbb{E}\left[Y^{(1)}\right]}\right)^2 \mid X\right]\right] \qquad\qquad \text{T is binary}$$

$$= \mathbb{E}\left[\mathbb{E}\left[\mathbf{1}_{\{T=1\}}\left(\frac{Y^{(1)} - \mu_1(X)}{e(X)\mathbb{E}\left[Y^{(1)}\right]}\right)^2 \mid X\right]\right] \qquad\qquad \text{T written as an indicator}$$

$$= \mathbb{E}\left[\frac{1}{e(X)^2\mathbb{E}\left[Y^{(1)}\right]^2}\mathbb{E}\left[\mathbf{1}_{\{T=1\}}\left(Y^{(1)} - \mu_1(X)\right)^2 \mid X\right]\right] \qquad\qquad e(X) \text{ is a function of } X$$

$$= \mathbb{E}\left[\frac{\text{Var}\left(Y^{(1)}|X\right)}{e(X)^2\mathbb{E}\left[Y^{(1)}\right]^2}\mathbb{E}\left[\mathbf{1}_{\{T=1\}} \mid X\right]\right] \qquad\qquad \text{Uncounf. \& } \mu_1(.) \text{ is func. of } X$$

$$= \mathbb{E}\left[\frac{\text{Var}\left(Y^{(1)}|X\right)}{e(X)^2\mathbb{E}\left[Y^{(1)}\right]^2}e(X)\right] \qquad\qquad \text{Definition of } e(X)$$

$$= \mathbb{E}\left[\frac{\text{Var}\left(Y^{(1)}|X\right)}{e(X)\mathbb{E}\left[Y^{(1)}\right]^2}\right].$$

For the second term in (135),

$$\mathbb{E}\left[\frac{T(Y - \mu_1(X))}{\mathbb{E}\left[Y^{(1)}\right]e(X)}\right]$$

$$= \mathbb{E}\left[\frac{T(Y^{(1)} - \mu_1(X))}{\mathbb{E}\left[Y^{(1)}\right]e(X)}\right] \qquad \text{Consistency}$$

$$= \mathbb{E}\left[\mathbb{E}\left[T\frac{Y^{(1)} - \mu_1(X)}{e(X)\mathbb{E}\left[Y^{(1)}\right]} \mid X\right]\right] \qquad \text{Total expectation}$$

$$= \mathbb{E}\left[\frac{1}{e(X)\mathbb{E}\left[Y^{(1)}\right]}\mathbb{E}\left[T(Y^{(1)} - \mu_1(X)) \mid X\right]\right] \qquad e(X) \text{ is a function of } X$$

$$= \mathbb{E}\left[\frac{e(X)}{e(X)\mathbb{E}\left[Y^{(1)}\right]}(\mu_1(X) - \mu_1(X))\right] \qquad \text{Unconf. \& } \mu_1(.) \text{ is func. of } X$$

$$= 0.$$

Therefore

$$\text{Var}\left(\frac{T(Y - \mu_1(X))}{\mathbb{E}\left[Y^{(1)}\right]e(X)}\right) = \mathbb{E}\left[\frac{\text{Var}\left(Y^{(1)}|X\right)}{e(X)\mathbb{E}\left[Y^{(1)}\right]^2}\right],$$

and similarly

$$\text{Var}\left(\frac{(1 - T)(Y - \mu_0(X))}{\mathbb{E}\left[Y^{(0)}\right](1 - e(X))}\right) = \mathbb{E}\left[\frac{\text{Var}\left(Y^{(0)}|X\right)}{(1 - e(X))\mathbb{E}\left[Y^{(0)}\right]^2}\right].$$

Besides,

$$\text{Cov}\left(\frac{T(Y - \mu_1(X))}{\mathbb{E}\left[Y^{(1)}\right]e(X)}, \frac{(1 - T)(Y - \mu_0(X))}{\mathbb{E}\left[Y^{(0)}\right](1 - e(X))}\right) = \mathbb{E}\left[\frac{T(Y - \mu_1(X))}{\mathbb{E}\left[Y^{(1)}\right]e(X)}\frac{(1 - T)(Y - \mu_0(X))}{\mathbb{E}\left[Y^{(0)}\right](1 - e(X))}\right]$$

$$- \mathbb{E}\left[\frac{T(Y - \mu_1(X))}{\mathbb{E}\left[Y^{(1)}\right]e(X)}\right]\mathbb{E}\left[\frac{(1 - T)(Y - \mu_0(X))}{\mathbb{E}\left[Y^{(0)}\right](1 - e(X))}\right]$$

$$= 0.$$

Gathering all these results into (134), we obtain

$$\text{Var}\left(\frac{T(Y - \mu_1(X))}{\mathbb{E}\left[Y^{(1)}\right]e(X)} - \frac{(1 - T)(Y - \mu_0(X))}{\mathbb{E}\left[Y^{(0)}\right](1 - e(X))}\right) = \mathbb{E}\left[\frac{\text{Var}\left(Y^{(1)}|X\right)}{e(X)\mathbb{E}\left[Y^{(1)}\right]^2}\right] + \mathbb{E}\left[\frac{\text{Var}\left(Y^{(0)}|X\right)}{(1 - e(X))\mathbb{E}\left[Y^{(0)}\right]^2}\right].$$

In order to rewrite the last term in (132), note that

$$\text{Cov}\left(\mu_1(X), \frac{T(Y - \mu_1(X))}{e(X)}\right) = \mathbb{E}\left[\mu_1(X)\frac{T(Y - \mu_1(X))}{e(X)}\right] - \mathbb{E}\left[\mu_1(X)\right]\mathbb{E}\left[\frac{T(Y - \mu_1(X))}{e(X)}\right]$$

$$= \mathbb{E}\left[\mu_1(X)\frac{T(Y - \mu_1(X))}{e(X)}\right]$$

$$= \mathbb{E}\left[\frac{\mu_1(X)}{e(X)}\mathbb{E}\left[T(Y^{(1)} - \mu_1(X))|X\right]\right]$$

$$= \mathbb{E}\left[\frac{\mu_1(X)}{e(X)}\mathbb{E}\left[\frac{T(Y^{(1)} - \mu_1(X))}{e(X)}|X\right]\right]$$

$$= \mathbb{E}\left[\frac{\mu_1(X)e(X)}{e(X)}(\mu_1(X)) - \mu_1(X))\right]$$

$$= 0.$$

Similar calculations leads to

$$\text{Cov}\left(\frac{\mu_1(X)}{\mathbb{E}\left[Y^{(1)}\right]} - \frac{\mu_0(X)}{\mathbb{E}\left[Y^{(0)}\right]}; \frac{T(Y - \mu_1(X))}{\mathbb{E}\left[Y^{(1)}\right]e(X)} - \frac{(1-T)(Y - \mu_0(X))}{\mathbb{E}\left[Y^{(0)}\right](1 - e(X))}\right) = 0.$$

Finally

$$V_{\text{RR,OS}} = \tau_{\text{RR}}^2\left(\text{Var}\left(\frac{\mu_1(X)}{\mathbb{E}\left[Y^{(1)}\right]} - \frac{\mu_0(X)}{\mathbb{E}\left[Y^{(0)}\right]}\right) + \mathbb{E}\left[\frac{\text{Var}\left(Y^{(1)}|X\right)}{e(X)\mathbb{E}\left[Y^{(1)}\right]^2}\right] + \mathbb{E}\left[\frac{\text{Var}\left(Y^{(0)}|X\right)}{(1 - e(X))\mathbb{E}\left[Y^{(0)}\right]^2}\right]\right).$$

An estimator $\hat{V}_{RR,OS}$ can be derived as follows:

$$\hat{V}_{RR,OS} = \frac{\hat{\tau}_{\text{RR,OS,n}}^2}{n}\sum_{i=1}^{n}\left(\Delta_i - \frac{1}{n}\sum_{i=1}^{n}\Delta_i\right)^2 \tag{136}$$

where

$$\Delta_i = \frac{\hat{\Gamma}_i(1)}{\hat{S}(1)} - \frac{\hat{\Gamma}_i(0)}{\hat{S}(0)}$$

with the intermediate for $t \in \{0, 1\}$ quantities defined as:

$$\hat{\Gamma}_i(t) = \hat{\mu}_t(X_i) + \mathbb{1}_{T_i=t}\frac{Y_i - \hat{\mu}_t(X_i)}{\hat{e}_t(X_i)} \quad \text{and} \quad \hat{S}(t) = \frac{1}{n}\sum_{j=1}^{n}\hat{\Gamma}_j(t)$$

$\square$

### 7.3.6. RISK RATIO AUGMENTED INVERSE PROPENSITY WEIGHTING

*Proof of Definition 3.12.* We use the derivations established in the proof of Proposition 3.11. Indeed, we showed in Section 7.3.5 that the influence function $\varphi$ of $\psi = \frac{\mathbb{E}[\mathbb{E}[Y|T=1,X]]}{\mathbb{E}[\mathbb{E}[Y|T=0,X]]}$ can be written:

$$\varphi(Z; P) = \frac{\mu_1(X) + T\frac{Y - \mu_1(X)}{e(X)}}{\psi_0} - \psi\frac{\mu_0(X) + (1-T)\frac{Y - \mu_0(X)}{1 - e(X)}}{\psi_0}.$$

Using Equation (85), and knowing that $\int \varphi(z; \hat{P})d\hat{P}(z) = 0$, we have:

$$\psi(\hat{P}) - \psi(P) = \int \varphi(z; \hat{P})d(\hat{P} - P)(z) + R_2(\hat{P}, P)$$

$$= R_2(P, \hat{P}) - \int \varphi(z; \hat{P})dP(z)$$

$$= \frac{1}{n}\sum_{i=1}^{n}\varphi(Z_i; P) - \frac{1}{n}\sum_{i=1}^{n}\varphi(z_i; \hat{P})$$

$$+ \frac{1}{n}\sum_{i=1}^{n}\left(\varphi(Z_i; \hat{P}) - \varphi(Z_i; P)\right) - \int\left(\varphi(z; \hat{P}) - \varphi(z; P)\right)dP(z)$$

$$+ R_2(P, \hat{P}).$$

As outlined in (A.Schuler, 2024), in the estimating equation approach, we assume that the efficient influence function for any given distribution depends solely on the target parameter $\psi$ and a set of nuisance parameters $\eta$. Therefore, instead

of denoting the efficient influence function as $\varphi(z; P)$, we can express it as $\varphi(Z; \psi, \eta)$. If the influence function can be represented in this form, we proceed by first estimating $\hat{\eta} = (\hat{e}, \hat{\mu}_1, \hat{\mu}_0)$ with crossfitting. For any fixed value $\hat{\eta}$, we find a value $\hat{\psi}$ such that $P_n \varphi_{\hat{\psi}, \hat{\eta}} = 0$, that is

$$\frac{1}{n} \sum_{i=1}^{n} \frac{\hat{\mu}_1(X_i) + T_i \frac{Y_i - \hat{\mu}_1(X_i)}{\hat{e}(X_i)}}{\hat{\psi}_0} - \hat{\psi} \frac{\hat{\mu}_0(X_i) + (1 - T_i) \frac{Y_i - \hat{\mu}_0(X_i)}{1 - \hat{e}(X_i)}}{\hat{\psi}_0} = 0,$$

which implies

$$\hat{\psi} = \frac{\sum_{i=1}^{n} \hat{\mu}_1(X_i) + T_i \frac{Y_i - \hat{\mu}_1(X_i)}{\hat{e}(X_i)}}{\sum_{i=1}^{n} \hat{\mu}_0(X_i) + (1 - T_i) \frac{Y_i - \hat{\mu}_0(X_i)}{1 - \hat{e}(X_i)}}.$$

$\square$

Hereafter, we propose a proof of Proposition 3.13 which does not use the influence function theory

*Proof of Proposition 3.13.*
**Asymptotic bias and variance of the crossfitted Ratio AIPW estimator** In this alternative proof, we further assume that $\mathrm{Var}[Y|X] \le \sigma^2$ for some $\sigma > 0$. Recall that we want to analyze $\sqrt{n} \left( \hat{\tau}_{\mathrm{RR,AIPW}} - \tau_{\mathrm{RR}} \right)$. Letting

$$\tau_{\mathrm{RR,AIPW}}^{\star} = \frac{\sum_{i=1}^{n} \mu_1(X_i) + \frac{T_i(Y_i - \mu_1(X_i))}{e(X_i)}}{\sum_{i=1}^{n} \mu_0(X_i) + \frac{(1 - T_i)(Y_i - \mu_0(X_i))}{1 - e(X_i)}} := \frac{\tau_{\mathrm{RR,AIPW,1}}^{\star}}{\tau_{\mathrm{RR,AIPW,0}}^{\star}} \tag{137}$$

be the oracle version of $\hat{\tau}_{\mathrm{RR,AIPW}}$ where the propensity score and both response surfaces are assumed to be known, we can rewrite

$$\hat{\tau}_{\mathrm{RR,AIPW}} - \tau_{\mathrm{RR}} = \hat{\tau}_{\mathrm{RR,AIPW}} - \tau_{\mathrm{RR,AIPW}}^{\star} + \tau_{\mathrm{RR,AIPW}}^{\star} - \tau_{\mathrm{RR}}. \tag{138}$$

Regarding the first term in (138), we have

$$\left| \hat{\tau}_{\mathrm{RR,AIPW}} - \tau_{\mathrm{RR,AIPW}}^{\star} \right| \tag{139}$$

$$= \left| \frac{\hat{\tau}_{\mathrm{RR,AIPW,1}}}{\hat{\tau}_{\mathrm{RR,AIPW,0}}} - \frac{\tau_{\mathrm{RR,AIPW,1}}^{\star}}{\tau_{\mathrm{RR,AIPW,0}}^{\star}} \right| \tag{140}$$

$$= \left| \left( (\hat{\tau}_{\mathrm{RR,AIPW,0}})^{-1} - (\tau_{\mathrm{RR,AIPW,0}}^{\star})^{-1} \right) \hat{\tau}_{\mathrm{RR,AIPW,1}} \right. \tag{141}$$

$$\left. + (\tau_{\mathrm{RR,AIPW,0}}^{\star})^{-1} \left( \hat{\tau}_{\mathrm{RR,AIPW,1}} - \tau_{\mathrm{RR,AIPW,1}}^{\star} \right) \right| \tag{142}$$

$$\le \left| \left( (\hat{\tau}_{\mathrm{RR,AIPW,0}})^{-1} - (\tau_{\mathrm{RR,AIPW,0}}^{\star})^{-1} \right) \hat{\tau}_{\mathrm{RR,AIPW,1}} \right| \tag{143}$$

$$+ \left| (\tau_{\mathrm{RR,AIPW,0}}^{\star})^{-1} \left( \hat{\tau}_{\mathrm{RR,AIPW,1}} - \tau_{\mathrm{RR,AIPW,1}}^{\star} \right) \right|. \tag{144}$$

We now show that

$$\left| \hat{\tau}_{\mathrm{RR,AIPW,1}} - \tau_{\mathrm{RR,AIPW,1}}^{\star} \right| = o_p \left( \frac{1}{\sqrt{n}} \right) \quad \text{and} \quad \left| \hat{\tau}_{\mathrm{RR,AIPW,0}} - \tau_{\mathrm{RR,AIPW,0}}^{\star} \right| = o_p \left( \frac{1}{\sqrt{n}} \right).$$

The following decomposition holds

$$\sqrt{n} \left| \hat{\tau}_{\mathrm{RR,AIPW,1}} - \tau_{\mathrm{RR,AIPW,1}}^{\star} \right| = \frac{1}{\sqrt{n}} \sum_{i \in \mathcal{I}_k} \left( \hat{\mu}_1^{\mathcal{I}_{-k}}(X_i) + T_i \frac{Y_i - \hat{\mu}_1^{\mathcal{I}_{-k}}(X_i)}{\hat{e}(X_i)} - \mu_1(X_i) - T_i \frac{Y_i - \mu_1(X_i)}{e(X_i)} \right)$$

$$\text{Further denoted } A_n^k \qquad = \frac{1}{\sqrt{n}} \sum_{i \in \mathcal{I}_k} \left( \left( \hat{\mu}_1^{\mathcal{I}_{-k}}(X_i) - \mu_1(X_i) \right) \left( 1 - \frac{T_i}{e(X_i)} \right) \right)$$

$$\text{Further denoted } B_n^k \qquad + \frac{1}{\sqrt{n}} \sum_{i \in \mathcal{I}_k} T_i \left( (Y_i - \mu_1(X_i)) \left( \frac{1}{\hat{e}(X_i)} - \frac{1}{e(X_i)} \right) \right)$$

$$\text{Further denoted } C_n^k \qquad - \frac{1}{\sqrt{n}} \sum_{i \in \mathcal{I}_k} T_i \left( \left( \hat{\mu}_1^{\mathcal{I}_{-k}}(X_i) - \mu_1(X_i) \right) \left( \frac{1}{\hat{e}(X_i)} - \frac{1}{e(X_i)} \right) \right).$$

In the following, we prove that the first two terms tend to zero in $L^2$.

**Regarding $A_n^k$** One can show that the expectation of $A_n^k / \sqrt{n}$ is null:

$$
\begin{aligned}
\mathbb{E}\left[\frac{A_n^k}{\sqrt{n}} \mid \mathcal{I}_{-k}\right] &= \frac{1}{n} \sum_{i \in \mathcal{I}_k} \mathbb{E}\left[\left(\hat{\mu}_1^{\mathcal{I}_{-k}}(X_i) - \mu_1(X_i)\right)\left(1 - \frac{T_i}{e(X_i)}\right) \mid \mathcal{I}_{-k}\right] \\
&= \frac{|\mathcal{I}_k|}{n} \mathbb{E}\left[\left(\hat{\mu}_1^{\mathcal{I}_{-k}}(X) - \mu_1(X)\right)\left(1 - \frac{T}{e(X)}\right) \mid \mathcal{I}_{-k}\right] \qquad \text{i.i.d.} \\
&= \frac{|\mathcal{I}_k|}{n} \mathbb{E}\left[\mathbb{E}\left[\left(\hat{\mu}_1^{\mathcal{I}_{-k}}(X) - \mu_1(X)\right)\left(1 - \frac{T}{e(X)}\right) \mid X, \mathcal{I}_{-k}\right] \mid \mathcal{I}_{-k}\right] \\
&= \frac{|\mathcal{I}_k|}{n} \mathbb{E}\left[\left(\hat{\mu}_1^{\mathcal{I}_{-k}}(X) - \mu_1(X)\right) \mathbb{E}\left[\left(1 - \frac{T}{e(X)}\right) \mid X, \mathcal{I}_{-k}\right] \mid \mathcal{I}_{-k}\right] \\
&= \frac{|\mathcal{I}_k|}{n} \mathbb{E}\left[\left(\hat{\mu}_1^{\mathcal{I}_{-k}}(X) - \mu_1(X)\right)\left(1 - \frac{e(X)}{e(X)}\right) \mid \mathcal{I}_{-k}\right] \\
&= 0.
\end{aligned}
$$

We will make use of this results in several calculations. Now,

$$
\begin{aligned}
\mathbb{E}\left[\left(\frac{A_n^k}{\sqrt{n}}\right)^2 \mid \mathcal{I}_{-k}\right] &= \operatorname{Var}\left[\frac{1}{n} \sum_{i \in \mathcal{I}_k}\left(\left(\hat{\mu}_1^{\mathcal{I}_{-k}}(X_i) - \mu_1(X_i)\right)\left(1 - \frac{T_i}{e(X_i)}\right)\right) \mid \mathcal{I}_{-k}\right] \\
&= \frac{1}{n^2} \operatorname{Var}\left[\sum_{i \in \mathcal{I}_k}\left(\left(\hat{\mu}_1^{\mathcal{I}_{-k}}(X_i) - \mu_1(X_i)\right)\left(1 - \frac{T_i}{e(X_i)}\right)\right) \mid \mathcal{I}_{-k}\right] \\
&= \frac{1}{n^2} \sum_{i \in \mathcal{I}_k} \operatorname{Var}\left[\left(\hat{\mu}_1^{\mathcal{I}_{-k}}(X_i) - \mu_1(X_i)\right)\left(1 - \frac{T_i}{e(X_i)}\right) \mid \mathcal{I}_{-k}\right] \qquad \text{iid} \\
&= \frac{|\mathcal{I}_k|}{n^2} \mathbb{E}\left[\left(\left(\hat{\mu}_1^{\mathcal{I}_{-k}}(X) - \mu_1(X)\right)\left(1 - \frac{T}{e(X)}\right)\right)^2 \mid \mathcal{I}_{-k}\right] \\
&= \frac{|\mathcal{I}_k|}{n^2} \mathbb{E}\left[\mathbb{E}\left[\left(\left(\hat{\mu}_1^{\mathcal{I}_{-k}}(X) - \mu_1(X)\right)\left(1 - \frac{T}{e(X)}\right)\right)^2 \mid X, \mathcal{I}_{-k}\right] \mid \mathcal{I}_{-k}\right] \\
&= \frac{|\mathcal{I}_k|}{n^2} \mathbb{E}\left[\left(\hat{\mu}_1^{\mathcal{I}_{-k}}(X) - \mu_1(X)\right)^2 \mathbb{E}\left[\left(1 - \frac{T}{e(X)}\right)^2 \mid X, \mathcal{I}_{-k}\right] \mid \mathcal{I}_{-k}\right] \\
&= \frac{|\mathcal{I}_k|}{n^2} \mathbb{E}\left[\left(\hat{\mu}_1^{\mathcal{I}_{-k}}(X) - \mu_1(X)\right)^2 \frac{1}{e(X)^2} \mathbb{E}\left[(e(X) - T)^2 \mid X, \mathcal{I}_{-k}\right] \mid \mathcal{I}_{-k}\right] \\
&= \frac{|\mathcal{I}_k|}{n^2} \mathbb{E}\left[\left(\hat{\mu}_1^{\mathcal{I}_{-k}}(X) - \mu_1(X)\right)^2 \frac{e(X)(1 - e(X))}{e(X)^2} \mid \mathcal{I}_{-k}\right] \\
&= \frac{|\mathcal{I}_k|}{n^2} \mathbb{E}\left[\left(\hat{\mu}_1^{\mathcal{I}_{-k}}(X) - \mu_1(X)\right)^2 \left(\frac{1}{e(X)} - 1\right) \mid \mathcal{I}_{-k}\right] \\
&\leq \frac{|\mathcal{I}_k|}{\eta n^2} \mathbb{E}\left[\left(\hat{\mu}_1^{\mathcal{I}_{-k}}(X) - \mu_1(X)\right)^2 \mid \mathcal{I}_{-k}\right] \qquad \text{Overlap.}
\end{aligned}
$$

Taking the expectation, we obtain

$$
\mathbb{E}\left[\left(\frac{A_n^k}{\sqrt{n}}\right)^2\right] \leq \frac{|\mathcal{I}_k|}{\eta n^2} \mathbb{E}\left[\left(\hat{\mu}_1^{\mathcal{I}_{-k}}(X) - \mu_1(X)\right)^2\right], \tag{145}
$$

that is

$$
\mathbb{E}\left[\left(A_n^k\right)^2\right] \leq \frac{1}{\eta} \mathbb{E}\left[\left(\hat{\mu}_1^{\mathcal{I}_{-k}}(X) - \mu_1(X)\right)^2\right]. \tag{146}
$$

Thus $A_n^k$ converges to zero in $L^2$ and thus in probability.

**Regarding $B_n^k$** The second term $B_n^k$ can also be controlled using similar arguments. By assumption,

$$\frac{\eta}{2} \leq \hat{e}(X) \leq 1 - \frac{\eta}{2}.$$

Thus,

$$\frac{1}{\hat{e}(X)} - \frac{1}{e(X)} = \frac{e(X) - \hat{e}(X)}{\hat{e}(X)e(X)} \leq 2\left(\frac{e(X) - \hat{e}(X)}{\eta^2}\right).$$

Derivations are very close to the ones for the first term, noting that,

$$\mathbb{E}\left[\mathbb{E}\left[\frac{1}{n}\sum_{i\in\mathcal{I}_k}T_i\left((Y_i - \mu_1(X_i))\left(\frac{1}{\hat{e}^{\mathcal{I}_{-k}}(X_i)} - \frac{1}{e(X_i)}\right)\right) \mid X_i, \mathcal{I}_{-k}\right] \mid \mathcal{I}_{-k}\right] = 0,$$

so that,

$$
\begin{aligned}
\mathbb{E}\left[\left(\frac{B_n^k}{\sqrt{n}}\right)^2 \mid \mathcal{I}_{-k}\right] &= \mathrm{Var}\left[\frac{1}{n}\sum_{i\in\mathcal{I}_k}T_i\left(Y_i - \mu_1(X_i)\right)\left(\frac{1}{\hat{e}^{\mathcal{I}_{-k}}(X_i)} - \frac{1}{e(X_i)}\right) \mid \mathcal{I}_{-k}\right] \\
&= \frac{1}{n^2}\sum_{i\in\mathcal{I}_k}\mathrm{Var}\left[T_i\left(Y_i - \mu_1(X_i)\right)\left(\frac{1}{\hat{e}^{\mathcal{I}_{-k}}(X_i)} - \frac{1}{e(X_i)}\right) \mid \mathcal{I}_{-k}\right] && \text{iid} \\
&= \frac{|\mathcal{I}_k|}{n^2}\mathbb{E}\left[T\left(Y - \mu_1(X)\right)^2\left(\frac{1}{\hat{e}^{\mathcal{I}_{-k}}(X)} - \frac{1}{e(X)}\right)^2 \mid \mathcal{I}_{-k}\right] \\
&\leq \frac{4|\mathcal{I}_k|}{\eta^4 n^2}\mathbb{E}\left[T\left(Y - \mu_1(X)\right)^2\left(\hat{e}^{\mathcal{I}_{-k}}(X) - e(X)\right)^2 \mid \mathcal{I}_{-k}\right] \\
&\leq \frac{4|\mathcal{I}_k|}{\eta^4 n^2}\mathbb{E}\left[\left(Y - \mu_1(X)\right)^2\left(\hat{e}^{\mathcal{I}_{-k}}(X) - e(X)\right)^2 \mid \mathcal{I}_{-k}\right] && \text{Sicne } T \leq 1 \\
&\leq \frac{4|\mathcal{I}_k|}{\eta^4 n^2}\mathbb{E}\left[\mathbb{E}\left[\left(Y - \mu_1(X)\right)^2\left(\hat{e}^{\mathcal{I}_{-k}}(X) - e(X)\right)^2 \mid X, \mathcal{I}_{-k}\right] \mid \mathcal{I}_{-k}\right] \\
&\leq \frac{4|\mathcal{I}_k|}{\eta^4 n^2}\mathbb{E}\left[\mathbb{E}\left[\left(Y - \mu_1(X)\right)^2 \mid X, \mathcal{I}_{-k}\right]\left(\hat{e}^{\mathcal{I}_{-k}}(X) - e(X)\right)^2 \mid \mathcal{I}_{-k}\right] \\
&\leq \frac{4|\mathcal{I}_k|}{\eta^4 n^2}\mathbb{E}\left[\mathrm{Var}\left[Y|X\right]\left(\hat{e}^{\mathcal{I}_{-k}}(X) - e(X)\right)^2 \mid \mathcal{I}_{-k}\right] \\
&\leq \frac{4\,\mathrm{Var}\left[Y|X\right]|\mathcal{I}_k|}{\eta^4 n^2}\mathbb{E}\left[\left(\hat{e}^{\mathcal{I}_{-k}}(X) - e(X)\right)^2 \mid \mathcal{I}_{-k}\right].
\end{aligned}
$$

Taking the expectation on both sides, since $\mathrm{Var}\left[Y|X\right] \leq \sigma^2$, we get

$$\mathbb{E}\left[\left(\frac{B_n^k}{\sqrt{n}}\right)^2\right] \leq \frac{4\sigma^2|\mathcal{I}_k|}{\eta^4 n^2}\mathbb{E}\left[\left(\hat{e}^{\mathcal{I}_{-k}}(X) - e(X)\right)^2 \mid \mathcal{I}_{-k}\right], \tag{147}$$

which leads to

$$\mathbb{E}\left[\left(B_n^k\right)^2\right] \leq \frac{4\sigma^2}{\eta^4}\mathbb{E}\left[\left(\hat{e}^{\mathcal{I}_{-k}}(X) - e(X)\right)^2 \mid \mathcal{I}_{-k}\right]. \tag{148}$$

Since, by assumption, the right-hand side term converges to zero, $B_n^k$ converges to zero in $L^2$.

**Regarding $C_n^k$** Regarding the last term, the approach is different and will involve another assumption on the product of residuals. More precisely,

$$
\begin{aligned}
\mathbb{E}[|C_n^k|] &= \sqrt{n}\frac{1}{n}\sum_{i\in\mathcal{I}_k}\mathbb{E}\left[\left|T_i\left(\hat{\mu}_1^{\mathcal{I}_{-k}}(X_i)-\mu_1(X_i)\right)\left(\frac{1}{\hat{e}^{\mathcal{I}_{-k}}(X_i)}-\frac{1}{e(X_i)}\right)\right|\right]\\
&= \frac{\sqrt{n}}{\eta^2}\frac{1}{n}\sum_{i\in\mathcal{I}_k}\mathbb{E}\left[\left|T_i\left(\hat{\mu}_1^{\mathcal{I}_{-k}}(X_i)-\mu_1(X_i)\right)\left(e(X_i)-\hat{e}^{\mathcal{I}_{-k}}(X_i)\right)\right|\right]\\
&= \frac{\sqrt{n}|\mathcal{I}_k|}{\eta^2}\frac{1}{n}\mathbb{E}\left[\left|T\left(\hat{\mu}_1^{\mathcal{I}_{-k}}(X)-\mu_1(X)\right)\left(e(X)-\hat{e}^{\mathcal{I}_{-k}}(X)\right)\right|\right]\\
&\leq \frac{\sqrt{n}}{\eta^2}\mathbb{E}\left[\left|\left(\hat{\mu}_1^{\mathcal{I}_{-k}}(X)-\mu_1(X)\right)\left(e(X)-\hat{e}^{\mathcal{I}_{-k}}(X)\right)\right|\right]\\
&\leq \frac{\sqrt{n}}{\eta^2}\sqrt{\mathbb{E}\left[\left(\hat{\mu}_1^{\mathcal{I}_{-k}}(X)-\mu_1(X)\right)^2\right]\mathbb{E}\left[\left(e(X)-\hat{e}^{\mathcal{I}_{-k}}(X)\right)^2\right]},
\end{aligned}
$$

which tends to zero by assumption. Each term $A_n^k$, $B_n^k$, and $C_n^k$ has been shown to be bounded by a term in $o_{\mathbb{P}}(1)$. Thus,

$$
\sqrt{n}\left|\hat{\tau}_{\text{RR,AIPW, }1}-\tau^{\star}_{\text{RR,AIPW, }1}\right| = \sum_{k=1}^{K}A_n^k+B_n^k+C_n^k \tag{149}
$$

tends to zero in probability. Similarly, one can show that

$$
\sqrt{n}\left|\hat{\tau}_{\text{RR,AIPW, }1}-\tau^{\star}_{\text{RR,AIPW, }1}\right| \xrightarrow{p} 0. \tag{150}
$$

According to (144), since for all $t\in\{0,1\}$, $|\hat{\tau}_{\text{RR,AIPW,t}}|$ tends to $\tau^{\star}_{\text{RR,AIPW,t}}$ which is lower and upper bounded, we have

$$
\begin{aligned}
\sqrt{n}\left|\hat{\tau}_{\text{RR,AIPW}}-\tau^{\star}_{\text{RR,AIPW}}\right| &\leq \sqrt{n}\left|\frac{\hat{\tau}_{\text{RR,AIPW, }1}}{\hat{\tau}_{\text{RR,AIPW, }0}\tau^{\star}_{\text{RR,AIPW, }0}}\right|\left|\hat{\tau}_{\text{RR,AIPW, }0}-\tau^{\star}_{\text{RR,AIPW, }0}\right|\\
&\quad+\sqrt{n}\left|\frac{1}{\tau^{\star}_{\text{RR,AIPW, }0}}\right|\left|\hat{\tau}_{\text{RR,AIPW, }1}-\tau^{\star}_{\text{RR,AIPW, }1}\right|
\end{aligned}
$$

which tends to zero.

Regarding the second term in (138), we can use Theorem 7.1 with $g_1(Z)=\mu_1(X)+\frac{T(Y-\mu_1(X))}{e(X)}$ and $g_0(Z)=\mu_0(X)+\frac{(1-T)(Y-\mu_0(X))}{(1-e(X))}$ where $Z=(T,X,Y)$. Hence, we have that $g_1(Z)$ is square integrable:

$$
\mathbb{E}\left[g_1(Z)^2\right] \leq 2\mathbb{E}\left[(\mu_1(X)^2\right]+2\mathbb{E}\left[\left(\frac{T(Y-\mu_1(X))}{e(X)}\right)^2\right],
$$

where $\mathbb{E}\left[(\mu_1(X)^2\right]=\text{Var}(Y^{(1)})+\mathbb{E}\left[Y^{(1)}\right]^2$ is finite. Using Consistency, Unconfoundedness, and definition or $\mu_1(X)=\mathbb{E}[Y\mid X,T=1]$, simple calculations show that

$$
\begin{aligned}
\mathbb{E}\left[\left(T\frac{Y-\mu_1(X)}{e(X)}\right)^2\right] &= \mathbb{E}\left[\left(T\frac{Y^{(1)}-\mu_1(X)}{e(X)}\right)^2\right] && \text{Consistency}\\
&= \mathbb{E}\left[\mathbb{E}\left[\left(T\frac{Y^{(1)}-\mu_1(X)}{e(X)}\right)^2\mid X\right]\right] && \text{Total expectation}\\
&= \mathbb{E}\left[\frac{\left(Y^{(1)}-\mu_1(X)\right)^2}{e(X)}\right]\\
&\leq \frac{\text{Var}(\mu_1(X))}{\eta}.
\end{aligned}
$$

Similarly, we can show that $g_0(Z)$ is square integrable. Since $\mathbb{E}\left[g_0(Z)\right] = \mathbb{E}\left[Y^{(0)}\right]$ and $\mathbb{E}\left[g_1(Z)\right] = \mathbb{E}\left[Y^{(1)}\right]$, we can apply Theorem 7.1 and conclude that

$$\sqrt{n}(\tau^{\star}_{\text{RR,AIPW}} - \tau_{\text{RR}}) \to \mathcal{N}(0, V_{\text{RR,OS}}). \tag{151}$$

Finally,

$$\sqrt{n}(\hat{\tau}_{\text{AIPW}} - \tau_{\text{RR}}) = \underbrace{\sqrt{n}(\hat{\tau}_{\text{RR,AIPW}} - \tau^{\star}_{\text{RR,AIPW}})}_{\xrightarrow{P} 0} + \underbrace{\sqrt{n}(\tau^{\star}_{\text{RR,AIPW}} - \tau_{\text{RR}})}_{\xrightarrow{d} \mathcal{N}\left(0, V_{\text{RR,OS}}\right)},$$

where

$$V_{\text{RR,OS}} = \tau^2_{\text{RR}}\left(\text{Var}\left(\frac{\mu_1(X)}{\mathbb{E}\left[Y^{(1)}\right]} - \frac{\mu_0(X)}{\mathbb{E}\left[Y^{(0)}\right]}\right) + \mathbb{E}\left[\frac{\text{Var}\left(Y^{(1)}|X\right)}{e(X)\mathbb{E}\left[Y^{(1)}\right]^2}\right] + \mathbb{E}\left[\frac{\text{Var}\left(Y^{(0)}|X\right)}{(1 - e(X))\mathbb{E}\left[Y^{(0)}\right]^2}\right]\right)$$

$\square$

# 8. Simulation

For the simulations we have implemented all estimators in Python using Scikit-Learn for our regression and classification models. All our experiments were run on a 8GB M1 Mac. The propensity scores is estimated based on the provided training data and covariate names. Depending on the chosen method, it either uses logistic regression with a high regularization parameter (parametric) or a random forest classifier with parameters determined by the training data size (non-parametric). The response surface is estimated based on the training data, covariate names, the method (parametric or non-parametric), and whether the response is binary or continuous. For parametric methods, it uses a stochastic gradient descent classifier for binary responses and a linear regression model for continuous responses. For non-parametric methods, it employs a random forest classifier for binary responses and a random forest regressor for continuous responses. Both methods fit the model using the training data to estimate the respective scores and surfaces, enabling flexible handling of various datasets and assumptions for causal inference analysis.

## 8.1. Randomized Controlled Trials

In this part we will simulate Randomized Controlled Trials (RCT) and test the following Ratio estimators: Ratio Neyman, Ratio Horvitz Thomson and the Ratio G-formula. Since we are in a Randomized Controlled Trials, the propensity score $e(.)$ is constant.

### 8.1.1. Linear RCT

The first DGP has linear outcome models (linear treatment effect and the baseline). The data is generated using:

$$m(X) = (c_1 - c_0) + (\beta_1 - \beta_0)^\top X \qquad c_0 = 6, \qquad c_1 = 12$$
$$b(X) = c_0 + \beta_0^\top X \qquad\qquad \beta_1 = (2, -5, 2, 8, -2, 8)$$
$$e(X) = 0.5 \qquad\qquad\qquad \beta_0 = (3, -7, 1, 4, -2, 2)$$

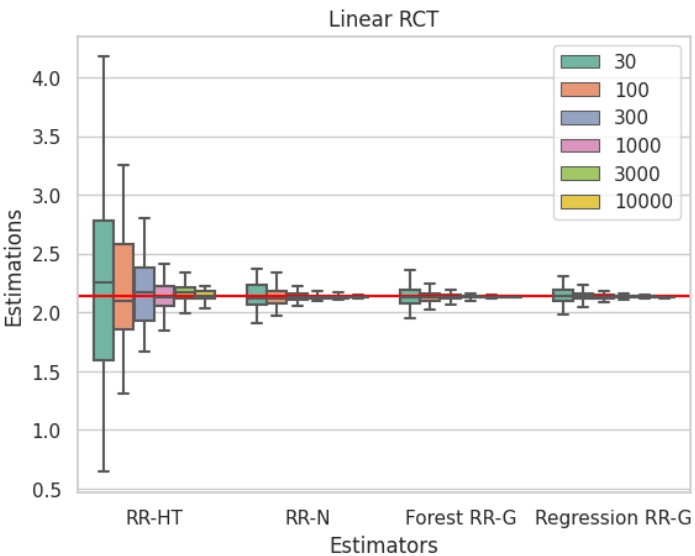

*Figure 6.* Comparison of RCT estimators in a Linear RCT

Given that $X$ has a zero mean, it follows that $\tau_{\mathrm{RR}} = c_1/c_0 = 2$. This scenario aligns with the linear setting outlined in Assumption 3.8. Referring to Figure 6, as proved in the previous sections all estimators converge to the true Risk Ratio as $n$ increases. Additionally, within this linear framework as per Lemma 7.9, the variance of the Neyman estimator exceeds the one of the G-formula. In such a linear environment, the parametric G-formula performs better than its non-parametric counterpart. Additionally, the Ratio Neyman estimator demonstrates lower variance compared to the Horvitz-Thomson estimator as indicated in Equation (5).

### 8.1.2. NON-LINEAR RCT

This DGP is also a Randomized Controlled Trials however, the outcomes are not linear this time:

$$m(X) = \sin(X_1) \cdot X_2^2 + \frac{X_3}{X_4 + 1} - \log(X_5 + 1) + X_6^3 + 1$$

$$b(X) = 4 * \max(X_1 + X_2 + X_3, 0) - \min(X_4 + X_6, 0) \quad \text{and} \quad e(X) = 0.5$$

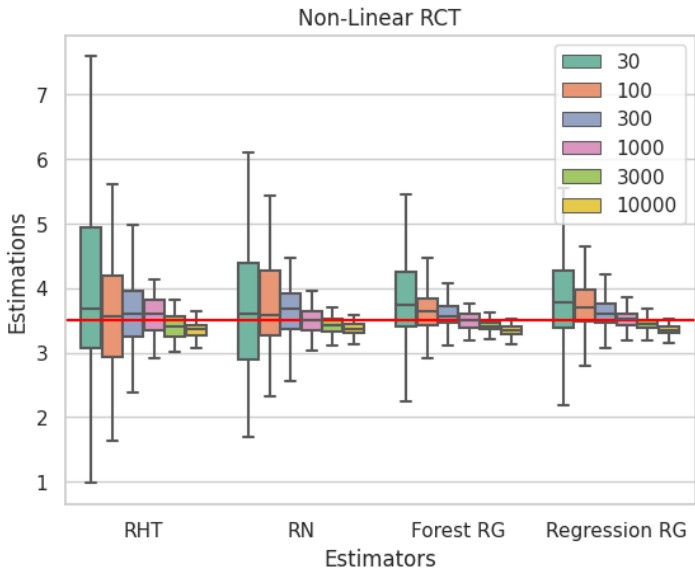

Figure 7. Comparison of RCT estimators in a Non-Linear RCT

The presence of trigonometric, exponential, logarithmic, and polynomial terms makes this setting non-linear. It's important to note that since we are in a Randomized Controlled Trial (RCT), the propensity function remains constant. As the sample size $(n)$ increases, all proposed estimators converge. A bias can be seen in 7 but decreases to 0 as $(n)$ increases as predicted in previous sections. Linear regression struggles with small $n$ values, failing to capture the intricate relationships between features and non-linearities. On the other hand, Random Forest, a non-parametric method, excels in capturing these complexities by segmenting the feature space and predicting based on response averages within those segments. However, predicting the complex function can be challenging, the Neyman estimator might outperform the G-formula, particularly when both parametric and non-parametric responses may lack consistency. Although we do not fall in assumptions of Equation (5) the Ratio Neyman estimator demonstrates lower variance compared to the Horvitz-Thomson estimator.

## 8.2. Observational Studies

### 8.2.1. NON-LINEAR AND NON-LOGISTIC DGP

We use the same simulations as in (Nie & Wager, 2020) using nonlinear models for every quantity, as detailed below, with $X \sim \text{Unif}(0, 1)^6$

$$m(X) = \sin\left(\pi X_1 X_2\right) + 2\left(X_3 - 0.5\right)^2 + X_4 + 0.5X_5 - \left(X_1 + X_2\right)/4$$
$$b(X) = \left(X_1 + X_2\right)/2$$
$$e(X) = \max\{0.1, \min(\sin\left(\pi X_1\right), 0.9)\}.$$

Results are presented in Figure 9. At first glance, all methods seem to have similar performances. However, estimators based on parametric estimators (last four) fail to converge to the correct quantity. They present an intrinsic bias, which does

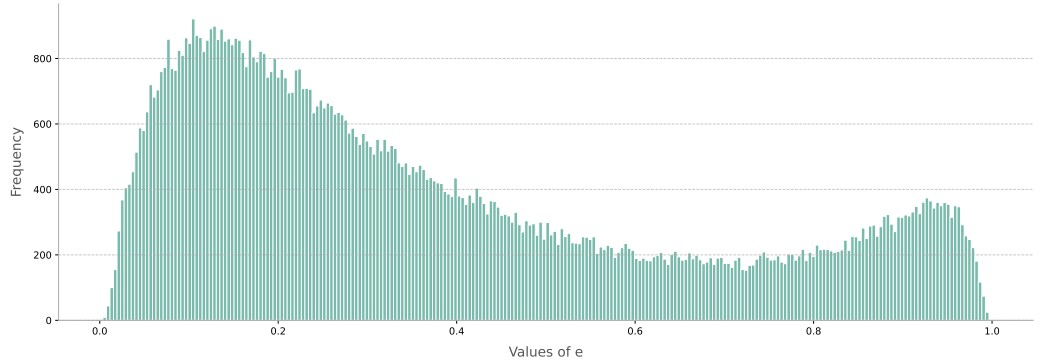

*Figure 8.* Histogram of the propensity score of Logistic DGPs

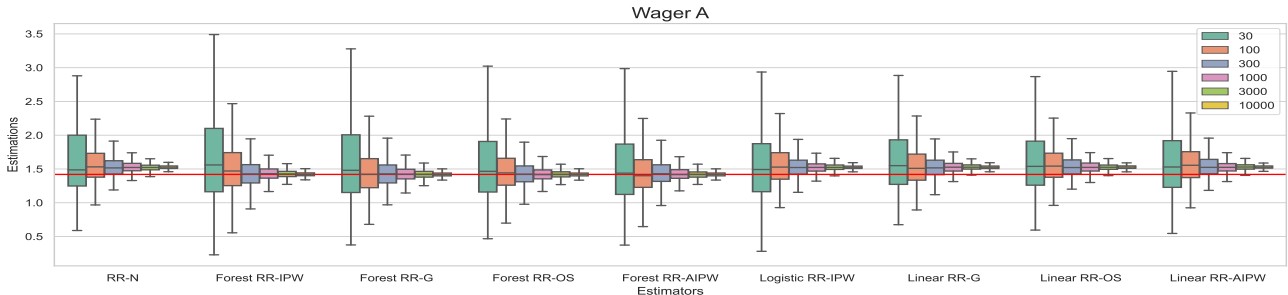

*Figure 9.* Estimations of the Risk Ratio with weighting, outcome based and augmented estimators as a function of the sample size for the non-Linear-non-Logistic DGP. Parametric (Regression) and non parametric (Forest) estimations of nuisance are displayed.

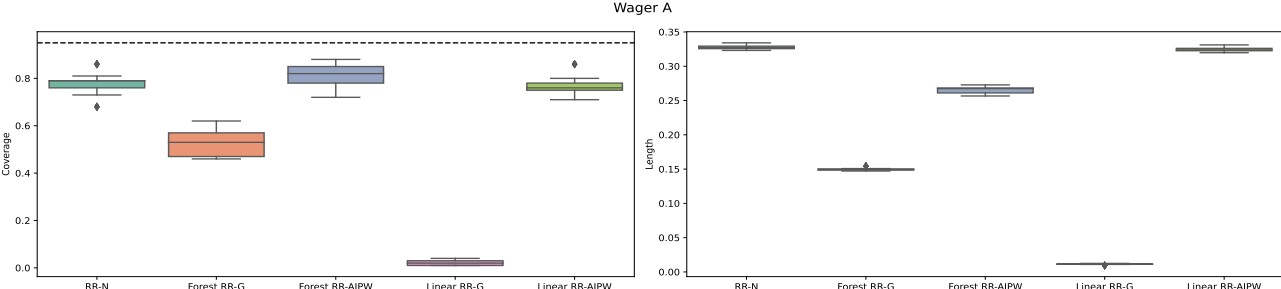

*Figure 10.* Average coverage (left) and average length (right) of asymptotic confidence interval derived from Section 2 and Section 3 for different estimators with $n = 1000$ and 300 repetitions for a Non-Linear and Non-Logistic DGP.

not vanish as the sample size increases. This was expected as linear methods are unable to model the complex non-linear generative process of this simulation. On the other hand, methods that employ random forests estimators achieve good performances: they are consistent and unbias even for small sample sizes. Note that RIPW has a larger variance than the other methods, with a small bias for very small sample sizes. Therefore, the G-formula and the two doubly-robust estimators that use random forests are competitive in this setting. Here again, both double robust estimators give similar performances. No estimator achieves 95% coverage, which is expected given the non-linear, non-logistic DGP. Linear estimators, such as Linear RR-G and RR-AIPW, struggle to accurately estimate the nuisance functions in this context. Additionally, the limited number of observations prevents the Forest estimators from converging effectively.

## 8.3. Semi-synthetic simulations

In our semi-synthetic experiment, we retain the original co-variates, $X$, and first regenerate the treatment assignment, $T$, using a logistic regression (see Section 8.3 for random forests classifier) trained on the original 17 co-variates to predict the observed treatment. Based on the estimated probabilities from this model, we sample a binary treatment from a Bernoulli distribution. Next, we simulate potential outcomes using two separate logistic regressions: one for the treated group and another for the control group, incorporating all co-variates. This process yields the potential outcomes, $Y^{(0)}$ and $Y^{(1)}$, for each individual, enabling us to compute the true relative risk, $\tau_{RR} = 1.2$. Finally, we estimate the relative risk and its variance using bootstrap resampling across different sample sizes (300, 1000, and 3000). Results are displayed in Figure 11. All estimators except Neyman appear to converge to the true value of the Risk Ratio. As anticipated, doubly-robust estimators are consistent and efficient (smallest asymptotic variance).

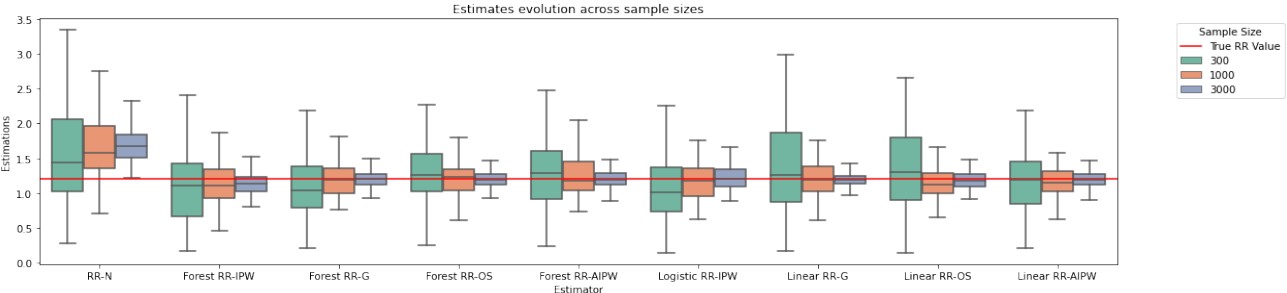

*Figure 11.* RR estimations with weighting, outcome based and augmented estimators as a function of the sample size for the logistic Semi-synthetic data. Parametric (Linear) and non parametric (Forest) estimations of nuisance are displayed.

In this approach, we replace logistic regression with random forest classifiers to model the treatment $T$ and the potential outcomes $Y^{(1)}$ and $Y^{(0)}$. As observed, all estimators, except for RR-N, converge toward the true value of the Risk Ratio. Moreover, since both the treatment assignment and potential outcomes are generated using Random Forest, estimators that also leverage Random Forest demonstrate reduced bias and lower variance. Conversely, linear estimators exhibit higher bias, likely due to model misspecification.

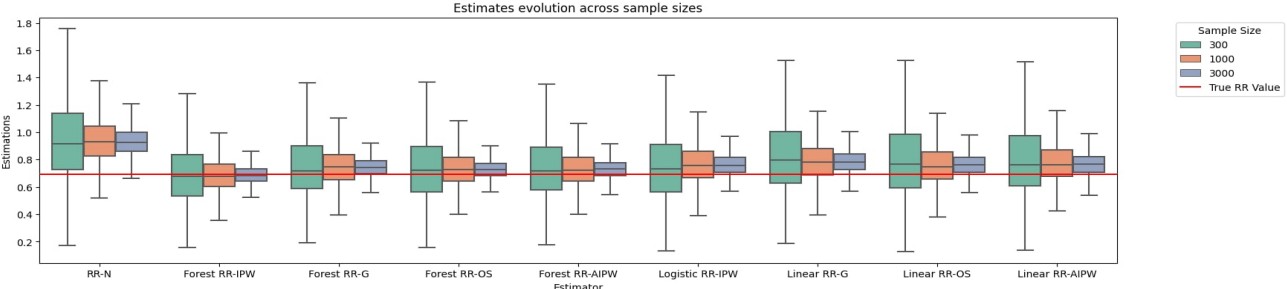

*Figure 12.* RR estimations with weighting, outcome based and augmented estimators as a function of the sample size for the Semi-synthetic data. Parametric (Linear) and non parametric (Forest) estimations of nuisance are displayed.

