# OpenReview forum: "Quantifying Treatment Effects: Estimating Risk Ratios via Observational Studies"
_ICML.cc/2025/Conference — ICML 2025 poster_

### Official Review · Reviewer_2hoQ · 2025-03-05

**Overall Recommendation:** 4

**Summary:**

This paper develops novel estimators for the Risk Ratio (RR) in observational studies to accommodate confounding in non-randomized settings. It introduces estimators based on inverse propensity weighting, the G-formula, and doubly robust techniques, and establishes their asymptotic normality. Simulation studies and a real-world application demonstrate that their estimators yield valid and efficient treatment effect estimates.

**Claims And Evidence:**

The claims made in the paper are well supported by both theoretical derivations and simulation studies.

**Essential References Not Discussed:**

All the essential references were discussed.

**Experimental Designs Or Analyses:**

N/A

**Methods And Evaluation Criteria:**

The proposed methods and evaluation criteria are standard and make sense for the problem at hand.

**Other Comments Or Suggestions:**

N/A

**Other Strengths And Weaknesses:**

The writing of figure captions could be improved for greater clarity and context.

For example, the caption for Figure 1 does not explain what the different colors represent.

**Questions For Authors:**

- Equation (17) requires consistency on both nuisance function estimators, yet in classic doubly robust settings it is often sufficient for only one to be consistent. Could you explain why both conditions are imposed here—does the non-linear nature of the risk ratio functional necessitate that both the propensity score and outcome model be estimated consistenly, or could a relaxation where only one of them is consistently estimated still yield valid inference?
- When the baseline risk is very low, the risk ratio may be unstable. Could you provide additional simulations to show the coverage of your estimator as the baseline risk approaches zero?

**Relation To Broader Scientific Literature:**

The paper’s key contributions are well presented and well connected to the broader literature on treatment effect estimation.

**Theoretical Claims:**

I did not check the correctness of any theoretical claims.

---

> ### Author Rebuttal · Authors · 2025-03-28
>
> - **Figure Caption Clarity:**
>   Thank you for pointing this out. We agree that the figure captions could be clearer. In particular, we will revise the caption to explicitly mention that the colors represent the "Sample Size".
>
> - **Doubly Robust Conditions:**
>   You are absolutely right that in classical doubly robust settings, it is sufficient for only one of the nuisance estimators (either the outcome regression or the propensity score) to be consistent for the overall estimator to remain consistent. This is commonly referred to as *weak double robustness*.
>
>   As discussed in Wager's *Causal Inference Book* ([Section 3](https://web.stanford.edu/~swager/causal_inf_book.pdf)), it is helpful to distinguish between **weak** and **strong** double robustness:
>
>   - **Weak double robustness** refers to the property that the estimator is consistent if either the outcome model or the propensity score model is estimated consistently. However, weak double robustness does not, in general, guarantee asymptotic normality or valid confidence intervals under model misspecification.
>   - **Strong double robustness**, on the other hand, refers to the setting where asymptotic normality can be achieved if both nuisance functions are estimated consistently at sufficiently fast rates. Specifically for the classical RD setting, this holds when both  $\hat \mu_{(t)}$ and $\hat{e}$ are estimated with root-mean squared error rates decaying faster than $n^{-\alpha_{\mu_t}}$ and $n^{-\alpha_e}$ respectively, and when $\alpha_{\mu_t} + \alpha_e \geq \frac{1}{2}$ for each $t \in \{0, 1\}$. Then AIPW is asymptotically normal.
>
>   In our paper, we establish *strong double robustness* for RR-AIPW. This result relies on both Equation (17) and Equation (14), which are strictly equivalent to the classical conditions for strong double robustness. It is worth noting that weak double robustness is also achieved for RR-AIPW but not for RR-OS since we always need $\hat{\mu}_{(0)}$ to be consistent.
>
> - **Low Baseline Risk:**
>   Thank you for raising this important point. When the baseline risk is extremely low, the risk ratio becomes inherently unstable, often resulting in wide confidence intervals. We have explored this empirically, and the results are shown in the table below.
>
>   As the baseline risk approaches zero, the estimated RR can become extremely large, while the absolute risk difference remains small: different causal measures lead to different interpretations, thus suggesting that several causal measures should be used simultaneously to properly understand the impact of a treatment.
>
>   This case also illustrates why it is important to report both an absolute measure (e.g., risk difference) and a relative one (e.g., risk ratio), as they provide complementary perspectives on treatment effects—particularly when baseline risks are extreme.
>
>   | **Baseline** | **RR**   | **Estimator**       | **Coverage (mean ± std)** | **Length (mean ± std)**   |
>   |--------------|----------|---------------------|----------------------------|----------------------------|
>   | 2.5          | 1.77     | Linear RR-G         | 0.878 ± 0.113              | 0.099 ± 0.002              |
>   | 2.5          | 1.77     | OLS RR-G            | 0.948 ± 0.050              | 0.120 ± 0.003              |
>   | 2.5          | 1.77     | Linear RR-AIPW      | 0.968 ± 0.047              | 0.185 ± 0.007              |
>   | 0.045        | 15.3     | Linear RR-G         | 0.992 ± 0.027              | 3.728 ± 0.513              |
>   | 0.045        | 15.3     | OLS RR-G            | 0.984 ± 0.037              | 3.037 ± 0.435              |
>   | 0.045        | 15.3     | Linear RR-AIPW      | 1.000 ± 0.000              | 4.734 ± 0.735              |
>   | 0.0018       | 112.1    | Linear RR-G         | 0.987 ± 0.022              | 884.427 ± 206.672          |
>   | 0.0018       | 112.1    | OLS RR-G            | 0.978 ± 0.032              | 1026.384 ± 214.754         |
>   | 0.0018       | 112.1    | Linear RR-AIPW      | 0.972 ± 0.036              | 2073.229 ± 509.709         |
>
>   **Table:** Coverage and length of confidence intervals for different RR estimators across varying baseline risks.

---

### Official Review · Reviewer_7moV · 2025-03-13

**Overall Recommendation:** 1

**Summary:**

The authors propose several estimators for the average risk ratio (RR). They begin by analyzing the RR version of the Neyman estimator under standard causal inference assumptions, proving its asymptotic normality and deriving an expression for its variance. They then extend this analysis to an RR variant of the Horvitz-Thompson estimator.

Next, the authors refine their results by assuming the true propensity scores follow a logistic model and fitting estimators using maximum likelihood estimation (MLE). They further establish the asymptotic normality and derive an explicit variance expression for an RR version of the G-formula—first using the true feature probabilities and then with an estimator where the conditional expectation is obtained via least squares.

The paper concludes with simulations and a real-world application using a dataset on traumatic brain injury.

Strengths
-The paper is well-executed, providing a thorough set of theoretical results and strong empirical validation, including a real-world case study.

Weaknesses
-The properties of RR estimators may already be well known, albeit I am not a causal inference expert.

**Claims And Evidence:**

Refer to summary

**Essential References Not Discussed:**

refer to Broader Scientific Literature

**Experimental Designs Or Analyses:**

Refer to summary

**Methods And Evaluation Criteria:**

Refer to summary

**Other Comments Or Suggestions:**

No extra comments

**Other Strengths And Weaknesses:**

Refer to summary

**Questions For Authors:**

No questions for the authors

**Relation To Broader Scientific Literature:**

In the introduction of this paper by Rose and Van Der Laan (2014) https://sci-hub.ru/https://doi.org/10.1093/aje/kwt318 , they talk about how there are methods to estimate the risk-ratio. Also I think Kennedy has a paper where he proves properties of the risk ratio condoned on the features.

**Theoretical Claims:**

Refer to summary

---

> ### Author Rebuttal · Authors · 2025-03-28
>
> - **Clarifying the Novelty and Contribution:**
>   We thank you for acknowledging the thoroughness of both the theoretical development and empirical validation. We would like to take this opportunity to clarify the main contribution of the paper.
>
>   One of the main contributions of this paper is to provide a comprehensive theoretical and empirical analysis of RR estimators, including new ones like RR-OS and RR-AIPW, in the context of observational studies. While RR is a well-established estimand, its estimation introduces distinct challenges due to its non-linearity.
>
>   To the best of our knowledge, there has not been a unified and rigorous asymptotic analysis of IPW, G-formula, or AIPW estimators for the marginal RR in the literature. Our work aims to fill this gap by deriving explicit variance expressions, exploring efficiency bounds, and validating the estimators both theoretically and empirically in both continuous and binary outcome settings.
>
> - **On the Work by Rose and van der Laan (2014):**
>   The cited work by Rose and van der Laan focuses on Targeted Maximum Likelihood Estimation (TMLE) for the risk difference in case-control studies. While this is an important contribution, the setting considered there differs from the randomized and observational study framework analyzed in our work and focuses on binary outcomes. Furthermore, while TMLE is a flexible approach, our paper focuses on IPW, AIPW, and G-formula estimators, whose asymptotic properties have not, to the best of our knowledge, been rigorously established in the context of marginal RR estimation using observational data — also for continuous outcomes. We thank the reviewer for pointing out this reference; we will mention it in the conclusion as a potential research direction for finite sample studies.
>
> - **On the Mention of Kennedy’s Work:**
>   We are not entirely sure what specific paper by Kennedy you refer to. We have read several papers by Kennedy, three of which are cited in our work, but we did not encounter specific studies by Kennedy (or anyone else) analyzing different estimators of the marginal risk ratio. That said, it is worth noting that the conditional risk ratio is not directly collapsible. This means that estimating the conditional RR generally does not provide a valid estimator for the marginal RR. In contrast, our work specifically targets estimators for the marginal RR and provides a detailed asymptotic analysis under standard causal inference assumptions.
>
> We hope that the above points help to dispel your doubts about the novelty of our work.

---

### Official Review · Reviewer_V8v7 · 2025-03-14

**Overall Recommendation:** 4

**Summary:**

The authors discussed theory of risk ratio estimation in observational data. Several RR estimators are proposed with theoretical investigation, including asymptotic normality and confidence intervals. Two doubly robust estimators are proposed and the authors recommended the use of one of them.

**Claims And Evidence:**

Yes

**Essential References Not Discussed:**

Not aware of any.

**Experimental Designs Or Analyses:**

Yes - the experimental design makes sense to me.

**Methods And Evaluation Criteria:**

Yes

**Other Comments Or Suggestions:**

1. Assumption 2.2, part (iii): independence between treatment assignment is already guaranteed by iid part.

**Other Strengths And Weaknesses:**

Strength: the work is very clearly written, and is a good addition to the cauasl inference literature for estimation of RR. I am very familiar with the observational study literature and the results for the different estimators are pretty natural. Overall I am positive about the paper and I will skip technical questions here.

Weakness:
1. Though in assumption it is assumes that conditional expectation of $Y(0)$ given $X$ is greater than zero, this is never guaranteed in finite sample using say AIPW. How should one control for this?
2. in term of simulation, I am interested in seeing the performance of RR-OS estimator and RR-AIPW under some wrong model misspecifications.
3. the whole paper did not discuss the implementation of several variance estimators. For example, how do we implement $V_{RR,OS}$.

**Questions For Authors:**

N/A

**Relation To Broader Scientific Literature:**

Adding contribution to estimation of risk ratio in observation study.

**Theoretical Claims:**

Yes, I have checked all the proof.

---

> ### Author Rebuttal · Authors · 2025-03-28
>
> Thank you for your thoughtful and constructive review. We appreciate your positive comments on the clarity of the writing and the contribution of our work to the causal inference literature. Please find our responses to your comments below:
>
> 1. **Regarding the assumption $E[Y(0) \mid X] > 0$:**
>
>    Our initial motivation for introducing it was to establish finite-sample bias and variance results, which are currently included only in the appendix. However, for the main theoretical results, we only need to assume $E[Y(0)] > 0$ to ensure that the risk ratio is well-defined.
>
>    We agree that this assumption may not always hold, but in practice, many biological quantities are positive (e.g., blood concentrations). Besides, if the variable $Y(0)$ is binary, $E[Y(0)] > 0$ holds as soon as one observation satisfies $Y_i(0) = 1$, which seems to be a reasonable assumption.
>
>    In the revised manuscript, we will move the stronger assumption $E[Y(0) \mid X] > 0$ to the appendix (where it is needed) and replace it in the main text with the weaker and more general condition $E[Y(0)] > 0$. We will also add the above discussion.
>
> 2. **On model misspecification in simulations:**
>
>    We completely agree that comparing the estimators under various forms of model misspecification is important. Due to space constraints, we were unable to include this analysis in the main body of the paper, but we incorporated two relevant scenarios in Appendix 8.2:
>
>    - **Wager C model (Appendix 8.2.1):**
>      Both response functions are nonlinear, while the propensity score is logistic. In this setting, Logistic RR-IPW and Linear RR-AIPW and RR-OS (using linear and logistic regressions) remain asymptotically unbiased, while Forest RR-OS and Forest RR-AIPW are biased for small sample sizes due to the difficulty of forests in estimating logistic functions with few observations.
>
>    - **Wager A model (Appendix 8.2.2):**
>      Both response functions and the propensity score are nonlinear/non-logistic. All linear estimators are biased in this case, but estimators that use Random Forests to estimate the nuisance functions exhibit decreasing bias as the sample size increases.
>
>    If the paper is accepted, we will use the additional page to move this analysis into the main paper.
>
> 3. **On the implementation of variance estimators:**
>
>    Thank you for pointing this out. In fact, we designed variance estimation procedures to construct the confidence intervals reported in the experiments, but we recognize that the implementation details were not highlighted enough.
>
>    In the current version, the relevant formulas are in the appendix (in Section 4.2). We will elaborate on this and add a dedicated section in the Appendix. In addition, we will better highlight in the main text that we have estimated all variances, as they can all be expressed using estimable quantities, and that we used these estimations in all our simulations.
>
> 4. **Redundancy in Assumption 2.2:**
>
>    You are totally right. We will revise this assumption to remove the redundancy. Thank you for pointing this out!

---

> > ### Comment · Reviewer_V8v7 · 2025-04-03
> >
> > Thanks for the careful responses. The authors have addressed my concerns and I have no further comments/questions.

---

### Official Review · Reviewer_KPHb · 2025-03-15

**Overall Recommendation:** 4

**Summary:**

Authors focus on risk ratio (RR), a measure of treatment effectiveness complementary to the more common "risk difference". They first analyze the standard estimators for RR in an RCT and derive some new asymptotic normality/variance results, such as for continuous outcomes in addition to binary outcomes. They then develop IPW, regression function, and AIPW estimators for the RR in the context of an observational study where one needs to adjust for confounders and to estimate nuisance functions. They derive asymptotic unbiasedness and normality results of their estimators under the assumption that the true underlying outcome and propensity score models are (generalized) linear models and derive an expression for the asymptotic variance. They also derive, using the influence function theory, and efficient AIPW estimator.

**Claims And Evidence:**

Theoretical results are well presented and supported.

**Essential References Not Discussed:**

I am not an expert on the literature about RR, but was satisfied with how the related work was covered.

**Experimental Designs Or Analyses:**

Synthetic setup makes sense and supports the asymptotic unbiasedness results of proposed algorithms, except for the Logistic RR-IPW. Although that is not surprising to me as IPW estimators almost always have very high variance with finite samples in practice. Authors could also plot histograms instead of boxplots to demonstrate asymptotic normality of their estimators.

**Methods And Evaluation Criteria:**

Yes. See experiments and benchmarks section below.

**Other Comments Or Suggestions:**

NA

**Other Strengths And Weaknesses:**

The paper is very well-written and structured. It is self-contained in terms of the statistical methodology it uses, which could be very helpful to those who are not very familiar this type of (e.g., CLT) results/derivations.

**Questions For Authors:**

NA

**Relation To Broader Scientific Literature:**

Formal results on RR estimators could be very useful to practitioners in practice. Methodologically, most of the techniques used are standard and does not reveal new insights, except may be for the results that require the derivation of influence functions for the RR estimand.

**Theoretical Claims:**

I did not check in detail. The parts I quickly skimmed looked fine and the proofs in the appendix seem to be well-written.

---

> ### Author Rebuttal · Authors · 2025-03-28
>
> Thank you very much for your positive review. We are glad that you found the paper well-written, well-structured with interesting results for practitioners. We also appreciate your suggestion regarding the use of histograms to illustrate the asymptotic normality of our estimators. We agree that this would provide a clearer visual understanding of the results and plan to incorporate it in the next revision.

---

### Decision · Program_Chairs · 2025-05-01

**Decision:**

Accept (poster)

**Comment:**

This paper studies the estimation of the risk ratio in observational studies under the assumption of confoundedness. This propose and compare several estimators for the risk ratio. However, the proposed estimators and the corresponding conclusions are fairly standard and closely parallel existing work on estimating the risk difference.

Four reviewers evaluated this work, but there was some discrepancy in their scores (1, 4, 4, 4). After discussing with the reviewers, the key concern centers on whether prior work has addressed a similar problem.  We found a paper that addresses an estimand (Lemma 1(a) of [1]) similar to the risk ratio. The work in [1] proposes a semiparametric efficient (doubly robust) estimator for this estimand. Since this is an ArXiv paper unpublished, we decided to accept this paper. However, the authors are encouraged to incorporate the reviewers' suggestions and include a brief discussion with [1] in the final version.


[1] Tian and Wu. Semiparametric Efficient Inference for the Probability of Necessary and Sufficient Causation. arXiv, 2024.